# Coexistence of state, choice, and sensory integration coding in barrel cortex LII/III

Pierre-Marie Gardères [1,2] ✉, Sébastien Le Gal[1], Charly Rousseau[1], Alexandre Mamane[1], Dan Alin Ganea[2,3] & Florent Haiss [1] ✉

During perceptually guided decisions, correlates of choice are found as upstream as in the primary sensory areas. However, how well these choice signals align with early sensory representations, a prerequisite for their interpretation as feedforward substrates of perception, remains an open question. We designed a two alternative forced choice task (2AFC) in which male mice compared stimulation frequencies applied to two adjacent vibrissae. The optogenetic silencing of individual columns in the primary somatosensory cortex (wS1) resulted in predicted shifts of psychometric functions, demonstrating that perception depends on focal, early sensory representations. Functional imaging of layer II/III single neurons revealed mixed coding of stimuli, choices and engagement in the task. Neurons with multi-whisker suppression display improved sensory discrimination and had their activity increased during engagement in the task, enhancing selectively representation of the signals relevant to solving the task. From trial to trial, representation of stimuli and choice varied substantially, but mostly orthogonally to each other, suggesting that perceptual variability does not originate from wS1 fluctuations but rather from downstream areas. Together, our results highlight the role of primary sensory areas in forming a reliable sensory substrate that could be used for flexible downstream decision processes.

The brain guides the body by processing incoming sensory information and using it to select contextually relevant actions. This perceptually dependent process has been shown to involve at least two components: the encoding of sensory evidence, transiently elicited in sensory areas, and the progressive accumulation of motor variables in a distributed network of decisional and motor areas[1–3]. To understand where and how sensory representations inform decisional processes, it is necessary to identify features of the neuronal activity that encode simultaneously sensory and choice information.

Pioneering studies addressed that issue using two-choice discrimination tasks, where primates had to judge the motion direction of a cloud of points[4,5]. Perception was reported by choosing one of two possible actions. These studies found that trial-by-trial fluctuations of

neuronal activity in the visual areas predicted choices of the animal, i.e., display choice-related activity[5]. The firing rate of neurons in sensory areas is hence thought to carry the code for sensory representation informing behavior. In the somatosensory system of the primate, similar approaches have revealed that choice-related activity arises from the secondary somatosensory area and is higher in the decisional hierarchy[2,6]. More recently, research in rodents has identified choice-related activity forming already at the level of the primary somatosensory area (S1)[7,8], It is yet difficult to conclude that the choice signals in S1 have a causal influence on behavior and perception.

First, many studies in the rodent model used a go/no-go paradigm and studied the detection of a stimulus close to the detection threshold. The widespread cortical activity related to the onset of

[1]Institut Pasteur, Université Paris Cité, Unit of Neural Circuits Dynamics and Decision Making, F-75015 Paris, France. [2]IZKF Aachen, Medical School, RWTH Aachen University, 52074 Aachen, Germany. [3]University of Basel, Department of Biomedicine, 4001 Basel, Switzerland. ✉e-mail: pmgarderes@gmail.com; florent.haiss@pasteur.fr

facial movement[9,10]–including in sensory areas–renders ambiguous the signals associated with choice if sensory integration and motor response are temporally overlapping. These approaches are also relatively vulnerable to animal biases and changes in motivation. It has thus been proposed that more complex designs, such as two alternative forced choice tasks (2AFC) featuring a delayed response, are needed to disentangle choice from these other sources of modulation[11].

Perhaps more importantly, choice signals are generally described as increases in activity indiscriminately in a fraction of sensory or non-sensory neurons. If a sensory representation is used to inform the behavior, its fluctuation from trial to trial should have an impact on the choice. In other terms, the sensory code, rather than the activity in the area, should co-fluctuate with the behavior[12]. Sensory and choice coding in the rodent S1 has been studied mostly separately, or at the level of single neurons. Therefore, a description of how choice versus early sensory representation is differentially organized in S1 neuronal populations is lacking.

To tackle these challenges, the focal and quasi-discrete sensory representation of single whiskers in S1 represents a considerable advantage. Here we developed a novel two alternative forced choice task that would allow us to monitor and manipulate the representations of the two competing sensory alternatives.

## Results

### Discrimination behavior of vibrotactile frequencies applied to adjacent whiskers

Mice were water-deprived and required to compare two frequencies (F1 and F2) delivered simultaneously to two adjacent whiskers on the same column of the whisker pad (e.g., B1/C1). The two target whiskers, designated thereafter as W1 and W2, were respectively associated with left and right waterspouts, designated thereafter as choice 1 and choice 2 (Fig. 1a). Tactile stimulation lasted 1 s after which waterspouts came in a reachable position. At that time, the mice were allowed to lick during a two-second decision period (Fig. 1b). A water reward was delivered if the subject first licked the spout associated with the whisker deflected at the highest frequency. Importantly, in this task, the frequencies are proportional to the average speed of whisker deflection and thus directly represent the intensity of the stimulation[13]. While the subject was performing the task, we recorded videos of its face and snout, allowing us to track multiple behavioral variables such as the pupil size, movement from the whisker, nose, and tongue (Fig. 1b; see Methods and Supplementary Movie 1).

The initial training phase for the 2AFC started with single whisker stimulation only and subjects were trained to respect the delay to the end of the stimulus (Fig. 1c). Once they reliably scored 70% in that task, F1 and F2 were set to vary in a range of frequencies going from 0 to 90 Hz, defining the stimulus space (Fig. 1d). To quantify the relationship between the stimulus and the sensory capabilities of mice, we fitted the probability of F1 being called higher with a psychometric function[14] which reveals a reliable dependence of the choice side on the stimulation frequency difference $\Delta F = F1 - F2$ (Fig. 1e) across animals. $\Delta F$ is the dimension explaining most performance across the stimulus space (Fig. S1). Importantly, mice were able to compare the frequencies at all points tested in the stimulus space (Fig. 1d). The same stimulus (e.g., F1 = 50 Hz) could be contextually categorized as a distractor or target, excluding that the task is solved based on the recognition of a specific target stimulus only[15]. Instead, we conclude that mice were capable of generalizing the abstract rule of frequency comparison across the entire stimulus space.

To distinguish the different behavioral outcomes, we used the animal's licking behavior and its facial movements. The 2AFC task design leads to three possible trial outcomes: correct trials, error trials, and no response, when no spouts are licked during the decision period (Fig. 1g). Most of the no-response trials were recorded in blocks at the end of the daily session, so we hypothesized a different state of task disengagement. This view is strengthened by a physiological marker, as we observe larger pupillary dilation in the baseline epoch and weaker pupillary response to stimulation, consistent with previous report[16] (baseline dilation $p = 0.036$, the difference between Correct and Miss, $n = 4$ mice, Friedmann test with Tuckey's post hoc correction; Fig. 1h). We found that within the engaged state, nose movements are the most indicative of the animal's earliest reaction and decision (Fig. S2). We thus used their onset as a proxy for the first movement executed by the animal in each trial. Accordingly, we distinguished three trial categories: first, trials with movements prior to stimulus onset ($24.4 \pm 3.1\%$ trials; mean $\pm$ s.e.m across animals, $n = 7$ mice.) which are excluded from further analysis; second, impulsive trials with movements prior to the end of the delay ($41.2 \pm 2.9\%$ trials), and third, trials in which the animals refrain from moving until the decision period ($34.4 \pm 5.4\%$ of the trials, no significant movement of the snout, tongue or whiskers, Fig. S2). Therefore, the delayed 2AFC task design with video tracking enables the distinction of engaged versus disengaged states and enables the separation of a motor from choice variables in a subset of trials with controlled delayed responses.

### Somatotopic whisker representation and increased activity level with stimulation frequencies

To characterize S1 sensory representations, we used two-photon calcium imaging guided by intrinsic optical imaging (IOI, Fig. 2a–c, see Methods), recording hundreds of L2/3 neurons simultaneously in the same field of view ($337 \pm 20$ neurons per FOV). Transient fluorescence responses evoked by single whisker deflections matched the spatial arrangement of IOI maximum intensity (Fig. 2c). We considered the activity of single neurons in response to single and dual whisker stimulation frequencies. Fluorescence traces were detrended, and firing rate (FR) was inferred using a deconvolution algorithm (see Methods; Fig. 2b). Consistent with previous studies[17–19], we found that a small fraction of neurons was highly responsive to tactile stimulation (~10%; Fig. 2c). Yet, a larger fraction of the population had significant response compared to baseline: $32.3 \pm 4.8\%$ of neurons were suppressed and $52.3 \pm 4.7\%$ of neurons were activated, mean $\pm$ s.e.m across $n = 11$ FOV, all trials included. Neurons responding to one whisker showed increased probability to also respond to the other whisker (Spearman's $r = 0.49$; $p < 0.001$; similarly to previous reports[20]), but paradoxically the fraction of most responsive neurons (~10%) displayed the highest whisker selectivity index (SI) and were matching more closely the somatotopic map (Fig. 2d and Fig. S3). In a separate set of animals with conditional genetic expression of GFP in GABAergic neurons (see methods and Supplementary Table 1), we quantified selectivity and spatial distribution of labeled inhibitory neurons (INs) and putative excitatory neurons (ENs) (see methods; Fig. 2e). IN population selectivity indexes were lower than the EN population but ranged on a similar scale ($p = 0.05$, $|\mu SI_{IC}| = 0.33$, $|\mu SI_{EC}| = 0.40$ LME model test, Fig. 2e). Whisker selectivity in both ENs and INs was intermixed spatially in a salt-and-pepper organization with a graded transition between the two columns (Fig. 2e). These results highlight the functional selectivity of inhibitory and excitatory neurons in two neighboring microcircuits with heterogeneous selectivity at the single neuron level. The spatial transition of functional selectivity illustrates the underlying somatotopic organization in columns.

To understand how stimulation frequencies influence these microcircuits' response level, we split the neuronal population in two, based on their preferred whisker (SI >0 and SI <0). In response to their preferred whiskers stimulation at 90 Hz, the subpopulations instantaneous firing rate increased by a factor of 3.3 immediately after stimulus onset, and then adapted but remained over baseline activity level (increased by factor $1.8 \pm 0.2$, mean $\pm$ s.e.m., 0.5 to 1 s after stimulus onset, $n = 11$ FOVs; Fig. 2f). Importantly, in response to increasing frequencies, we observe an increase in the average population

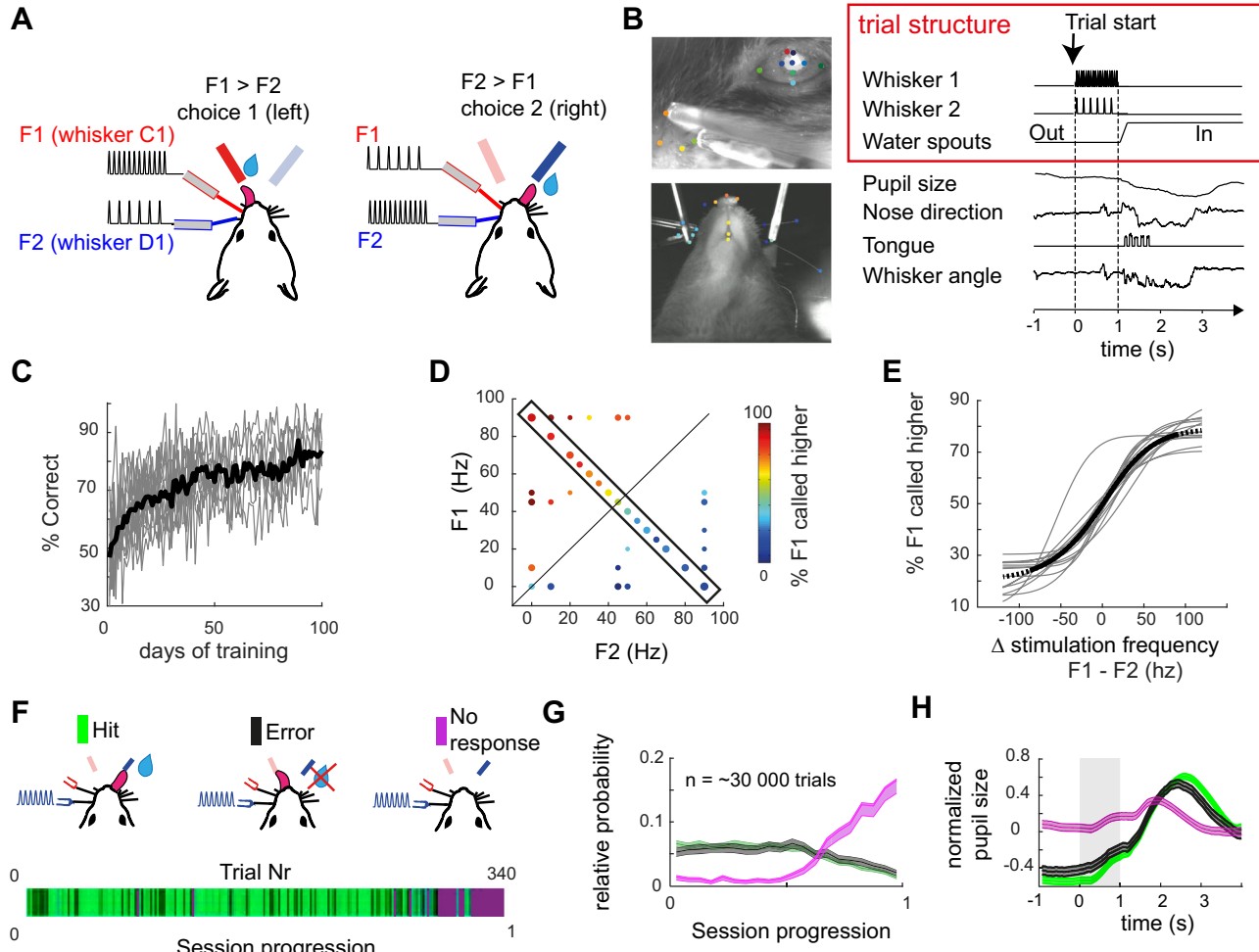

**Fig. 1 | Performance and behavior during discrimination of simultaneous vibrotactile stimulation applied to adjacent whiskers. A** Schematic of the task. left: W1 whisker is deflected with higher frequency than W2, left licks trigger a reward delivery. Right: F2 > F1, licks to the right trigger a reward delivery. **B** Face tracking and trial structure. Left: two example views simultaneously recorded. Some face parts are tracked using DeepLabCut (ref. 87) and are labeled with random colors (see methods). Right: trial time structure with example face tracking (downsampled to 30 Hz). Waterspouts come in a reachable position at the end of the whisker stimulation. Scales are normalized between minimum and maximum amplitudes. **C** Learning curves of 15 individual mice. Calcium imaging and further analysis were carried out on a subset of sessions after the animal was considered expert (reached >70% Correct trials). **D** Behavioral performance in the stimulus space. Each point represents a set of two frequencies applied to the two whiskers.

The diagonal represents the task boundary (F1 = F2). **E** Psychometric function drawn from conditions within the rectangle in **D** (i.e., with a constant summed frequency F1 + F2 = 90 Hz). Traces are psychometric fits for individual animals (see methods). The black thick line represents average fit. **F** Three possible behavioral outcomes in one example session. Top: 3 possible behavioral outcomes: first lick right, first lick left, and no lick during the response window. Bottom: example session with behavioral outcome color coded. The trial index is a normalization of trial number between 0 and 1 (respectively first to last trial of the session). **G** Relative distribution of hits, errors, and misses within the daily session progression (average across sessions and mice), normalized by category represented as mean values ± s.e.m. Misses typically appeared in blocks at the end of the session. **H** Normalized pupil size in the pre-stimulus period predicts engagement of the subject; represented as mean values ± s.e.m.

activity (Fig. 2f, g, principal population). Adjacent populations respond positively but weakly to the stimulation of the principal whisker alone. Finally, adding increasing adjacent whisker's stimulation over the principal whisker's stimulation held constant, seemed to lower the response of the principal whisker's population (Fig. 2g), suggesting a suppressive effect of multi-whisker stimulation. As frequencies are directly proportional to the mean speed deflection of the whisker, frequencies also represent the intensity of stimulation[13]. In accordance with this concept, our data show that increasing frequencies are represented by a monotonical increase in activity level in the principal whisker column.

### Neural activity in a single wS1 cortical column drives the perception of the whisker stimulation intensity

As the two sensory alternatives have distinct spatial representations, we evaluated the possibility of manipulating them selectively. Optogenetic

widefield stimulation was combined with two-photon functional imaging[21] (Fig. 3a; see methods). Channelrhodopsin was conditionally expressed in GABAergic neurons (VGAT cre-line), whose optogenetic stimulation reduced the activity in the local network. The efficiency and selectivity of optogenetic stimulation was surveyed in four distinct conditions: no light, broad illumination of the two barrels, and selective stimulation of either single barrel (Fig. 3b, c). Across layers of the neocortex, light was shaped in a cylindrical manner (Fig. S4). Maximum light intensity stimulation (17.9 mW mm² with 450 μm light disk, as measured below the objective) during epochs of spontaneous activity induced a significant increase in activity for 23.2% cells, being putative IN, and a significant decrease for 20.8% cells, being putative EN ($p < 0.01$; activity between light and no light, $t$-test). These proportions were dependent on the distance to the illumination center with a spread of inhibition beyond the cortical column limit, possibly due to direct inhibition or network-dependent effects (Fig. S4). We then explored how optogenetic

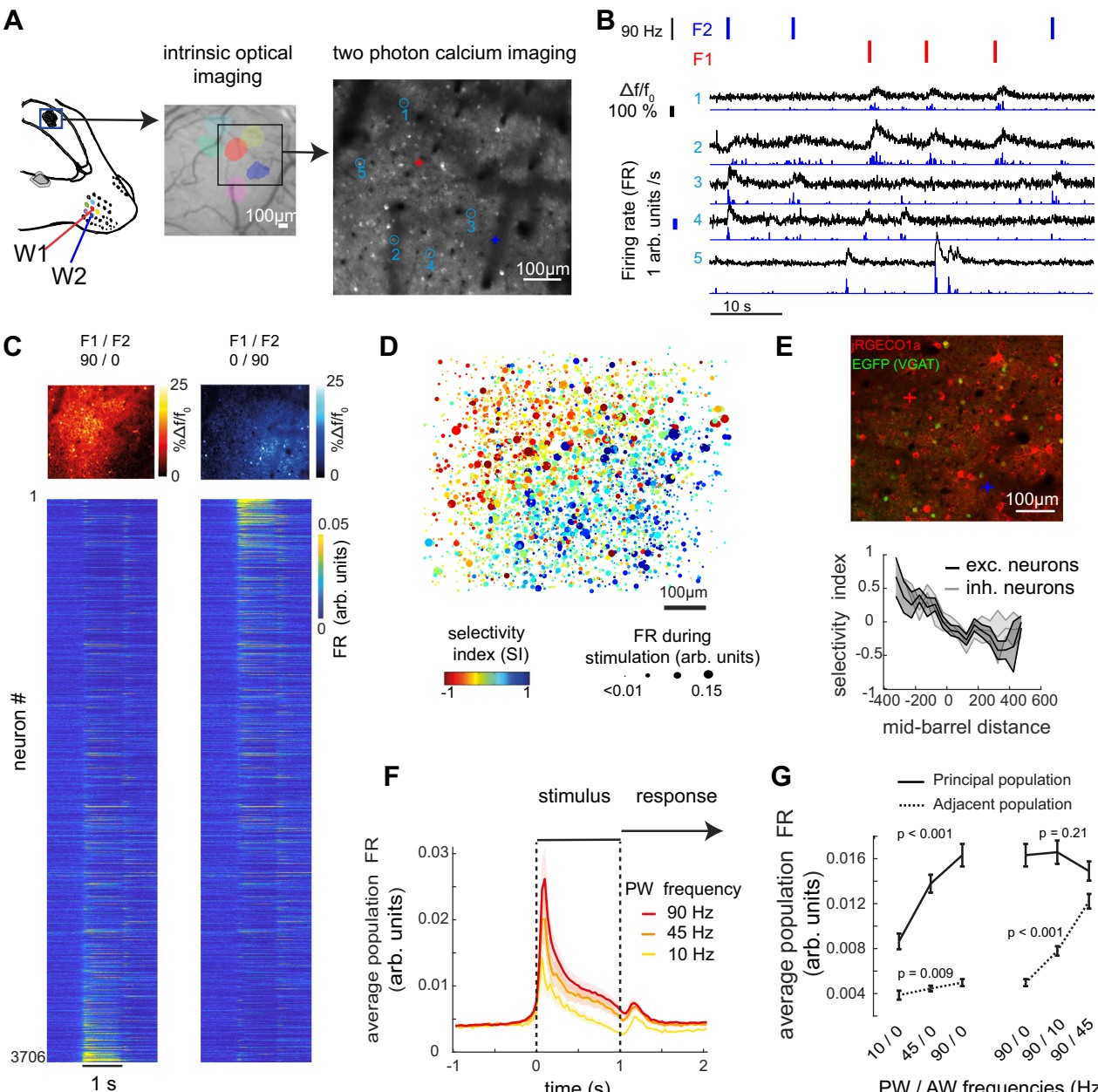

**Fig. 2 | Activity levels in neighboring cortical columns increase with stimulation frequencies of their preferred whisker. A** Simultaneous Imaging of W1 and W2 cortical representations in wS1 L2/3. Left: Contralateral recording in the head-fixed mouse. Middle: Example Intrinsic optical imaging (6 replications with similar results). Right: Example Ca2+ imaging field of view (FOV) spanning the two barrels; blue circles indicate example ROIs for which calcium transients are depicted in (**D**). **B** Five example neurons fluorescence (black) and deconvolved firing rate traces (blue). **C** Response to single whisker stimulation at 90 Hz (W1 and W2, top and bottom rows respectively). Left: pixel-wise change in fluorescence $\Delta f/f0$ is color-coded. Right: average FR of each neuron in response to single whisker stimuli. Neurons are sorted by increasing $FR_1$ - $FR_2$, with $FR_1$ and $FR_2$ representing firing rate during 90 Hz stimulation of W1 and W2. Same sorting order in top and bottom plots. **D** All neurons aligned to the mid-points between cortical columns. Selectivity index (SI) is represented by color and the average FR is represented as dot size. SI = $(FR_1 + FR_2)/ (FR_1 - FR_2)$. **E** Inhibitory versus excitatory response to whisker stimulation. Top: Example FOV of a VGAT-cre mice. Cortical column centers are marked by a cross. Bottom: SI for INs and putative ENs ($n$ = 267 and 974 neurons, respectively, from three animals). SI are represented as mean ± CI95 across neurons in bins of 50 µm. **F** Population average FR in response to increasing single whisker stimulation frequency. Error shades represent s.e.m. across FOVs, $n$ = 11 FOV from seven animals. **G** Average FR of the principal whisker (PW) population and adjacent whisker (AW) population in response to single and multi-whisker stimulation. The statistical test represents the increase or decrease of population activity with the stimulation frequency of the PW alone (left) or AW while PW is held to 90 Hz stimulation (right). ***$p < 0.001$ (LME model test). $N$ = 18 columns from 9 FOV. FR is represented as mean ± CI95 across columns. Figure 2A left panel is adapted from C S Barz, P M Garderes, D A Ganea, S Reischauer, D Feldmeyer, F Haiss, Functional and structural properties of highly responsive somatosensory neurons in mouse barrel cortex, Cerebral Cortex, Volume 31, Issue 10, October 2021, Pages 4533–4553, https://doi.org/10.1093/cercor/bhab104 and released under a Creative Commons CC-BY-NC license: https://creativecommons.org/licenses/by-nc/4.0/.

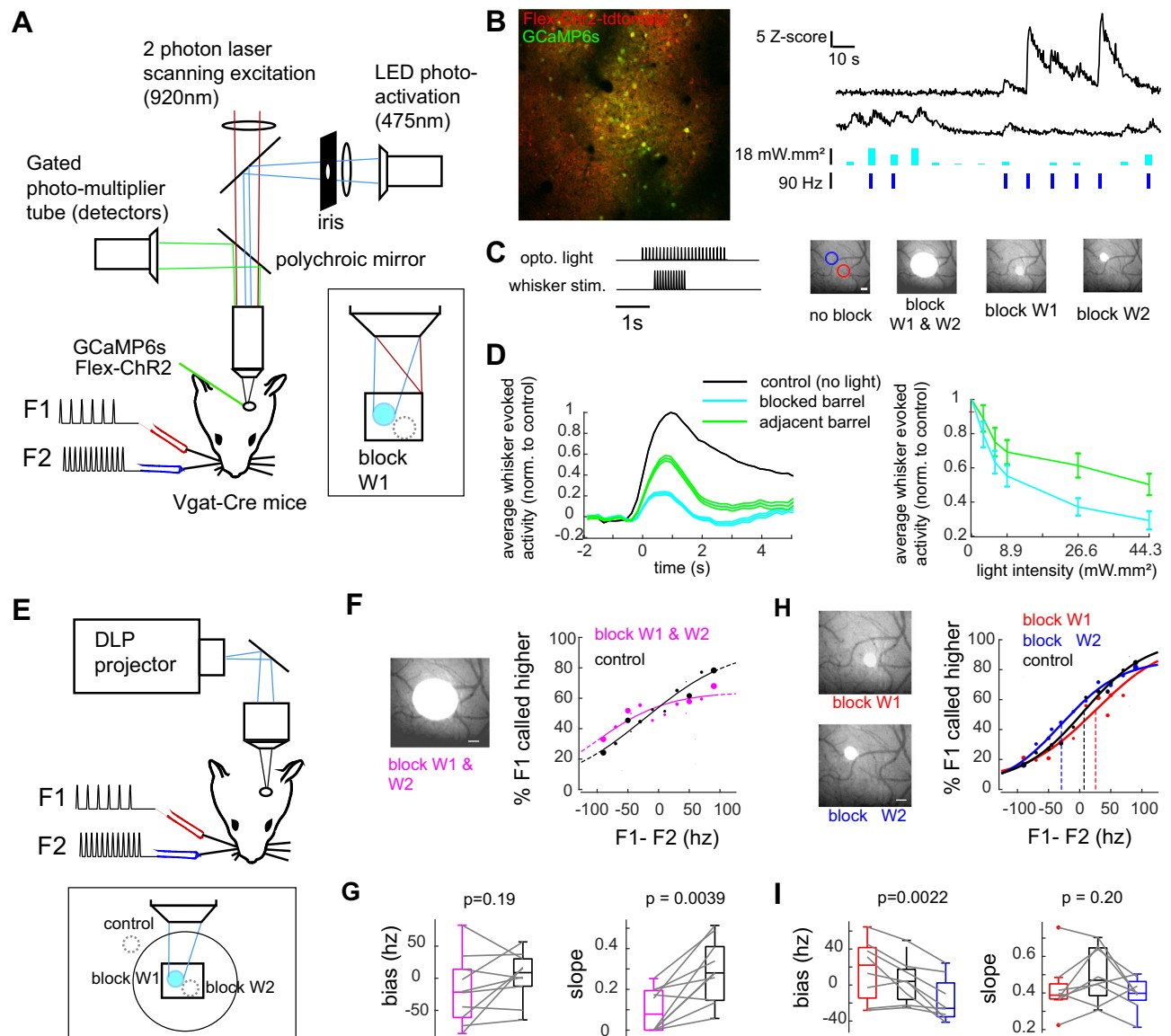

**Fig. 3 | Differential optogenetic inhibition of a single barrel induces selective shifts in perception. A** Setup for combined optogenetic and calcium imaging (see methods). **B** Left: Example FOV of one VGAT-cre animal expressing GCaMP6s and ChR2. Right: example fluorescence transients in response to whisker and light stimulation for a putative excitatory neuron (top) and a putative inhibitory neuron (bottom). Light and whisker stimulation amplitude are indicated below. **C** Optogenetic stimulation parameters. Left: Trial structure. Right: Light patterns, from left to right: - no block (W1 and W2 columns are illustrated as 200 μm diameter circles); - Illumination of the two columns (450 μm diameter disk, FWHM); - selective illumination of W1 column (105 μm diameter disk, FWHM); - selective illumination of W2 column. **D** Impact of selective inhibition (105 μm diameter disk) onto targeted and adjacent barrels evoked response (*n* = 901 neurons from three animals), during 90 Hz stimulation of the preferred whisker. Inhibitory neurons and non-whisker responsive neurons were excluded from the analysis (see details in the

methods). Left: time course of whisker evoked response. Right: mean residual activity across different optogenetic light intensities. Data were represented as mean values ± CI95. **E** Setup for optogenetic during behavior. A DLP projector was used to display light patterns on the cortex. **F** Non-selective barrel inhibition. Psychometric fit during intermingled trials with either sham optogenetic (black) and 450 μm diameter disk optogenetic blocking (magenta). **G** Psychometric slope and bias quantification during non-selective column inhibition Friedman test, *n* = 10 mice. **H** Selective barrel inhibition. psychometric fit during intermingled trials with either inhibition of the W1 barrel (red), inhibition of the W2 barrel (blue), or sham control (black). Data pooled from eight mice. **I** Psychometric slope and bias quantification during selective column inhibition. In **F–I**, Optogenetic trials were paired with the closest occurring sham control trial having the same stimulation frequencies. Trials were matched for all statistical analysis and psychometric fits. Friedman test, *n* = 8 mice.

stimulation of a single barrel INs influences the sensory signals associated with single whisker deflection (Fig. 3d, see methods). Optogenetic stimulation decreased sensory response to the preferred whisker in both target and neighboring columns. But the reduction of sensory activity was always stronger for the optogenetically targeted barrel (Fig. 3d). Taken together, these results indicate that it is possible to differentially manipulate the two sub-ensembles, shifting bidirectionally the ratio of response toward one or the other whisker representation.

To identify the involvement of S1 in the sensory-guided decision, trials with and without optogenetic stimulation were interleaved during the behavioral task (Fig. 3 e–i). Optogenetic stimulation was provided in 25% of the trials, versus a control condition in which light was pointed on the dental cement next to the brain (Fig. 3e). In response to the broad stimulation pattern, we found that the behavioral discrimination was significantly impaired in all stimulation conditions as compared to the control condition (slope of the psychometric fit;

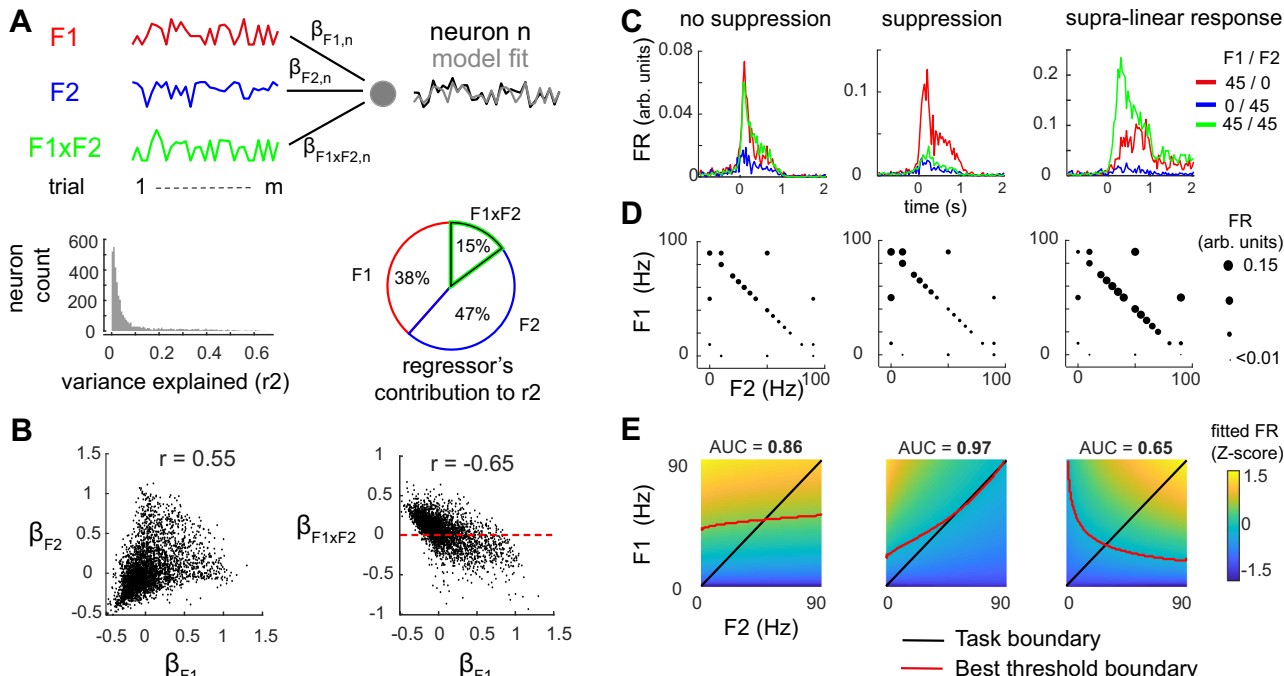

**Fig. 4 | Multi-whisker suppression improves frequency comparison performance in the stimulus space. A** Linear regression of single-neuron activity on whisker stimulation frequencies. Top row: Example regression of one neuron in 20 trials (see text and methods). Bottom left: fraction of explained variance across neurons. Bottom right: fraction of variance explained by each regressor (average over $n = 3706$ neurons), relative to the total variance explained by the model. **B** Cell by cell distribution of model weight. Note the positive correlation between single whisker weights (left) and the negative correlation between single and interaction weights (right). **C**–**E** Three example neurons (left to right columns) showing different forms of multi-whisker integration. **C** Average firing rate during three stimulation conditions: stimulation at 45 Hz of W1 only (red), of W2 only (blue), or of W1 and W2 together (green). **D** Activity level in the stimulus space, measured as FR and represented by the size of the solid circle. **E** Bottom row, activity level in the stimulus space, fitted with the model in (**A**). The best threshold boundary (red line) maximizes discrimination of the target F1 > F2 over the stimulus space. Discrimination is measured as the area under the curve (AUC) using receiver operating characteristic (ROC) analysis. Note that multi-whisker suppression (example neuron 2) increases discriminability compared to no suppression (example neuron 1).

$p = 0.0039$, $n = 10$ mice; Fig. 3f, g; Wilcoxon sign-rank test). This result suggests that wS1 is causally involved in the perceptual decision-making task used here. However, the perturbation of activity homeostasis at a larger scale -including other brain areas-[22], might potentially be responsible for the impairment of behavior, independent of the manipulation of the sensory representation.

To test whether single whisker column manipulation selectively promotes one choice versus the other, we used the selective inhibition of barrels corresponding to whisker W1 or whisker W2 (Fig. 3h, i) intermingled with the control condition. The single-column inhibition shifted the psychometric curve depending on the silenced barrel (psychometric bias, $p = 0.002$, Friedman test, $n = 8$ mice; Fig. 3i). The trial-by-trial analysis shows a significant bidirectional behavioral effect with a bias of W1 blocking versus control ($p < 0.001$) and W2 blocking versus control ($p < 0.001$), tested using a Generalized Linear Mixed-Effects (GLME) model on $n = 15,412$ trials (Supplementary Table 2). This horizontal shift in the psychometric curve gives us indications of the relationship between S1 activity and subjective report. At the used light intensities, optogenetic inhibition of column W1 induces on average a rightward shift of the curve of $13.4\,Hz \pm 6.5$ (mean ± s.e.m.), and inhibition of column W2 a leftward shift of $26.4\,Hz \pm 5.9$ (mean ± s.e.m.). To control whether the optogenetic intervention alters the motor response instead of the sensory perception, we compared proportion of choice 1 in trials with optogenetic but without whisker stimulation and found it to be similar, although slightly biased towards ipsilateral choices (fraction of choice 1, W1 barrel blocking: $0.559 \pm 0.043$; W2 barrel blocking: $0.522 \pm 0.034$, mean ± s.e.m. $p = 0.21$, Wilcoxon signed-rank test, $n = 7$ mice). Thus, despite a seemingly modest amplitude, the bidirectional bias provides a strong level of evidence for the causal influence of single-column activity on the perceived whisker stimulation intensity.

## Multi-whisker suppression improves and generalizes neighboring frequency comparison across the stimulus space

To further dissect single neuron tuning to the whisker stimulation parameters, we applied a multiple linear regression (Fig. 4a) (ref. 6):

$$Rn = \beta_0 + \beta_{F1}\sqrt{F1} + \beta_{F2}\sqrt{F2} + \beta_{F1xF2}\left(\sqrt{F1x}\sqrt{F2}\right) \quad (1)$$

Where three parameters of whisker stimulation best explain the activity of neurons. First and second are the square roots of single whisker stimulation frequencies (F1 and F2; Fig. 4a). Third, the supra-linear interaction of the two whiskers frequencies F1×F2 increases the model fit (i.e., lower Akaike criterion and higher explained variance; Fig. S5). Other regression parameters were tested, such as divisive interaction terms or full categorical regressor (i.e., modal tuning to single frequencies). These could only marginally improve the overall model fit of single neuron activity at the cost of more complex designs, and were thus excluded. Overall, only a fraction of single neurons' activity variance could be explained by the stimulation parameters ($r^2 = 8.1 \pm 0.2\%$ mean ± s.e.m.; $n = 3706$ neurons; Fig. 4a), in agreement with a previous report[9]. We observe across the S1 population a positive correlation between $\beta_{F1}$ and $\beta_{F2}$ weights (Fig. 4b, $r = 0.55$; Pearson correlation between β1 and β2), meaning that neurons respond to the two whiskers and that representations of the whiskers are overlapping in the population. However, the more the neurons are driven by single whisker frequencies, the more they are being suppressed by the interaction of the two whiskers (Fig. 4b, $r = -0.65$, $r = -0.62$; Spearman's correlation between $\beta_{F1}$ and $\beta_{F1xF2}$, and between $\beta_{F2}$ and $\beta_{F1xF2}$, respectively, $p < 0.001$). At the level of the whole population, multi-whisker (MW) integration drives only a subtle sub-linear integration (Fig. S6c, d), likely due to the heterogeneity of MW integration across

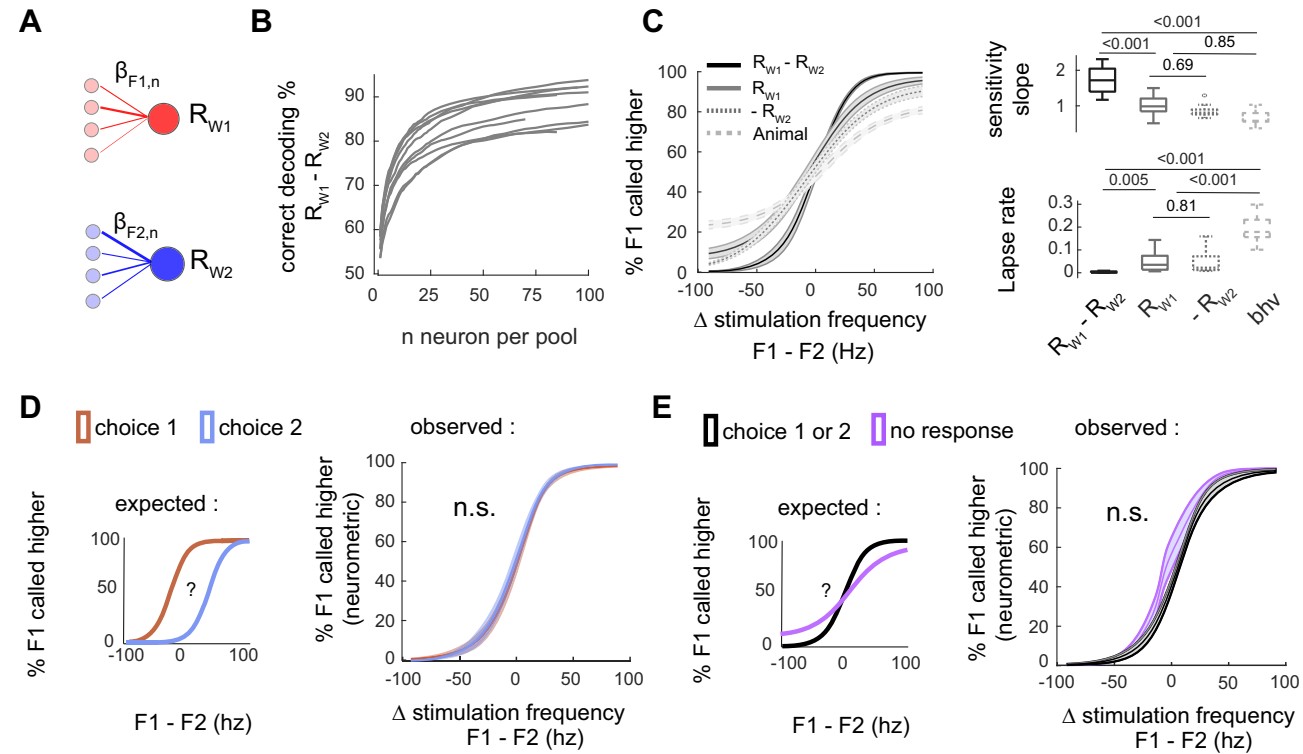

**Fig. 5 | Weighted population averages display reliable frequency categorization across behavioral outcomes. A** Sensory evidence pooled in two subpopulations weighted averages Rw1 and Rw2. Neurons were split in two pools depending on their selectivity for W1 or W2. β weights are obtained directly from the single neuron linear model in Eq. (1). **B** Decoding of the target side (i.e., F1 > F2) from the difference Rw1–Rw2 with an increasing number of neurons in each pool. $n = 11$ FOVs from seven mice. **C** Neurometric and psychometric functions compared. Left: psychometric/neurometric curves Data were represented as mean values ± CI95 across $n = 11$ FOVs from seven mice. We decoded negative Rw2 (-Rw2), so neurometric curves are in the same direction. Right: Comparison of the

fitted slope and lapse rate of the three psychometric/neurometric functions. **D** Neurometric function of Rw1 - Rw2 computed separately in trials with choice 1 or choice 2. not significant when testing sensitivity slope ($p = 0.93$), lapse rate (0.61) and bias ($p = 0.26$), $n = 11$ FOV (LME model test). Threshold for decoding were near zero (mean ± sem $5.5 \times 10^{-4} \pm 2.2 \times 10^{-3}$). **E** Neurometric function of Rw1 - Rw2 for engaged (black) and disengaged (gray) trials. $p > 0.05$ when testing sensitivity slope ($p = 0.09$), lapse rate ($p = 0.72$) and bias ($p = 0.20$), $n = 11$ FOVs from 7 mice (LME model test). Threshold for decoding were near zero (mean ± sem $8.1 \times 10^{-4} \pm 2.7 \times 10^{-3}$).

the population. Overall, this regression analysis confirms that a single neuron's response increases with stimulation frequency and highlights multi-whisker integration as a contributing factor shaping the activity of L2/3 neurons.

Does MW integration help solve the comparison task? To evaluate how this computation affects activity level in the stimulus space, three example neurons were compared: One with MW suppression, one without MW integration and one with supralinear MW responses (neurons 1, 2, and 3, respectively in Fig. 4c–e). The neuron 3 with supralinear MW response shows little difference in activity level on either side of the boundary task (quantified by area under the curve measurement; AUC = 0.65). Therefore, its activity level is little informative in regard to the task and may rather represent a source of noise in our paradigm. By contrast, MW suppression in neuron 2 leads the gradient of activity in the stimulus space to become perpendicular to the task boundary. Activity level scales better to the difference in stimulation frequency F1-F2 rather than to F1, the stimulation frequency of the neuron's preferred whisker (Fig. 4e). As a result, this neuron's output encodes MW stimuli in a manner that is relevant to the generalization of F1-F2 discrimination across the stimulus space (AUC = 0.97; Fig. 4e). Across the population, we find that both the AUC across the stimulus space and direct decoding performance increase with the degree of MW suppression (Fig. S6a, b). Besides, the comparison between two models, one with a MW interaction term and one without, indicates that populations of neurons exhibiting MW suppression derive benefits from the MW integration term in decoding the target (Fig. S7). Together, these findings reveal how MW suppression

improves and generalizes the discriminability of neighboring stimulation frequencies.

## Weighted population averages display reliable frequency categorization across behavioral outcomes

The subjects likely pool activity of multiple sensory neurons to inform its choice so we tracked representation of the two whiskers stimulation intensities at the population level. To do so, the population response Rw1 and Rw2 for the two competing sensory alternatives was modeled as a weighted population average of neurons preferring W1 or W2, respectively (Fig. 5a; see graphical method summary in Fig. S5).

$$R_{W1} = \sum (\beta_{F1,n} R_n) / \sum (|\beta_{F1,n}|) \text{ and } R_{W2} = \sum (\beta_{F2,n} R_n) / \sum (|\beta_{F2,n}|) \quad (2)$$

Where, the coefficients $\beta_{F1,n}$ and $\beta_{F2,n}$ from the multiple linear regression were used to weigh the activity of single neurons into population averages. By simply thresholding Rw1 and Rw2, we can decode the target side (i.e., is F1 higher or F2 higher?), estimating their neurometric performance in the frequency comparison task. This performance can be compared to that of the animals (Fig. 5b, c). When including an increasing number of randomly picked neurons to compute Rw1 and Rw2, we find that integrating information over 6.6 neurons on average (range from 3 to 14; $n = 11$ FOV) in each sensory pool was sufficient to match the animal's psychometric performance (same or higher decoded fraction correct; Fig. 5b). When including all sampled neurons, neurometric categorization of Rw1 and Rw2 shows

more reliable discrimination than the behavior across trials (lower "lapse rate" for Rw1 and Rw2 neurometric fits; $p < 0.001$, LME model post hoc test,) but have similar sensitivity (slopes compared to the psychometric function; $p > 0.1$). The integrated reading of these two signals by subtraction Rw1 - Rw2 is better at solving the task, improving sensitivity slope and lapse rate (LME model post hoc test, Fig. 5c). Overall, these results indicate that neurometric and psychometric functions covary, both as a function of the frequency difference ΔF. Besides, integrating over a small fraction of the population is sufficient to match behavioral performance in the task.

Sensory encoding was then compared across different behavioral outcomes. If the behavioral read-out depends on the difference of activity between Rw1 and Rw2, we expect that animals respond left as the sensory evidence Rw1 - Rw2 >0 and right as the sensory evidence Rw1 - Rw2 <0. Hence, we should observe a left/right shift in the neurometric functions drawn from trials with left/right choice (Fig. 5d, expected). We observed no difference in the bias of the neurometric functions of Rw1, Rw2, or Rw1 - Rw2, whether the animal responded left or right ($p > 0.05$, $n = 11$ FOVs from 7 animals; Fig. 5d and Fig. S8, LME model analysis). Excluding all trials with impulsive movements, (i.e., reaction time <1 s), to avoid measuring activity related to licks or uninstructed facial movements, we obtained the same results (all $p > 0.05$, $n = 9$ FOVs from 5 animals, LME model test; Supplementary Table 2). In addition, the slope and lapse rate of neurometric functions were unchanged whether the animal responded correctly or incorrectly (all $p > 0.1$, $n = 11$ FOVs from 7 animals, not shown).

The same analysis was repeated to compare trials with or without behavioral responses (Fig. 5e). One might expect a decline in sensory discrimination when the animal is not engaged in the task, visible as a decrease in the slope of the neurometric (Fig. 5e, expected). Again, the behavioral outcome has no impact on the sensory encoding for frequency categorization (lapse rate, slope, and bias, $p > 0.05$; decoding in engaged versus disengaged trials; Fig. 5e and Fig. S8). Excluding all trials with impulsive movements confirms the absence of significant change in bias, sensitivity or lapse rate ($p > 0.05$, LME model test). Conjointly, these results imply that the animal's perceptual errors are not due to failure in sensory encoding and reciprocally that spontaneous fluctuations in the frequency coding does not affect the choice of the animal.

## Sensory and choice coding are orthogonal

We then sought to describe whether and how choice is represented in wS1. Area under the receiver operating curve (AUROC, see methods) is a standard metric to test whether a neuron's activity is correlated to the choice of the subject (termed choice probability or CP[23,24]. AUROC measures how well the firing rate of a neuron discriminates between two conditions. We selected a matched number of left and right response trials, with small frequency differences (ΔF ≤ 30 Hz), to measure target side discriminability (i.e., F1 > F2 versus F1 < F2; Fig. 6a) and choice side discriminability (i.e., choice 1 vs choice 2; Fig. 6b). Restricting analysis to sensory ambiguous trials (ΔF ≤ 30 Hz) is expected to reveal the highest impact of sensory "noise" on behavioral choice[5]. Across these trials, 19.9% of neurons have a different level of activity depending on the choice side; 22.7% of neurons depend on the target side, and 5.1% have significant changes in activity depending on both the choice and target sides (AUROC different from 0.5; comparison versus bootstrapped distribution with a 95% confidence interval Fig. S9c). Importantly, we found a significant topographic distribution of AUROC choice (Fig. S9a, b; $p < 0.001$, $n = 3118$ neurons, LME model test) that was weak and reverted compared to the AUROC target.

If fluctuations in behavioral choice arise from the feed-forward readout of sensory coding fluctuations, it is expected that populations tuned to W1 and W2 have preferences for choice 1 and choice 2, respectively. How does selectivity for whiskers relate to selectivity for

choice? W1 and W2 populations have significantly different AUROC targets, averaging <0.5 and >0.5 respectively, which reflects their whisker tuning (mean AUROC = 0.47 and 0.54 for W1 and W2 populations, respectively, $p < 0.001$, LME model test; Fig. 6a). However, W1 and W2 populations do not have significantly different AUROC choice, averaging both slightly <0.5 (mean CP = 0.481 and 0.486, respectively; $p = 0.64$, $n = 9$ FOV neurons in total; LME model test, Fig. 6b). The average below 0.5 indicates a bias toward contralateral responses (choice 1). Changing the trial selection (e.g., including trial with ΔF > 30 Hz), or considering only highly whisker tuned neurons (Fig. 6b) leads to similar outcomes. These analyses indicate that sensory and choice coding can co-exist at the single neuron level, but that sensory selectivity does not predict choice selectivity.

We next questioned the representation of stimulus and choice in the entire population of layer 2/3 neurons. Choices 1 and 2 were included explicitly as regression variables to the multivariate linear model of single-cell activity (Fig. 6c; see methods). The model was fit with an equal number of left/right choices for each stimulation condition, which eliminates collinearity and enables independent calculation of neuronal selectivity for (1) choice and (2) whisker frequency. Only trials with motor response at the end of the stimulus were included. The distribution of model weights for choice selectivity ($\beta_{C1}$ - $\beta_{C2}$) versus weights for whisker frequency selectivity ($\beta_{F1}$ - $\beta_{F2}$) were uncorrelated ($r = -0.03$, $p > 0.1$, LME model test). Single-neuron activity was then pooled using the model weights for choice selectivity ($\beta_{C1}$ - $\beta_{C2}$), generating latent representations of left and right choices (Rc1 and Rc2, respectively, Fig. 6d), similar to the computation of Rw1 and Rw2. Decoding the animal's choice is increased when including both Rc1 and Rc2 as compared to each of these in isolation and yields a choice decoding around 68% (±0.018; mean ± s.e.m; across $n = 11$ FOV; Fig. S10). Sensory and choice information had opposite dynamics with sensory information peaking at stimulus onset and remaining high until stimulus offset (Fig. 6e). Choice information increases slightly above chance at stimulus onset and ramp to peak at the response time. Thus, decoding of the population reveals reliable coding of choice, with decision and sensory information having different dynamics over the trial time.

The pooling approach allows us to further describe the relationship between sensory and choice encoding with reduced dimensionality, on a trial-by-trial basis (Fig. 6f, g). To visualize this relationship, we selected two dimensions of relevance that separate best the stimulus target side and response side respectively (Rw1 - Rw2 and Rc1 - Rc2, Fig. 6f). A representational angle is calculated from the translation of left and right choices in the sensory and choice dimensions. In accordance with our previous observations, choice and sensory representations are encoded orthogonally, (90.2° and 90.0° on average for left and right target trials, $n = 2300$ and $n = 2152$ trials; Fig. 6f). This orthogonality is maintained across stimulus conditions and animals (Fig. 6g), and when considering Rw1 and Rw2 encoding subpopulations in isolation or when considering subpopulations analysis of neurons with high selectivity (Fig. S11a–d; but see Fig. S11e when selecting neurons coding for sensory choice intersection). Orthogonality enables the co-existence of reliable sensory and choice representations in two distinct dimensions of neuronal population activity.

## Sensory and engagement coding are mostly orthogonal

How does sensory encoding fluctuate upon different behavioral states? Under the 2AFC design, the absence of behavioral answers (i.e., no licks) cannot lead to a reward. As most no-lick trials occur in blocks at the end of the session, we hypothesize that these represent a distinct state of engagement in the task. We thus sought to characterize some physiological modulation associated with the states of engagement/disengagement. First, a pupillary constriction in engaged versus disengaged trial during the pre-stimulus epoch is observed ($p < 0.001$; spearman's correlation between no-lick probability and pupil diameter; Fig. 7a). Furthermore, engagement was accompanied by a

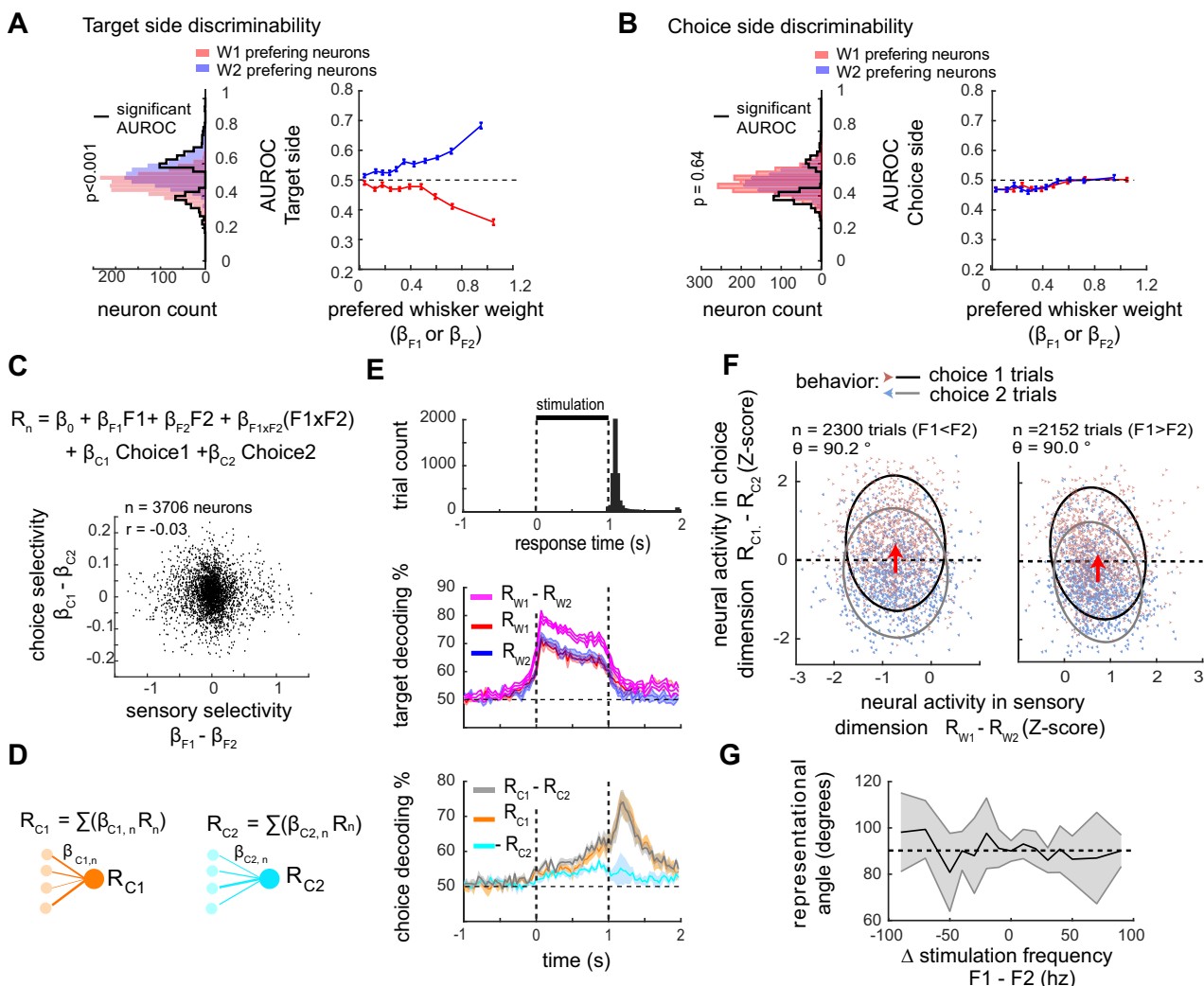

**Fig. 6 | Sensory fluctuations in wS1 are not correlated with behavioral choices.**
**A** Target side discriminability (i.e., F1 > F2 versus F2 < F1). left; AUROC distribution of all neurons ($n = 3118$) right: AUROC depends on selectivity. The population is split into 10 bins of increasing preferred whisker beta weight. Only trials with $|\Delta F| <= 30$ Hz and no impulsive movements are included in the analysis. Data were represented as mean values ± CI95. **B** Response side discriminability (i.e., choice 1 versus 2). Left: AUROC choice distribution of all neurons, depending on their preferred whisker. Right: AUROC choice doesn't depend on whisker selectivity. Population split as in (**A**). Data were represented as mean values ± CI95. **C** Top: model including choice 1 and choice 2 as regressor (respectively $\beta_{C1}$ and $\beta_{C2}$). Bottom: Model weights for sensory and choice selectivity are uncorrelated (Spearman $r = -0.03$; LME model test $p > 0.05$, $n = 3706$ neurons). **D** Choice coding evidence pooled into two subpopulations weighted averages Rc1 and Rc2. **E** Time

course of choice and sensory information. Top: Earliest response time of the animals (from video analysis, see Fig. S2). Middle: target side decoding (F1 > F2 versus F2 > F1) from Rw1, Rw2, or Rw1-Rw2. Bottom: animal's choice side decoding from Rc1, Rc2, or Rc1-Rc2. Matched number of Choice 1/Choice 2 trials in each stimulus condition. Error shades represent s.e.m. **F** Neural activity in the sensory and choice dimensions. Left: trials with F2 > F1 only. Right: trials with F1 > F2 only. The red arrow represents the transition from choice 1 to choice 2 trials (averages of neural activity across trials). θ is the angle between the x-axis and the transition arrow orientation. Fitted ellipses contain 80% of data points. Choices are best separated on the choice axis, and hardly on the sensory axis. **G** Breakdown of the representational angle for the different stimulation conditions. conditions with at least five FOVs and five trials per FOVs are included. Angles are represented as mean ± s.e.m. across FOVs $n = 5$ to 11 FOVs.

decrease in neural oscillations in the 2 to 10 Hz frequency band (Fig. 7b). These pre-stimulus markers strengthen the view of a different brain state during phases of disengagement[25,26], in which spontaneous activity of L2/3 is governed by synchronized, large amplitude oscillatory fluctuations of activity.

We then investigated changes in neuronal activity and sensory coding between the two states. To control for movement-related activity, we only included trials with a response occurring after the end of the stimulation window. A regressor for engagement was added to the linear model (1) of single neuron activity (Fig. 7c), and the latent variable for engagement (Reng) computed. Across neurons, model weights for engagement related modulation ($\beta_{Eng}$) versus weights for whisker frequency selectivity were not correlated ($r = -0.05$; $p > 0.05$,

LME model test; Fig. 7c). Decoding engagement from weighted average of sub-population preferring engaged versus non-engaged state revealed a significant representation of encoding throughout the duration of the trial (fraction decoded correct = 0.68 during baseline, 0.72 during stimulus presentation; average across FOVs, $n = 11$ FOVs; Fig. 7d). We next represented neuronal pooled activity in the engagement-related dimension as a function of the neural activity in the task-relevant sensory dimension (Fig. 7e). The shift in activity from disengaged to engaged trials was poorly represented in the sensory discriminative dimension with a representational angle close to 90°, (86.0° and 89.7° on average for left and right target trials, $n = 1958$ and $n = 1980$ trials; Fig. 7e). In accordance with our previous observations showing no change in neurometric performance (Figs. 7e and S6),

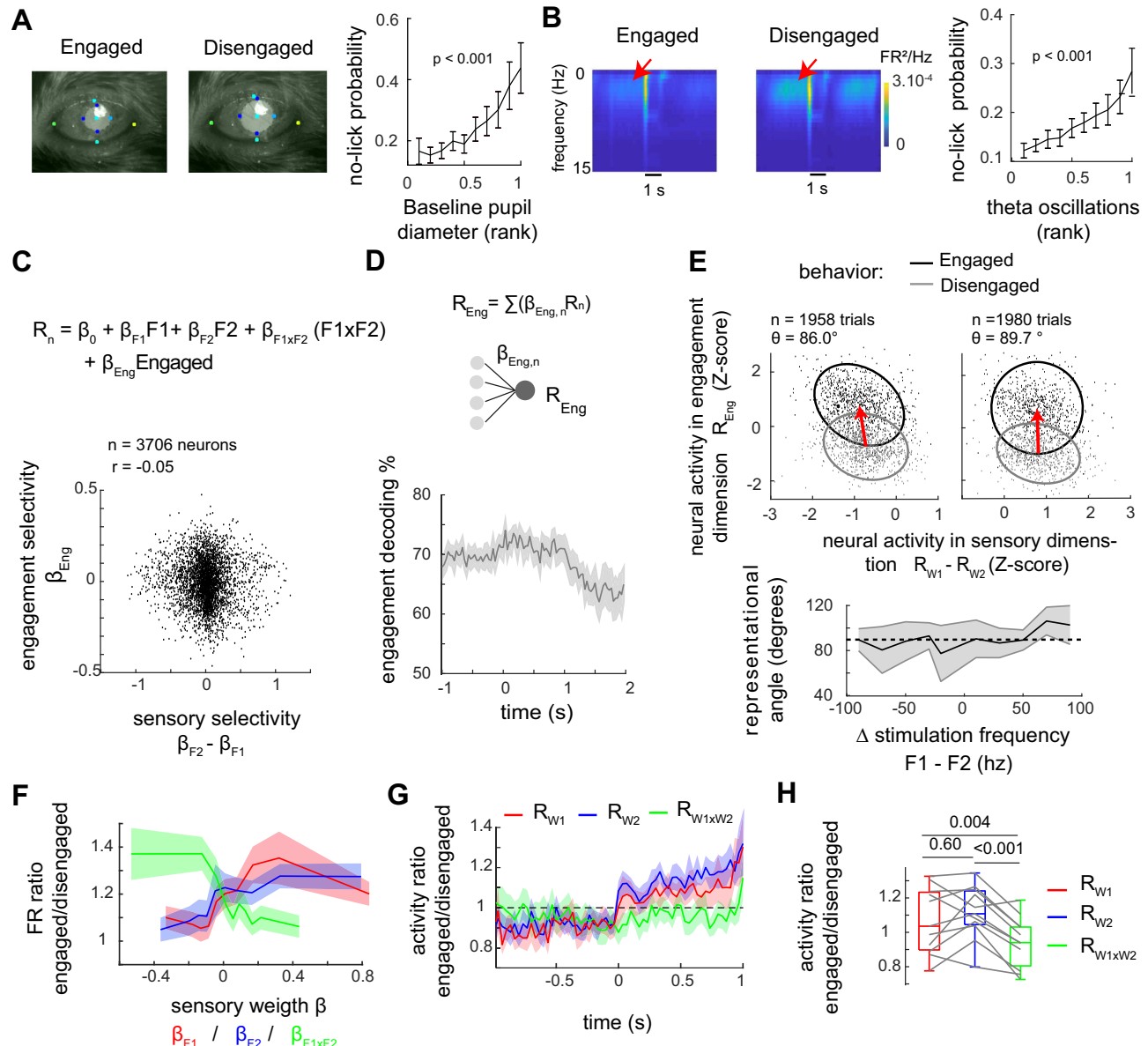

**Fig. 7 | An orthogonal representation of engagement coexists with a selective gain for single whisker representation. A** Engagement-related pupillary contraction prior to stimulus onset. Right: Miss probability in ten deciles with increasing rank of pupil diameter (normalized per session). Data were represented as mean values ± CI95; *n* = 10,000 trials from five mice (LME model test). **B** Engagement decreases slow oscillations prior to stimulus onset. Left, spectral power density from one example neuropil, averaged over all trials. Right, Miss probability in ten deciles with increasing theta power density (normalized per session). Data are represented as mean values ± CI95. *n* = 30,440 trials from seven mice; (LME model test). **C** Top: model including engagement regressor (with weight $\beta_{eng}$, see methods). Bottom: Model weights for sensory and engagement selectivity are uncorrelated (Spearman *r* = −0.05; LME model test *p* > 0.05, *n* = 3706 neurons from seven mice). **D** Decoding of engagement over time using. engaged/disengaged trials matched for stimulus condition. Data were represented as mean

values ± s.e.m. **E** Neural activity in the sensory and engagement dimensions. Left: trials with F2 > F1 only. Right: trials with F1 > F2 only. The red arrow represents the transition from engaged to disengaged (averages of neural activity across trials). θ is the angle between the sensory x-axis and the transition arrow orientation. Fitted ellipses contain 80% of data points. Bottom Angles in different stimulation conditions are represented as mean ± s.e.m. across FOVs. *n* = 5 to 11 FOVs. **F** Engagement ratio of activityas a function of neuronal sensory weights. Computed in engaged/disengaged trials with matched stimulus conditions. The neuronal population of *n* = 3706 neurons is split into 10 equal bins of beta weights ($\beta_{F1}$, $\beta_{F2}$, or $\beta_{F1xF2}$). Data were represented as mean values ± s.e.m. **G** Rw1, Rw2, and Rw1xw2 Engagement ratio over time. Rws are computed from independent pools of neurons. Data were represented as mean ± s.e.m across *n* = 11 FOVs. **H** Quantification of engagement ratio during the stimulus period in (**G**). Statistical comparison across *n* = 11 FOVs. LME model post hoc test.

engagement representation lies in a dimension mostly orthogonal to that of the task-relevant sensory representation (Rw1 - Rw2).

However, the analysis for single whisker representation provides a slightly different picture with representational angles differing from orthogonality (93.2 and 98.3 for Rw1 and Rw2, respectively, across trials; see details in Fig. S12). In fact, during stimulus presentation, we observe either positive or negative modulation of single neurons firing rate related to the state of engagement, with the average amplitude of

single neuron modulation significantly depending on the tuning to whisker frequencies (Fig. 7f). Weights for single whisker frequencies ($\beta_{F1}$ or $\beta_{F2}$) are positively correlated to engagement related gain, whereas the weights for supralinear whisker interaction ($\beta_{F1xF2}$) are negatively correlated with engagement modulation. Using a GLME model, we found that among the three sensory weights, $\beta_{F1xF2}$ alone captures engagement-related modulation related to sensory selectivity (Supplementary Table 2). The two pooled averages, Rw1 and Rw2,

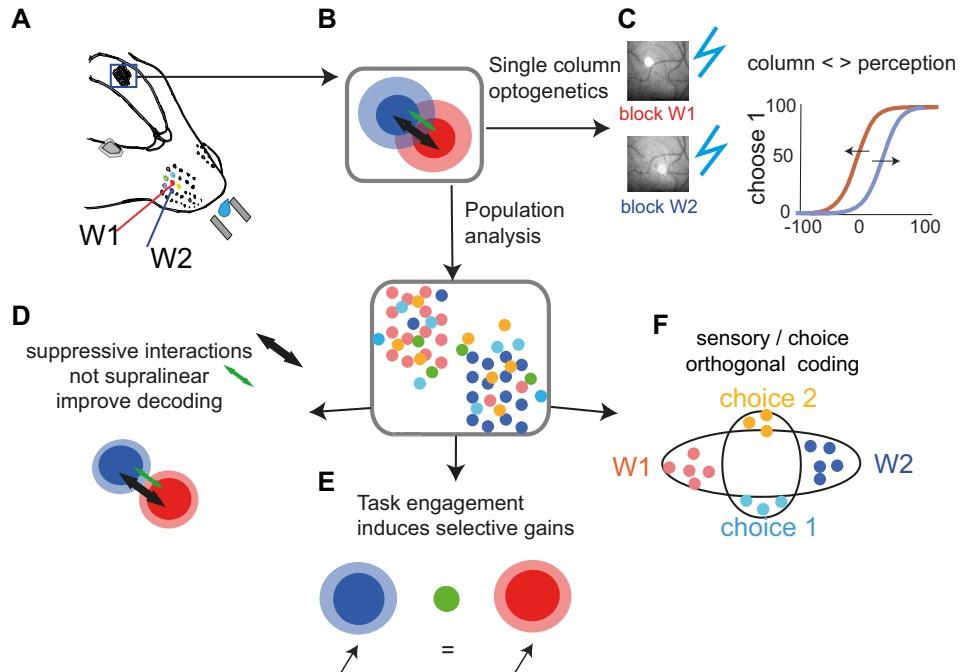

**Fig. 8 | Graphical summary of the main findings. A** We developed a 2AFC task in which mice had to compare the intensity of stimulation of two adjacent whiskers. **B** This paradigm enables the manipulation and simultaneous recordings of two rival sensory alternatives during perceptual decision-making. **C** The optogenetic silencing of individual cortical columns resulted in predicted shifts of psychometric functions, linking the perception of a stimulus feature to its early sensory representations. **D** Combination of two whiskers stimulation results mostly in suppressive interactions, with only a small fraction of neurons showing strong supralinear responses. Suppressive interactions improved the decoding of the target whisker stimulation. **E** When the animal engages in the task, we observe a sensory gain (increased responsiveness) which correlates positively with whisker selectivity and negatively with supralinear interactions. Such engagement-dependent gain may selectively enhance the relevant stimulus features (here, whisker identity). **F** At the level of a large neuronal population, it appears that the coding of choice is independent of the coding of the two rival sensory alternatives. Thus, choice-related activity is represented orthogonally to the sensory representation. Generally, sensory encoding remains reliable across all behavioral outcomes. Figure 8A is adapted from C S Barz, P M Garderes, D A Ganea, S Reischauer, D Feldmeyer, F Haiss, Functional and structural properties of highly responsive somatosensory neurons in the mouse barrel cortex, Cerebral Cortex, Volume 31, Issue 10, October 2021, Pages 4533–4553, https://doi.org/10.1093/cercor/bhab104 and released under a Creative Commons CC-BY-NC license: https://creativecommons.org/licenses/by-nc/4.0/.

show increased response amplitude in the engaged state, while the third pooled average of multi-whisker tuned neurons, Rw1xw2 shows the opposite effect (Fig. 7g, h). Engagement modulation of single whisker pools Rw1 and Rw2 is significantly different from Rw1xw2 during the stimulus period. When considering stimulus-evoked activity (i.e., normalized by baseline activity), engagement modulation amounts of +22 and +16% for Rw1 and Rw2 and −7% for Rw1xw2. Engagement thus promotes selective representations of single whiskers versus representation of the multi-whisker combination.

## Discussion

Understanding where and how the brain uses sensory inputs to inform perception and behavior is a fundamental question in neuroscience. Here, we developed a 2AFC task in which mice had to compare the intensity of stimulation of two whiskers. Few studies required mice to discriminate neighboring whiskers[27,28]. This is, to our knowledge, the first of these discrimination tasks using a variation of stimulation intensity, a critical parameter to study perception[15,29,30]. This paradigm takes advantage of the somatotopic map in wS1 to allow manipulations and simultaneous recordings of two rival sensory alternatives. The main findings are represented in a graphical summary (Fig. 8).

### A reliable and generalizable neuronal activity substrate in L2/3 neurons for discrimination of neighboring inputs

As described in previous work, we found that evoked activity in L2/3 displays a somatotopic organization with salt-and-pepper whisker tuning of single neurons[20,31], is sparsely distributed in the population[17–19] and adapts rapidly in most neurons[18,32]. Increasing

stimulation frequency of a single whisker increases monotonically population response[18,33,34]. Different codes could support the perceptual representation of the stimulus, including a rate code and temporal codes[35,36] (e.g., periodicity of firing). Previous studies have argued that neurometric coding performance by firing rate is closer to psychometric performance than that of temporal coding[33,37,38]. In our task, the level of activity in its home column is thus a candidate for supporting the perception of the whisker stimulation frequency. Our data show a common proportionality between the Δ frequencies, the Δ columnar activation, and the fraction of F1 called higher, suggesting that animals judge frequencies from the relative cortical activation in the two columns (Fig. 4).

We observed multi-whisker integration, found to be mostly suppressive (as previously reported[39–42] with the exception of some neurons responding supralinearly to multi-whisker stimuli[43,44]. Suppression is generally thought to be at least partially cortical dependent[39,45]. It is proposed to be beneficial in sharpening boundaries and information coding[46,47]. In the context of our task, we show that decoding linearly from neurons encoding one single whisker frequency is insufficient to compare frequencies across stimulus space. Rather, multi-whisker suppression provides a generalized activity signal for judging the highest of the two stimuli intensities across the range of stimuli tested (Fig. 4).

### Single-column activation drives perception of the sensory feature it encodes

The results of selective optogenetic manipulation experiments have several implications (Fig. 3). Firstly, earlier studies demonstrated the

persistence of tactile-driven behavior despite the total ablation of S1[48], sparking controversy over the role of S1 in driving learned, tactile-driven behavioral responses. Besides, most previous studies involving S1 used inactivation to show transient impairments in behavioral performance. However, these impairments could stem from various factors such as creating distracting perceptions for the animal or inducing long-distance perturbations of neuronal homeostasis[22,49] (off-target effects). By employing somatotopically targeted manipulation to induce a bidirectional shift in behavior, our results provide compelling evidence supporting the involvement of S1 in the task.

Secondly, a contemporary debate revolves around the size of the neuronal population ensemble necessary to elicit learned perceptual-driven behaviors. Recent manipulations, involving the activation of excitatory neurons or stimulus-selective sub-ensembles, have been shown to trigger learned behaviors[4,7,32,50–52]. However, these manipulations might rely on perceptions from alternative pathways not utilized by the animal under normal sensory circumstances. Our results provide direct evidence that neural activity in a single column drives the behavioral report of the local point or feature it encodes. The columnar scale is particularly significant, as functional columns are considered the fundamental building blocks of the mammalian brain, prevalent across nearly all brain areas[53]. While our study specifically addresses early sensory representation in the somatosensory system[54], its implications may extend to other perceptual representations in areas displaying cortical maps.

Thirdly, our experiment partially addresses the identity of signals used for the perceptual readout of S1 activity. A recent study[55] reported that selectively manipulating choice, but not sensory neurons, influence behavior during the session, raising the possibility that the animal relies on choice neurons only for its decision-making. In our task, choice-related activity exhibits weak and reversed somatotopy compared to sensory-related activity (Supplementary Fig. 11). Hence, the optogenetic manipulation is aligned with the sensory representations but not the choice representations, which should result in a selective shift in the sensory representation only. Consequently, our results argue against the readout of choice neurons only and suggest that the trial-by-trial readout predominantly relies on the sensory encoding dimension.

From a technical perspective, some limitations of our approaches could be overcome in the future. For instance, we observed inhibition outside of the stimulated area (Fig. S4), a challenge that could be mitigated by the use of soma-targeted opsins. In addition, optogenetic stimulation used here is likely to inhibit activity in deeper layers, which could be addressed by conditional genetic expression of the opsin in single layers. To address directly the identity of neurons used for readout however, it would be required to use holographic activation of single target neurons, in column size, functionally defined ensembles; or using manipulation of different functional codes (e.g., temporal codes)[32,56,57] which may provide further insights into perception-related circuits and dynamics[58,59].

### Representation of choice in rodent wS1

The relatively subtle encoding of choice observed here in wS1 is congruent with previous studies of vibrotactile discrimination in non-human primates[60]. However, it may seem contradictory with more recent results in the murine model, showing a large difference in evoked activity between hits and misses before the onset of licking in a go/no-go paradigm, and a correlation between sensory and choice activity[8,61]. It is still unclear whether these discrepancies are due to the different animal models, the nature of the perceptual tasks, or the different behavioral paradigms. During the detection of near-threshold stimuli, activity in the primary sensory cortices conveys choice signals, in both primates[62,63], and rodents[8]. In a situation when animals are asked to perform discrimination of multiple visual stimuli, the choice might actually be poorly predicted from the primary visual cortex

activity[64]. In our paradigm, trials with response only slightly increase the total L2/3 activity, arguing against a large contribution of motivation and motor-related activity on total activity change. Therefore, we hypothesize that the presence of strong choice signals in primary sensory areas depends on the sensory stimulation parameters, with near-detection threshold stimuli possibly leading to the highest choice predicting activity. A distinction between detection and discrimination has been made in non-human primate studies[65]. These studies could help design studies in rodent research to disambiguate the representation of choice from motor-related activity and motivation of reward expectations. Future studies may use near-threshold stimulus detection with a 2AFC and task reversal to address this question.

In the classical view of perceptually guided decision, sensory evidence builds gradually in decisional and motor areas[1,60,66] from pooling multiple sensory neurons that are weakly correlated with choice. The pooled, correlated variability in sensory coding would, in turn, explain part of the variability in behavior. Here, we show that the animal's psychometric sensitivity is matched by a random pool of 6 to 7 neurons only and outperformed by larger S1 populations (Fig. 5), similarly to what was found previously in the somatosensory system[67,68]. Hence, the sensory information content should not limit the animal's choice. More importantly, the sensory information remains reliable across perceptual successes and errors (Fig. 5). These results suggest that perceptual variability does not originate from variability in sensory coding in wS1 but rather from state or choice fluctuations in downstream areas.

At the population level, fluctuations on the sensory axis do not correlate with S1 choice representation (or the behavioral choice; Fig. 6). Instead, our results add to the growing evidence that choice coding lies in dimensions orthogonal to that of sensory representation, enabling multiplexing of information in different neuronal subspaces[55,69,70]. This pattern is allowed by the uncorrelated coding of sensory and choice variables at the single neuron level, which suggests different origins for those two signals. In addition, sensory and choice information display opposite dynamics. Choice-related information peaks at the onset of movement, while sensory information peaks at stimulus onset and persists until stimulus offset. Previous studies have indicated that subjects use mainly the earliest temporal component of sensory activity to form their choice[7,71], while the manipulation of later components correlating with decision signals have less causal influence on them[7] and originates most likely from top-down modulation[70,72,73]. Our results agree with a different origin of sensory versus choice signals and further show that these later decision signals are unrelated to sensory encoding at the population level.

Choice-related activity does not appear to causally drive behavior but may fulfill a different function. In a recent experiment, the manipulation of choice neurons' ensembles in S1 biased bidirectionally performance throughout the session but not between stimulated versus non-stimulated trials[55]. This finding is compatible with a hypothetical function of choice neurons in reinforcing sensory-motor association and guiding subsequent decisions. In this regard, orthogonal coding provides the advantage of not corrupting sensory information, which could allow for flexible use of sensory information, should the task or behavioral requirement change.

### Sensory information and representation across states

In our task, the engaged state is associated with a pupil contraction during the pre-stimulus period, in line with recent work using a whisker mediated 2AFC[16], although at odds with earlier findings[74,75]. We also observed a suppression of slow oscillatory activity[26]. These characteristics resemble in part to the active mode that displays, when compared to a quiescent awake mode, a depolarization of excitatory L2/3 cells[76,77], and a suppression of the large subthreshold oscillations. These effects were shown to be partly under thalamic[77], as well as cholinergic influence[78,79]. Besides, it was recently shown that brain-

wide fluctuations of activity were explained by face motion[9,80] but also pupil dilation in the awake animal[80]. In the later study, motion-related activity was found to be encoded orthogonally to that of the sensory coding, but such a relationship was not clear for state-only related activity. Here, using cross-validated decoding and excluding face motion from video tracking, we find that task engagement is reliably represented by L2/3 neurons. Importantly, engagement-related activity covary little with variability in sensory encoding from trial to trial and is present prior to the stimulus presentation (Fig. 7). We thus conclude that correlates of the engagement state are present but primarily in a dimension that lies orthogonally to that of the two whiskers selectivity.

However, we discovered an intriguing and subtle pattern of engagement-related modulation: neurons suppressed by multi-whisker interaction exhibit an increase in stimulus responsiveness with engagement, while neurons enhanced by multi-whisker interaction do not (Fig. 7f). Consequently, the representation of single versus multi-whisker inputs is heightened in the engaged state, potentially favoring a more detailed spatial map of sensory inputs. This gain was independent of the stimulus condition, thus not enhancing the discriminability of the stimulus (i.e., akin to an additive gain). However, a downstream reader could potentially benefit from this selective up-modulation. It has been proposed that attention can selectively increase the representation of stimulus features relevant to the task[81]. Interestingly, a recent study showed that decoding neurons integrating two compared visual stimuli correlates with choice[82]. While we did not find direct evidence in our data that multi-whisker suppressed neurons are preferentially used for read-out, the increased responsiveness of these neurons in the engaged state argues for their importance for downstream processing.

In summary, the task and optical techniques presented here allow for a flexible combination of manipulative and correlative approaches, necessary for understanding the substrate of perception and decision making[12,83]. The analysis of population representation from trial to trial highlighted the multiplexing of behavioral and sensory information, which allows reliable sensory encoding and provide a potential frame for reinforcement learning or flexible routing of sensory information. These results provide insight into the role of primary sensory areas during complex perceptual decisions, namely the formation of reliable sensory signals used for downstream decision processes.

## Methods

### Animals
The experiments described in the present work were conducted in accordance with the guidelines on the ethical use of animals from the European Community Council Directive of 24 November 1986 (86/609/EEC) and in accordance with institutional animal welfare guidelines and were approved by Animalerie Centrale, Médecine du Travail and the Ethics Committee CETEA of Institut Pasteur, protocol numbers.

All mice were aged 8 to 16 weeks at the time of surgery. 7 C57BL/6J male mice were obtained from Charles River. Four GAD-67 and nine VGAT-cre male mice were bred in-house at Animalerie Centrale, Institut Pasteur. All animals were housed in polycarbonate individually ventilated cages equipped with running wheels and were kept under constant temperature and humidity with a 12 h light-dark cycle and food available *ad libitum*. At the end of the experiment, mice were aged at a maximum of 32 weeks.

### Surgery
Mice were anesthetized with isoflurane (Piramal Critical Care, UK; induction: 4%, maintenance: 2%) and placed in a stereotactic frame (Kopf Instruments, USA). Mice were injected with buprenorphine (Vetergesic CEVA, France; 0.02 mg/kg, subcutaneous.) for pain management, and their eyes were protected from desiccation by applying ointment. The fur was removed over the skull, the skin was disinfected with betadine (Mylan, USA), and xylocaine (Bayer, Germany; 0.25%, 0.05 ml) was injected subcutaneously for local analgesia. The skull was exposed and several whisker barrels (at least three barrels among one of these: rows A–E within arcs 1–2, and/or alpha beta gamma) in the right hemisphere were identified using intrinsic optical imaging together with whisker stimulation by means of piezo control[18]. A 3 mm diameter round craniotomy was then performed, and a virus was injected at 350 μm depth from the pia, with multiple injection spots (from 200 to 400 nl injected in each) patterned in a grid injection spaced by 500 μm, such that viral expression would span the entire window. The virus carried either a red calcium indicator alone (7 mice; jRGECO1a; AAV1.Syn.NES.jRGECO1a.WPRE.SV40; a gift from Douglas Kim & GENIE Project (Addgene plasmid # 100854; http://n2t.net/addgene:100854; RRID:Addgene_100854) (ref. 84), or in combination with EGFP (2 mice pAAV.synP.DIO.EGFP.WPRE.hGH, 1/20 dilution, viruses mixed prior to the injection). For the optogenetic, the virus carried either Channelrhodopsin 2 alone (five mice, pAAV-EF1a-doubleloxed-hChR2(H134R)-EYFP-WPRE-HGHpA; a gift from Karl Deisseroth (Addgene viral prep # 20298-AAV9; http://n2t.net/addgene:2029) or with the green calcium indicator GCaMP6s(4 mice); pAAV.Syn.GCaMP6s.WPRE.SV40 was a gift from Douglas Kim & GENIE Project (Addgene plasmid # 100843; http://n2t.net/addgene:100843; RRID:Addgene_100843) 1/10 dilution[85]. Injections were performed using beveled glass pipettes (Drummond, USA) with an inner diameter of 15–30 μm. The craniotomy was then sealed with glass coverslip (3 mm diameter round window; UQG Optics Ltd, UK). Dental cement (DE Healthcare Products, UK) was applied to keep the window in place and to form a head cap holding a custom-made head-post made of titanium. Throughout surgery, body temperature was maintained at 37 °C with a feedback-controlled heating pad (Thorlabs, USA). After surgery, mice received carprofen in a gel (Dietgel, clear H2O, USA) for pain management (0.02 mg/kg; every 24 h; 3 d postoperatively). Mice were single-housed for the rest of the experimental procedures to avoid potential damage to the implant.

### Intrinsic optical imaging
Prior to two-photon calcium imaging, intrinsic optical imaging (IOI) was performed to identify the single whisker representation in the neocortex using a 50 mm tandem lens system. Imaging was conducted at 30 Hz with a 12-bit CCD camera such that the field of view (FOV) spanned the entire cortical region covered by the window. The imaged region was continuously illuminated with a red-light emitting diode (630 nm wavelength). For stimulation, a single whisker was inserted into a glass capillary mounted on a piezoelectric element powered by a piezo controller (MTD693B, Thorlabs, USA). The whisker was moved rostro-caudally for 6 s with a 10 Hz square wave pulse (amplitude: -0.5 mm). This stimulation protocol was repeated at least ten times with inter-trial intervals (ITI) of 20 s. The change in absorbed light was computed as:

$$\Delta r = r_{stim} - r_0 \tag{3}$$

with $r_0$ being the average image in the 5 s window prior to stimulation and $r_{stim}$ the average image in the 5 s window prior to the end of stimulation. $\Delta r$ revealed discrete areas of functional activity corresponding to the single whisker representation (see example in Fig. 2A, areas are delineated as a change in absorbed light ≥0.15%, following an isotropic Gaussian smoothing with σ = 38 um). A green-light emitting diode (530 nm) was used to visualize the vasculature pattern to match IOI images to subsequent two-photon imaging and optogenetic experiments. Cortical column centers were defined manually from two-photon neuropil response when available (in Figs. 2, 3), and from IOI otherwise.

## Behavioral task

Water deprivation of the animals started a minimum of 2 weeks following surgery to allow for recovery. Animals were handled twice a day with 1 ml of water delivered manually by the experimenter through a syringe and progressively habituated to head fixation and to the experimental setup. This procedure has been described in detail in a previous article[86]. Once the animals were habituated to the setup, all whiskers but two were trimmed below 3 mm, on the opposite side of the craniotomy. The pair of target whiskers were neighboring on the same arc: either B1/C1 or C1/D1 were picked. At the beginning of each session, the two target whiskers were inserted in two capillary glasses mounted on two independent piezo elements set at 3 mm from the whisker pad. Animals were first exposed to stimulation and got free water delivered 500 ms after the onset of the stimulus. Water was delivered according to the contingency decided as follows higher whisker on the arc (either B1 or C1) was associated with left waterspout and lower whisker on the arc (either C1 or D1) with the right water. The percentage of water delivered automatically was progressively lowered (-10% per day), and thus water was only rewarded when the animal performed a lick on the correct side. To avoid frustration during this learning phase, droplets of water were sometimes added manually on the correct side via a direct command on the custom software, after the animal produced licks to the incorrect side. Progressively, animals learned the contingency, and performed better over the course of days/weeks with a variable learning rate (Fig. 1). Once this association was mastered by the animal (>70% correct responses in at least two consecutive days), simultaneous stimulation of the two whiskers was progressively introduced at a low rate. Once the performance was maintained at a high level on a simultaneous whisker stimulation task ($p < 0.05$ assessed by a chi² compared to chance level, test across discrimination conditions), we started the imaging daily; Image collections were performed throughout a period of less than 1 month.

## Psychometric/neurometric analysis

Evaluation of the performance of the animal/classification of neuronal signals was done by quantification of parameters of a psychometric/neurometric function. A logistic function of the following form was fitted[14]:

$$\psi(x,\alpha,\beta,\gamma,\lambda) = \gamma + (1 - \gamma - \lambda)F(x;\alpha,\beta) \qquad (4)$$

$$F(x;\alpha,\beta) = \frac{1}{1 + \exp[\alpha - x\beta]} \qquad (5)$$

$\gamma$ and $\lambda$ correspond respectively to the lower and higher asymptote of the fit. $\alpha$ the steepness of the curve, and $\beta$ the value of the midpoint, or bias (where fractions of left and right licks are equal). All parameters of the fit were left free to vary. Comparison of psychometrics function were performed as the comparison of best-fit parameters between groups of mice or FOV (e.g., comparison of $\alpha$, the best fit in engaged versus disengaged states with a Wilcoxon signed-rank test or LME model test, respectively).

## Behavioral video data analysis

Images of the animals' faces were acquired by video tracking at two possible rates: 60 Hz (mice 1–3) or 300 Hz (mice 3–7). Images were processed with the DeepLabCut toolbox[87] at the frame rate of acquisition. For the video tracking from below the snout, we first tagged 400 frames picked randomly across all datasets with the following tag: nose left, nose right, nose center, philtrum, teeth center, chin, tongue, front right whisker base, front right whisker tip, back right whisker base, back right whisker tip, left paw, and right paw. The neural network was trained to detect the markers on a dataset of 500 frames randomly selected among all videos. Through visual inspection, another 200

frames with unsatisfying tagging were manually labeled and added to the training dataset; the network was then re-trained with all labeled frames. All videos were processed with the latter trained network. The marker positions measured in pixels and the likelihood of detection provided by DLC software were used to compute the movement of the different body parts. In each session, we computed the standard deviation of the x-y position of the nose and whisker markers (in pixels). These measures were ranked and averaged for each trial. We then identified the 5% trials with the most and 5% trials with least face movements, $5_{high}$ (so trials with high movements), and $5_{low}$ (trials with the least movement), respectively. The idea behind this strategy stems from the observation that in every session, there are trials with high levels of face motion and trials in which the animal remains still. These trials are, in our view, the most reliable way to normalize datasets across sessions and imaging parameters. From this strategy, it is possible to compute relative values of face, snout and whisker movement within the session, as used throughout our study, although these are not given as absolute measures in degree angle or millimeters.

To estimate tongue movements, we used the baseline epoch from the $5_{low}$ trials to define a threshold, computed as average plus three standard deviations of the likelihood of appearance. For instance, the tongue was considered as outside of the mouth when its likelihood at a given time point was superior to the above-described threshold. When detected, tongue movements were classified as left or right depending on the marker relative x position compared to teeth (fixed, computed as the average of all frames within the session). The whisker angle was first computed as the average of all four whisker markers in the y dimension. Nose position was first computed as the average between the three nose markers in the X dimension. Both nose and whisker measures were Z-scored session per session, as follows:

$$Z(t) = \frac{x(t) - \hat{W}}{\sigma W 5 high} \qquad (6)$$

With $\hat{W}$ being the average position during baseline across trials, and $\sigma W_{5high}$ being the standard deviation of position across the $5_{high}$ trials. From several possible normalization procedures tested, this normalization provides the most comparable distribution histograms of relative nose and whisker positions across sessions. Finally, to compute the earliest reaction time, we used the Z-scored nose position. If the nose position changed by more than 0.1 Z-score between two frames, the first of the two frames was counted as the reaction time.

The same labeling procedure was used on videos acquired from the side of the face (all recorded at 30 Hz from four animals), with the following set of markers: eyelid top, eyelid bottom, pupil left pupil top, pupil right, pupil bottom, pupil center. The pupil diameter was computed as the Euclidean distance between markers 'pupil left' and 'pupil right' markers. When the likelihood of detection of any of these two markers was below 0.5, the trials were thrown away. From a measure in pixels, pupil diameter was normalized between the 5th and 95th percentile for each session separately as follows: P(t) = (P(t) - P5th)/(P95th - P5th). This measure yields a relative change in pupil size as compared to the dynamic range it can achieve, with 90% of the values between 0 and 1. The use of low/high percentile was preferred to min/max because of possible outliers.

## Two-photon calcium imaging data acquisition

Two-photon imaging was carried out after a minimum of 2 weeks following surgery to allow for sufficient viral expression. Functional images were acquired at -29.7 Hz using bidirectional scanning and an image resolution of 512 × 512 pixels spanning 738 × 605 microns. The field of view was centered using an optical imaging signal (Fig. 2a) in order to cover two barrels of whiskers within the same FOV. We recorded in depth ranging from 110 to 200 μm (see Supplementary Table 1 for detailed parameters of recording in each FOV, and Fig. S13

for stability of expression over time). The genetically encoded calcium indicator was excited with a Ti:sapphire laser (jRGECO1a at 1040 nm, GCaMP6s at 920 nm; pulse frequency of 80 MHz, Chameleon; Coherent) using an equivalent power of ~100 mW for jRGECO1a and a power ranging from 25–120 mW depending on the depth of recording for GCaMP6s. Emitted light was recorded through a 16x, 0.8 NA objective (Nikon, Japan)and detected with two photomultiplier modules having the following filter settings: bandpass filter 540/40 and 617/73 for green and red, respectively and short pass filter BrightLine 750/sp for each channel (Semrock, USA). Two-photon laser scanning was controlled using the ScanImage software 2018[88].

## Simultaneous two-photon imaging and optogenetic stimulation

Simultaneous optogenetic stimulation and calcium data acquisition were performed in the same setup as standard two-photon calcium imaging with the following differences. Images were acquired continuously in 5 planes, at a total volume imaging rate of 4.58 Hz. Optogenetic stimuli consisted of a train of 1 ms pulses delivered at 40 Hz from −0.25 s before the onset of the stimulus to 1 s after the offset of the stimulus. Blue light pulse trains were generated with an LED (470 nm, M470L4, Thorlabs, driver LEDD1B, Thorlabs) taking continuous voltage from a breakout box (National Instruments, SCB-68A) as driving input. From the light source, we mounted serially a diffuser, a lens, and an iris was positioned such that its image is formed in the first imaging plane. The iris' image diameter was set to either 0.10 or 0.45 mm for the selective and broad inhibition, respectively. This setup only allowed us to perform the three optogenetic conditions described in Fig. 3 in the same two-photon areas, but in different sessions. To do so, the imaging planes were first matched, and the image of the iris was then positioned using a XY translation mount. Trials with different light intensity (0 to 44.3 mW mm² for selective and 0 to 17.9 mW mm² for broad light patterns) were randomly alternated with trials without light. We used a Polychroic mirror (Chroma, zt470/561/nirtrans) transmitting the infrared and blue light, while reflecting the green to the gated PMTs (Hamamatsu, H11706). The triggerable PMTs shutter was synchronized with each pulse of blue light and engaged for 1.33 ms. For each pulse of blue light, a fraction of the frame was missing and thus interpolated from previous and following frames, on a line-by-line basis (MATLAB interp2 function, linear interpolation). Because we observed a decay of neuropil fluorescence stemming from the blue light pulse, the interpolation was carried over a total duration of 10 ms per light pulse (i.e., roughly one-third of the imaging frame). Two-photon imaging data then followed the same processing pipeline as standard two-photon imaging. To estimate the inhibition level, we included cells showing (1) significant response to whisker (paired $t$-test for each neurons for different level of fluorescence in baseline versus whisker stimulation without optogenetic light) and (2) no activation by optogenetic light (paired $t$-test for different level of fluorescence with versus without light stimulation $p < 0.01$; to exclude inhibitory neurons directly activated by optogenetic light) and (3) lying within 200 μm of the axis passing the two barrel centers. Only neuron meeting these criteria are analyzed and displayed in Fig. 3D.

## Optogenetic inhibition during behavior

A projector (DLP4500, Texas Instruments, USA) was synchronized with the behavioral software. Light intensities were matched to calibrations made in the simultaneous two-photon and optogenetic. 40 Hz trains of 1 ms pulses were used. The surface of the projected disk was set to 0.105 mm for the selective inhibition (using from 8.3 to 35.9 mW mm², measured under constant light) and 0.45 or 0.64 mm for the broad inhibition (5.0 to 21.8 mW mm²). In the three animals for which individual calibration was available we used an individualized light power ranging from 8.3 to 19.4 mW mm². In other animals which only expressed channelrhodopsin, we chose a light power of 19.4 mW mm². We did not observe changes as a function of light power applied (not shown), thus behavior in trials across light power were pooled together. For each individual animal, a set of images corresponding to the three stimulation conditions was generated upstream from behavior, based on the intrinsic optical imaging data. Every behavioral session started with the positioning of the system using a manual translation mount. The image of the projector was focused ~150 μm below the brain surface. Trials with and without optogenetic were randomly intermingled with an average proportion of roughly one out of four trials with light stimulation on a brain location, and three out of four on the dental cement, i.e., sham condition.

## Two-photon calcium imaging data preprocessing

In 11 FOV from seven animals with jRGECO1a, we carried out a detailed analysis of complete population statistics. Raw images from calcium imaging were first motion-corrected in X and Y dimensions using the Suite 2p motion correction module. ROIs were then delineated using the routine from suite2p[89] and manually curated, and every visible cell with an event rate >0.005 Hz was included. A local neuropil fluorescence was defined as an annulus surrounding the ROI, with an external diameter of three times the ROI diameter, and an internal diameter of 1.5 times the ROI diameter (padding). A noise-adjusted Df/F0 was then computed from the mean ROI fluorescence and neuropil fluorescence using a custom algorithm. In brief, this algorithm runs four processing stages iteratively until finding a stable estimate of the neuropil correction factor α: (1) correction of neuropil and slow trends, (2) percentile-based Z-scoring, (3) estimation of putative periods of non-activity (Ena) from the Z-score, (4) linear regression to estimate the neuropil correction factor α. α is initialized to 0.5 (other values return the same results but converge more slowly).

1. Neuropil is subtracted with factor α: $F1(t) = F(t) - \alpha * Fp(t)$. We then subtract a 10th percentile in a running window of 30 s. $F2(t) = F1(t) - F_{10th}(t)$.

2. A percentile-based Z-scoring is performed that scales to the noise level. Based on the assumption that activity-related calcium transients are upward, we predict that lower percentiles of the fluorescence distribution are not contaminated by activity and can be used to infer underlying noise statistics. Following an assumed Gaussian distribution of the residual noise, we define the variance sigma $sig = pr(16) - pr(2.3)$, and the average baseline fluorescence $mu_f = pr(2.3) + 2 \times sig$; with $pr(2.3)$ and $pr(16)$ being the 2.3rd and 16th percentile of the fluorescence distribution. $Z = (F2(t) - mu_f)/sig$

3. Z is smoothed with a boxcar filter of width 3, and an index of autocorrelation of order 1 is calculated (AR1). Increase in AR1 track with high sensitivity consistent increases of fluorescence (i.e., increase in fluorescence consistent in more than two to three frames). $AR1(t) = Z(t)*Z(t+1)$. Values of AR1 above an arbitrary threshold of 0.5 are considered periods of possible activity, and epochs of at least five frames with AR1 below 0.5 are considered possible epochs of non-activity ($E_{NA}$).

4. factor α was calculated on by least squares regression of $F(t_{ENA}) = \alpha * Fp(t_{ENA})$, multiple time in 30 s long time windows (to avoid regressing slow drifts) and only including timepoints in $E_{NA}$. Updated α is finally computed as the average across all 30 s windows with at least five data points.

Steps (1) to (4) were repeated until, α varies by less than 0.025, or iterated at least ten steps.

More than 95% of neuropil correction factors computed this way were between 0.5 and 0.85; values below or above this range were respectively set to these boundaries. We then applied steps (1) and (2) to compute the final percentile-based Z-score, and a noise-adjusted Df/F0. Df/F0 was computed as F2(t)/F0 with F0 being the highest value between (a) the median of $F_{10th}$ or (b) the median of neuropil

fluorescence Fp(t) was taken. Finally, a constrained deconvolution was applied[89] to return a continuous spiking estimate. This preprocessing strategy and criteria were optimized from the freely available jRGE-CO1a dataset from the CRCNS website[84] to match the state-of-the-art algorithmic performance in event detection[90].

### Linear regression on single neuron firing rate

We used a linear model to regress the recorded spiking activity of single neurons against the stimulation frequency parameters applied to the two whiskers (F1, F2, F1×F2). The same design model was fitted separately for each neuron. The fitted data points represent spiking activity averaged over 1-s intervals during baseline epochs (1 s before the stimulus onset) and whisker stimulation epochs (from 0 to 1 s after the stimulus onset; the whole stimulus duration) in individual trials. These averaged activities were then concatenated into a single vector, denoted as "$R_n$" for the neuron n. Rn was then regressed against the square root of the two whisker frequencies (F1 and F2) and the product of these two regressors (F1×2) for each trial. F1, F2, and F1×2 were set to zeros for baseline epochs:

$$R_n = \beta_0 + \beta_{F1}\sqrt{F1} + \beta_{F2}\sqrt{F2} + \beta_{F1xF2}\sqrt{F1x}\sqrt{F2} + \varepsilon \tag{7}$$

$\beta_0$ represents baseline activity (the intercept), other βs represent weights for the regression variables, and $\varepsilon$ represents residuals. In the versions of the model that explicitly included choice (Fig. 6) or engagement (Fig. 7), we used only delayed response or no response trials, so that activity related to face movement did not interfere with the analysis. These models for engagement and choice were fitted with a tenfold cross-validation. For the choice model, we used two binary regressors separately for left and right choice (set to NaN during baseline epochs) set to 0 or 1 depending on the choice side and included disengaged trials in the ongoing trials:

$$R_n = \beta_0 + \beta_{F1}\sqrt{F1} + \beta_{F2}\sqrt{F2} + \beta_{F1xF2}\sqrt{F1x}\sqrt{F2} + \beta_{c1}C1 + \beta_{c2}C2 + \varepsilon \tag{8}$$

To model engagement, we included a single regressor (Engaged) that was set to 0 in non-engaged trials or 1 in engaged trials, in both baselines and during stimulation. In this model, non-engaged trials were defined as the animal not licking a spout and occurring in the last third of the daily session. Engaged trials were defined as trials with a response from the animal and occurring in the first half of the session:

$$R_n = \beta_0 + \beta_{F1}\sqrt{F1} + \beta_{F2}\sqrt{F2} + \beta_{F1xF2}\sqrt{F1x}\sqrt{F2} + \beta_{eng}\text{Engaged} + \varepsilon \tag{9}$$

Before regression was fitted, we first normalized the activity of single neurons within each session and Z-scored the concatenated vector of activity. This normalization enables the comparison of weights across cells as they would be proportional to the variance explained by the regressor. With the aim of comparing the weights between regressors, we also z-scored the regressor's values. establishing thus a direct proportionality between regressors weights and activity level. The model fitting was performed with a QR decomposition algorithm. Later, the reconstruction of sensory evidence is computed separately for each regressor as the sum of weighted neuronal activity divided by the absolute sum of weights. Pooled activity summaries (Rw1, Rw2,...) are computed from independent pools of neurons, by choosing only cells tuned to the summarized signals more than to the other alternative (e.g., pooling only neurons preferring w1 to compute Rw1, for instance). The result of this computation is a relative estimate of whisker identity strength. Another decoding strategy would be to reconstruct the frequency via maximum likelihood estimation using Bayesian inference[91]. However, we chose linear pooling for its simplicity as it provides an intuitive way of summarizing

activity over pools of neurons and does not rely on any assumption of independence between neuron's activity.

### Decoding and AUROC analysis

To construct the neurometric function, we decoded the stimulus category (i.e., F1 > F2 versus F1 < F2) by applying thresholding to weighted population averages (either Rw1, Rw2, or Rw1 - Rw2; Fig. 5B, C). For example, if (Rw1 - Rw2) > threshold, the decoded response is F1 > F2, and vice versa. An optimal threshold was determined through AUROC analysis using the MATLAB perfcurve function. In decoding of the stimulus category across different behaviors, as shown in Fig. 5D, E, we utilized the same threshold across behaviors. This threshold was computed using sets of trials equalized for stimulus conditions (e.g., an equal number of left and right choices in the same stimulus conditions). For the decoding of choice and target side in Figs. 6, 7, the fitting of neuronal weights, average pooling, computation of the threshold, and decoding were all performed within a tenfold cross-validation framework.

AUROC analysis, commonly employed in binary classification problems, serves to estimate classification accuracy. While it is conventionally set to yield values greater than 0.5, in Fig. 6A, B, we opted to use values <0.5 when the firing rate is higher for F1 > F2 trials or Choice 1 trials, and values >0.5 when the firing rate/activity is higher for F1 < F2 trials or Choice 2 trials. This method allows us to keep track of the preferred whisker (Fig. 6A) and choice (Fig. 6B).

### Statistical methods

For statistical comparisons, the Wilcoxon signed-rank test was used for paired samples, and Friedman's test when multiple comparisons of paired samples were needed. When dealing with multiple observations per mouse (comparing FOV or individual neurons), which could introduce dependencies among samples, we employed linear mixed-effects models (LME). LME mitigates these potential confounding effects, treating individual mice as random effects[92]. We have included a comprehensive table detailing the sample sizes (n), p values, the number of mice, the statistical model employed, the grouping variable, the magnitude of the effect sizes, and their corresponding confidence intervals (see Supplementary Table 2). All statistical tests were two-sided, and an alpha value of 0.05 was used to determine significance, unless stated otherwise. No adjustments were made for multiple testing unless stated otherwise.

Box plots were constructed as follows: Central mark indicates the median and the box boundaries indicate the 25th and 75th percentiles. The whiskers extend to the most extreme data points not considering outliers; outliers if they are greater than $q_3 + w \times (q_3 - q_1)$ or less than $q_1 - w \times (q_3 - q_1)$, where $w$ is the maximum whisker length, and $q_1$ and $q_3$ are the 25th and 75th percentiles of the sample data, respectively.

### Reporting summary

Further information on research design is available in the Nature Portfolio Reporting Summary linked to this article.

## Data availability

Data are available upon request to the corresponding author. Source data are provided as a Source Data file. Source data are provided with this paper.

## Code availability

Customized code is available upon request to the corresponding author.

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

## Acknowledgements

We are grateful to Dan Feldman and Gael Moneron for their helpful discussions and comments on the manuscripts. We thank Gabriel Lepoussez for providing the VGAT-cre mouse line.

## Author contributions

P.-M.G. and F.H. designed the project and experiments; P.-M.G. and S.L.G. performed optogenetic experiments; P.-M.G. performed calcium imaging and designed analysis, P.-M.G., C.R., and A.M. performed analysis; F.H., P.-M.G., C.R., and D.A.G. designed and built the experimental setup. F.H. acquired funding and supervised the project; P.-M.G. performed visualization and drafted the manuscript with inputs from F.H. All authors provided comments on the manuscript.

## Competing interests

The authors declare no competing interests.
