## [Peer Review File · Nature Communications]

Coexistence of state, choice, and sensory integration coding in barrel cortex LII/IIIREVIEWER COMMENTS

Reviewer #1 (Remarks to the Author):

This is an exceedingly elegant study in which the authors combine a well-controlled behavioral task, high density neuronal recordings, and advanced data analyses to provide new insights into contemporary debates. The study is full of gems, the most fully developed in the manuscript being the overlap between sensory and choice encoding within primary somatosensory cortex. The approaches and their implementations are rigorous, and the conclusions are novel and important. I commend the authors on generating this important contribution. While this is already a strong manuscript, I have included many comments below for the purpose of making this study even more impactful.

Major Comments

Title- The title doesn't sufficiently capture the importance of this study. I encourage the authors to update the title to reflect what they believe is the most important conclusion of the study.

Results- Analyses presented in Figure 4 are spectacular, and some of the highlights of this study. However, I believe they can be developed in further detail. Particularly the conclusion that "This comparison reveals how multi-whisker suppression increases the discriminability of neighboring stimulation frequencies." This would seem to be the case only in neurons with strong whisker selectivity, correct? Please include summary data from neural populations, rather than only example neurons (as shown in Figure 4E).

Results- For the analyses of Figure 5A-C, did this include all neurons, or only whisker selective neurons? If only whisker selective neurons, please mention this explicitly in the Results, and also include analyses from all neurons (as is likely to be experienced by a downstream reader).

Results- The authors make strong conclusions about the columnar nature of layer 2/3 which is not supported by their results. Figures 2D and 2E illustrate a graded transition of selectivity, which peaks at $|0.5|$ instead of 1. I believe their data is more in line with their own data presented in Figure 4B and the findings of Clancy et al 2015, arguing for mixed selectivity in layer 2/3. Note, this doesn't in any way nullify the elegant findings in Figure 3H-I, showing topographical organization of the sensory evidence that is used for decision-making. In fact, it is in further support of what I see as the central premise of the study, that representation alone does not prove function.

Results- Apparently contradictory findings of Figure 2D vs 3G – Cross-whisker suppression (2D) would predict dis-inhibition from the optogenetic suppression. However, optogenetic suppression is causing reduced activity of the adjacent whisker response (3G). Please elaborate in the Results and/or Discussion (possible sources could be direct optogenetic suppression or suppression of lateral excitation).

Results- Figure 3I – Please include post-hoc pair-wise comparisons of each population. Are ‘block 1’ and ‘block 2’ both significantly different from ‘control’?

Results- Figure 7F – Interpreting the Beta(F1xF2) data (green). My interpretation is that the neurons with negative B weights (below zero on x-axis) are the population of neurons showing multi-whisker suppression. The finding that these neurons increase firing with engagement (above zero on y-axis) would be consistent with the importance of these neurons for sensory discrimination (Figure 4E, middle).

Discussion- Some of the pupil findings seem odd to me. Pupil dilation with disengagement (Figure 7A) contrasts with decades of literature regarding the relationship between arousal and task engagement. Please discuss your findings in context of the existing literature.

Discussion- “This is a first indication that behavioral variability does not stem from alterations of sensory encoding but could originate rather from state related fluctuations in downstream processes” – seems like a strong conclusion, given ~70 years of recording sensory responses from neocortex under anesthesia. Perhaps reword?

Discussion- Choice probability in S1. I believe the Sachidhanandam et al. 2013 study is not interpreted correctly here. My read is that they attribute choice processing to the late activity. Also, other studies have compared CP from multiple areas in whisker-based tasks. For example, Zareian et al., 2021 demonstrated CP in motor cortex emerging before CP in S1, which may account for late CP signals in S1. The Zareian study should be added to the Discussion.

Minor Comments

Pg 7, line 17 – Not clear what type of optogenetic stimulation is being implemented here. Please clarify which neurons are being stimulated and for what purpose.

1E legend refers to ‘circles’ which are not evident in the figure.

P15, line 31- I believe should refer to 7E.

Figure 7E “dimension”

P18, line 3, odd use of “Besides”

P20, line 30 “tThese”

Reviewer #2 (Remarks to the Author):

The paper is valuable both in presenting 1) a dataset of calcium imaging population recordings from somatosensory cortex in a difficult, elegant, interesting and well-conceived task and 2) in aiming to investigate behavioral correlates in information encoded in neural activity rather than in activity itself.

While the dataset appears great and the questions of great interest, it seems to me that the paper still falls short of really making a breakthrough in understanding how sensory information is encoded and used for computations in populations of neurons.

The main conclusions from the abstract that perception depends on the formation of early sensory representations and that stimulus and choice signals (as well as engagement) are partly decoupled does not seem neither strongly supported by the data nor particularly unexpected or novel. In what follows, I give my constructive suggestions on how the analyses and conclusions could be improved and how further analyses of the dataset could relatively easily lead to more interesting advances. I hope that the authors find some of these suggestions to be useful.

The whole column optogenetics suggests that the somatosensory area is involved in perception of sensory stimuli, but it is not entirely surprising (because this area has been long implicated in perception) and also does not bear evidence for the sensory populations code proposed and investigated here to be relevant for perception (it will perturb all neurons not only those in the imaged layers, etc). In my view, this seems more of interest as a compelling and finely resolved optogenetic screening to prove that the investigated recording locations are involved for the task.

Also, the strength of the claims of almost orthogonality between stimulus and choice signals is difficult to evaluate. The double regression with stimulus and choice will first pick up the main selectivity of the

neuron (stimulus or choice) and then the residual selectivity which is not explained by the other. It is thus unclear whether selectivities are really similar (e.g. whether neurons that prefer stimuli associated with lick left also prefer the stimulus associated with lick left).

However, the manuscript presents a nice discovery in terms of sensory coding. It shows the presence of an interaction term between the stimulations of the two whiskers (Eq 1), and the authors argue its important for sensory coding. (This has conceptual similarities with the recent study of Kira..Harvey Nature Communication 2023, that showed population codes with the presence of interaction terms between two types of sensory cues in 2AFC tasks) The authors use this result to argue that it is better to decode stimuli using the difference of activity, which does not seem convincing to me. I think that, instead, it would be really interesting if the authors could show that the presence of the interaction term is key to increase the information encoded in single trials by the population (for example, computing decoding accuracy from real data using decoding models that do or do not have the interaction term). Also, it would be interesting to study if this term has an impact on accuracy of behavior. The authors report that their sensory information population code along the “difference between pools” has single trial information that is not diminished in incorrect trials with respect to error trials. The authors argue that errors are not due to failures in sensory coding. One alternative possibility is that the sensory code based on differences of activity (neglecting the non-linear interaction term) is not the sensory code used by the brain, but that the interaction between the two stimuli is instead the one encoding the key information. It would be interesting to test this hypothesis computing whether the population code including the interaction term (or alternatively, neurons that have primarily an interaction encoding) has information that is critically different between correct and error trials. In Kira et al Nature Communications 2023, the authors did not find information difference between correct and error trials in population codes ignoring interactions between sensory cues, but they instead found that information in correct trials was diminished specifically in the interaction between the cues. Performing a similar analysis on these data would be of interest, and it would provide novel insights in the role of mixed selectivity for sensory coding.

Minor. Page 11, line 24..”is best for solving the task”. The use of “best” suggests that the author demonstrate that the difference in activity is optimal, whereas the author seem only to compare it to two alternatives. I suggest either proving optimality or write “better”.

Reviewer #3 (Remarks to the Author):

Studies have shown that neurons in the barrel cortex respond to sensory stimuli as well as the animal's choice in behavioral tasks. Yet, it has remained unclear if these aspects are encoded in the same population of cells. Gardères et al. clearly show that different neuronal populations code for choice and input differentiation in S1 in this elegant study. .

Towards this aim, Gardères et al. developed a novel 2-AFC task in which mice were trained to compare stimulation frequencies applied to two adjacent vibrissae and reported by licking right or left based on the frequency difference of the two stimuli. Using calcium imaging of individual neurons, optogenetic silencing, and advanced analysis using linear regression models, they found that neurons encode the properties of sensory input, animal choice, and engagement in the task. These varied substantially but are orthogonal to each other. That led them to conclude that perceptual variability does not originate from sensory encoding neurons but from state or choice fluctuations in downstream areas. The experimental design is exceptional, and the methods and analysis are superb.

Accordingly, the delayed 2-AFC task, which they designed, effectively separates the animal's actions from the stimulation time. This is crucial for testing the hypothesis and is more reliable than the go/no-go paradigm. The analysis supporting the hypothesis is presented in figures 6-7. Both figures are based on a regression model that effectively supports their conclusions.

However, the manuscript contains numerous grammatical/spelling errors and formatting issues that detract from its overall quality and readability. These problems should be addressed to improve the text's clarity and coherence. In addition, while I strongly support this paper for publication, it was difficult to understand both the experimental details and the analysis in particular. The explanations for the methods used to analyze the data depicted in Figures 4 to 7 are limited, making it difficult to comprehend the information presented in each panel without making assumptions. The methods section is unfortunately scarce in explaining the analysis and did not help here. The authors should add more explanations both when describing the results and also in the methods section. I suggest the authors test their changes by sending the paper to colleagues and students and see if the figures and text are clear.

General remarks

Fig. 2F - Add color so it will be possible to distinguish between the lines.

Fig. 2G - Present the complementary stimulation frequencies. In other words, if data exists, present also 0/10; 0/45; 0/90.

Fig. 5C - Why is Rw2 missing in the panel? Please explain if there is a reason for that. Is it similar to Rw1? If not, explain.

P2 L36 - "two adjacent whiskers on the same row of the whisker pad (e.g. B1/C1)". - B1 and C1 are in different rows. Should it be: two adjacent whiskers on the same column of the whisker pad?

Add to methods how intrinsic imaging was performed, which system, and how the barrel borders were delineated.

Add a scale bar size to Fig. 2A - intrinsic imaging

P11 L29-31 - (Fig. 5D) - "If the behavioral read out depends directly on the relative sub-populations firing rate, we should observe a left/right shift in the neurometric functions drawn from trials with left/right choice"

A clarification is needed on how the shift would look like for the two curves. Could such a shift be simulated and shown as an additional panel/line?

P11 L29-37 - A more in-depth explanation and clarification of the graphs are needed to understand the concepts better.

P12 L22 - More information on the AUROC is needed in the Methods. Not all readers are familiar with this method. In particular, it is not well explained in the context of the plots in panels 6A and B.

P 13 - second paragraph - Please explain why one choice factor (R_{c1}) predicts the choice much better than the other choice factor. Is it experimental bias?

Fig. 6F - Write $f_1 > f_2$ and $f_2 < f_1$ on top of each panel to speed up understanding of the graphs.

Fig. 6F. Can the two point sets be plotted in different distinguishable colors?

Fig. 6 - Explain why $\Delta F > 30$ Hz was selected.

Fig. 6E bottom panel - why do the curves for R_{c1} and $R_{c1}-R_{c2}$ overlap? Any meaning for this or is it expected?

What are the results for trials with a big frequency differences ($\Delta F > 30$ Hz)?

Fig. 7 G-H - How come that similar labels have a three orders of magnitude difference? Perhaps it is obvious to the authors, but this discrepancy is an example of the limited explanations in the methods and results.

Fig. S1B - Show the real data (along with the fits).

Reviewer #4 (Remarks to the Author):

Gardères et al. develop a novel 2AFC whisker discrimination task and use it to study choice predictive activity in mouse whisker S1. Specifically, they stimulate two whiskers, with one whisker receiving a higher frequency and the other a lower frequency. The animal must indicate the whisker that was given the higher frequency by licking one of two lickports. First, they show how different populations respond to each whisker, reflecting the somatotopy of the area. Next, they perform an elegant barrel-specific silencing experiment that demonstrates the whisker-specific contribution of the stimulus they provide, imaging with 2P simultaneously and thereby demonstrating how the neural perturbation relates to the activity changes they induce. They then show that cross-whisker suppression is a key contributor to neural activity. Next, they show that it is fairly easy to decode the frequency difference from the sensory stimulus, with only 6 neurons needed to reach animal performance; larger pools of neurons exceeded animal performance. Finally, they show that sensory activity does not predict choice, and that engagement does not explain whisker-evoked activity differences.

Overall, the data are of high quality, the analyses are well done, and I found the paper fairly clearly written. However, there are issues - both major and minor - that need to be addressed before publication.

Major:

1 The task design was probably my favorite thing in this paper. This is a great behavior and will be useful for many questions. I really commend the authors on this - training mice for 3+ months is no small thing, and developing a true 2AFC task in mouse vS1 is really impressive.

2. It is unclear to me what trials are used in the decoding analyses. In Fig. S2D, the authors very commendably segregate trials based on whether orofacial (nose) movements could be discerned. It

sounds, however, as though perhaps most analyses (Figs. 4 onward) that purport to look at choice look at both trials with and without early movements; only trials with pre-sensory movements are excluded, from what S2D indicates. The major analyses in the paper that are looking at choice should only use trials without early movement, as any movement will contaminate the result with reafference. I *think* this is being correctly done based on Fig 6.E, but it was very difficult to find a clear and concise statement to this effect; please make it very very clear that you ONLY use the trials with no early movement. If this is not the case, analyses should be restricted to these trials only.

3. There are two major statistical issues I see with in this paper:

First, in several analyses, the authors aggregate across all neurons in the dataset (e.g., Fig. 2E, 4B, many Fig. 6 and 7 panels). This is highly problematic. First, there is clearly a large variability in neuron count and this skews the results toward specific animals. Second, within-animal one can reasonably expect things to co-vary, and independence assumptions are thus very much violated relative cross-animal. It would be ideal to adopt per-animal summary statistics and then do any tests on those statistics. For instance, in 6A, show an example animal's choice propability histogram and then do statistics across the medians from each animal.

Second, the authors segregate animals into 11 FOVs and use these as the level of analysis. Sometimes this may be appropriate, but in many cases, within animal behavior and neural activity will vary less than across animals, and using FOVs as the independent observation stops being appropriate. At minimum, this needs to be discussed, but I would recommend against this in most cases used here; use the animal as the level of analysis.

4. I found many of the population analyses problematic. Fig. 4A histogram shows that, as with many vS1 studies, a small minority of cells is actually doing the interesting work. If you look at all the cells in all your analyses (Figs. 4B, 6, 7), you will generally be recording noise and masking the few neurons that are doing interesting things. This likely accounts, for example, for the "orthogonality" results in Fig. 6F and Fig 7E. This analysis would be far more compelling if it were restricted to the most discriminative/sensory responsive neurons. Do these have a choice signal? Do they show modulation by engagement? As it stands, I don't find F6 and 7 particularly compelling. An easy fix would be to shift these analyses towards single neurons, and focus on the most interesting ones.

I would recommend 1) Fig. 6: subselect the 5-10% of neurons that have the highest/lowest CP ; repeat the analyses with only these cells, and you may see something more interesting; 2) Fig. 7: subselect neurons with highest sensory selectivity > how much are these neurons impacted by engagement?

Along this line, while you do cite Buetfering 2022, I think this is really undercited as it is incredibly relevant to the present work and seems to show the opposite -- that a few neurons do have choice modulation and that manipulating them perturbs behavior. At minimum this should be more explicitly discussed.

5. Behavioral example images are missing and need to be added. Especially with regards to the analysis in Figure S2, which to me is a linchpin of this study since early movement will contaminate the choice signal and put the conclusions of the paper into question.

Minor:

1. In general there are several things that are unclear / difficult to discern: how many neurons and, if applicable, how many from what category ; depth at which you imaged; how many days post injection ; what whiskers were used ; how many neurons per FOV and how many FOVs per animal - I would recommend a table to make this clear, as there are so many different things being done.

2. Please refrain from calling yes/no tasks, like the typical MT dot kinetogram, 2AFC. I am not sure where this error originated, but it is clearly pervasive in the literature, with luminaries like Britten and Salzman committing it. 2AFC has a clear relationship to yes/no tasks mathematically in terms of psychometric performance because 2AFC tasks have two stimuli whereas yes/no tasks have one. Both have two response contingencies. More information in the most basic case means $\sqrt{2}$ improvement in certain performance metrics for a 2AFC task vs. equivalent yes/no task. Romo's task is indeed 2AFC (2IFC, to be specific, but the math is the same), as is the present task. If you have two dot kinetograms and you ask the monkey to indicate the one with, e.g., more motion then you have yourself a 2AFC task. One dot field = yes/no, which is what Salzman and the rest of that literature use.

3. I disliked the characterization of go/nogo as problematic due to the early onset of movement. That is not a fundamental problem of go/nogo designs; it is a problem of designs without a delay period and exclusion of trials where such movement occurs. Please adjust the relevant paragraph.

4. Behavior: this task is a potentially important contribution to the field; please quantify other basic psychometric parameters (lapse rate, bias). Perhaps a vertical dotted line at $F1-F2=0$ and $\%F1 \text{ higher}=50$ would be of use in 1E.

5. Fig 1C: can you use color to make it possible to see individual animals? as it is there are too many to discriminate the individual animal lines; propagate this to E

6. Fig 2C: I can understand why you opted to have time be vertical, but that is, at least for me, rather odd. Please see if you can't rotate it so time is horizontal as it is almost everywhere else.

7. Depth of recordings is never reported ; please report it

8. Choice probability should be computed s.t. the more common occurrence is values > 0.5 . Reported values of 0.486 are equivalent to .514, it is an arbitrary choice how one computes it (also, please don't do the pooling neuron aggregation -- that is bad for reasons outlined above).

9. You use chronic AAV-based expression of jRGECO and GCaMP but training takes nearly 3 months. This makes me worry about expression-based cytopathology. Please provide some data showing that your cells are not filled or some other control demonstrating that cells are still healthy that long after infection.

10. Z-scoring of dFF traces does not make sense. Proper dFF calculation will appropriately normalize so that 0 is 'correct' (equivalent to mean subtraction in proper z-score). As you did quite nicely with the videography analysis, you should instead compute a noise statistic (e.g., variance of the dFF signal during periods of inactivity) and divide dFF by that to get an 'noise-adjusted' dFF. That is all that needs to be done to a properly computed dFF trace.

Z-scoring in the conventional sense will be especially bad as distribution of dFF in active neurons are long-tailed, and the mean is actually not the correct additive normalizer - FO is, and presumably this is already incorporated in the dFF measurement.

Color and formatting code:

Original comments from the reviewer*

Response from the authors

“Original manuscript text”

Addition to the original manuscript

~~Suppression of text~~

*Note: we split and assigned numbering to all reviewers comment to ease the point by point discussion: M stands for Major comment, m for minor comments, R for reviewer. We refer to distinct reviewers points as e.g. “R2M3” or “R4m2”

Page and line number (e.g. P3L12) refers to the revised manuscript without highlighted changes.

Reviewer #1 (Remarks to the Author):

This is an exceedingly elegant study in which the authors combine a well-controlled behavioral task, high density neuronal recordings, and advanced data analyses to provide new insights into contemporary debates. The study is full of gems, the most fully developed in the manuscript being the overlap between sensory and choice encoding within primary somatosensory cortex. The approaches and their implementations are rigorous, and the conclusions are novel and important. I commend the authors on generating this important contribution. While this is already a strong manuscript, I have included many comments below for the purpose of making this study even more impactful.

Major Comments

M1 - Title- The title doesn't sufficiently capture the importance of this study. I encourage the authors to update the title to reflect what they believe is the most important conclusion of the study.

We considered an alternative title for the study carefully. However, we believe that the current title effectively conveys the information about the most important aspects of the study. We appreciate the reviewer's encouraging comments.

M2 - Results- Analyses presented in Figure 4 are spectacular, and some of the highlights of this study. However, I believe they can be developed in further detail. Particularly the conclusion that “This comparison reveals how multi-whisker suppression increases the discriminability of neighboring stimulation frequencies.” This would seem to be the case only in neurons with strong whisker selectivity, correct? Please include summary data from neural populations, rather than only example neurons (as shown in Figure 4E).

Our conclusion that multi-whisker suppression improves discriminability describes generalization of discrimination across the stimulus space (as described in Fig. 4E). Following this comment and others, we investigated further the increase of whisker discriminability due to multi-whisker (MW) suppression in two additional supplementary figures (Fig S6 and Fig S7).

In Fig S6A-B, we explored the impact of the MW in single neurons on decoding the target $F1 > F2$ in real data. An obstacle is that MW suppression is strongly correlated to single whisker coding and selectivity across neurons (as expected from Fig. 4B). Thus to visualize the impact of MW suppression on each neuron's discrimination ability, we split and binned neurons depending on both their selectivity index and the fitted weight of the MW effect. We find that discriminability increases with MW suppression, independently of the whisker selectivity. Discriminability improvement is measured as increase in AUC (generalize the $F1 > F2$ comparison in the stimulus space, Fig S6A) and as an increase in fraction decoded correct (Fig S6B).

In Fig. S6C-D we included summary data from the neural population defined as weighted pools $Rw1$ and $Rw2$, as suggested by the reviewer. We note that at the whole population level, MW suppression is relatively low given the heterogeneity of MW suppression across single neurons, but clearly sublinear. Interestingly, the response to the non-preferred whisker is slightly delayed compared to that of the preferred whisker. MW suppression seems maximal at stimulus onset, and lower in the subsequent phase of sustained activity.

In Fig S7, and following comments R2M4, we tested whether the interaction has an impact on decoding real data by comparing two models: (1) decoding with the interaction term and (2) without the interaction term (Interaction variable was shuffled prior to fitting single neuron's activity). We found that the interaction term did not change decoding of the entire population, nor of subpopulations comprising neurons with MW enhancement or no MW integration. But integrating the MW term to neurons with MW suppression significantly improved decoding (Fig S7).

In summary, these additional results further show that the MW interaction term is beneficial to decoding the target $F1 > F2$ in neurons with MW suppression. This improvement generalizes to populations of MW suppressed neurons but not to the decoding from the entire population.

Supplementary Figure 6: Multi-whisker (MW) suppression improves decoding performance at the single neuron level, but has little net effect on the whole population average.

A, Discrimination of $F1 > F2$ over the stimulus space depends on selectivity and MW suppression across neurons. Discrimination is measured as the area under the curve (AUC) of $F1 > F2$ over the entire stimulus space, as in Fig 4E. The pixel colors represent the average AUC value for neurons in the bin. The neuronal population from all animals is split into bins defined in two dimensions on their MW suppression ($Bf1 \times F2$; y-axis) and their absolute selectivity index (SI; x-axis). Note the increase of AUC along both x and y axis, suggesting the MW suppression improves discriminability independent of its relationship with whisker selectivity.

B, Decoding performance depends on selectivity and MW suppression. Decoding performance is quantified as the fraction of correctly decoded trials. Decoding was performed with a GLM using logit as the link function and assuming binomial distribution of $F1 > F2$. Binning performed as in A. Please note the increase of AUC along both the x and y axis.

C-D, Neural population pools R_{w1} and R_{w2} respond mostly to whisker 1 and 2 but also slightly to the adjacent whisker. R_{w1} and R_{w2} show sublinear integration of the two whiskers stimulated simultaneously. R_{w1} and R_{w2} were computed from two independent pools of neurons based on the sign of $Bf1 - Bf2$. Error shades represent s.e.m. across FOV. Firing rate and fitted firing rate in the middle and right plots are averaged across FOV. Note that the model with interaction does not fully account for the population firing rate which might be due to the diverse tuning within the population, including MW enhancement (Fig. 4b).

Supplementary Figure 7: Impact of multi-whisker (MW) interaction term on (sub)-population decoding performance.

A, Schematic procedure for decoding **with** or **without** interaction. When decoding without interaction, the regression variable F1xF2 is shuffled from trial to trial in the model without interaction. Each neuron activity is fitted trial per trial as described in Fig. 4. Activity of all neurons (or subpopulations) is combined into R w1, R w2 and R w1xw2, (respectively latent representations of F1, F2 and F1xF2). The target side (defined by the sign of F1-F2) is decoded separately by three combinations of latent variables. Decoding is performed using a GLM with logistic regression.

B, Comparison of decoding **with** or **without** interactions in different subpopulations of neurons (rows) with decoding of different combinations of latent variables (columns). Subpopulations contain 10 % of neurons with most MW enhancement (top), 10 % null MW interaction (middle) or 10 % most MW suppression (bottom). The interaction term significantly improves decoding performance for the subpopulation of neurons with MW suppression. P-value is computed with linear mixed effect model across FOV using mice identity as grouping variable.

In Fig. 4B, we added a “0” line to separate MW suppression from MW whisker enhancement, and illustrate the net effect of multi-whisker integration across neurons.

We modified the paragraph title and main text as follow P10 L17:

Multi-whisker suppression improves and generalize neighboring frequency comparison across the stimulus space ~~Multi-whisker suppression enables generalization of the frequency comparison task in the stimulus space~~

“At the level of the whole population, multi-whisker (MW) integration drives only a subtle sub-linear integration (Fig. S6c-d), likely due to heterogeneity of MW integration across the population. Overall, ~~t~~This regression analysis confirms that single neuron’s response increases with stimulation frequency, and highlights multi-whisker integration as a contributing factor shaping the activity of L2/3 neurons. ~~and shows that multi-whisker suppression is an important factor of L2/3 neuronal activity.~~

Does ~~multi-whisker suppression~~ MW integration help solving the comparison task? To evaluate how this computation affects activity level in the stimulus space, three example neurons were compared: One with ~~multi-whisker~~ MW suppression, one without ~~multi-whisker~~ MW integration and one with ~~multi-whisker~~ supralinear MW responses (neurons 1, 2 and 3 respectively in Fig. 4c-e). The neuron 3 with ~~multiple-whisker enhancement~~ supralinear MW responses shows ~~no little~~ difference in activity level on either side of the boundary task (quantified by area under the curve measurement; AUC = 0.65). Therefore, its activity level is ~~not little~~ informative in regard to the task, and may rather represent a source of noise in our paradigm. ~~On the contrary, multi-whisker suppression~~ By contrast, MW suppression in neuron 2 leads the gradient of activity in the stimulus space to become perpendicular to the task boundary. Activity level ~~reflects~~ scales better to the difference in stimulation frequency F1-F2 rather than to F1 the stimulation frequency of their the neuron’s preferred whisker (Fig. 4e). Therefore, the output of this neuron encodes the stimuli in a relevant way for the generalization of frequency discrimination, as compared to the neuron without suppression. This comparison reveals how multi-whisker suppression increases the discriminability of neighboring stimulation frequencies” As a result, this neuron’s output encodes MW stimuli in a manner that is relevant to the generalization of F1-F2 discrimination across the stimulus space (AUC = 0.97; Fig 4e). Across the population, we find that both the AUC across the stimulus space and direct decoding performance increase with the degree of MW suppression (Fig. S6a-b). Besides, the comparison between two models, one with a MW interaction term and one without, indicates that populations of neurons exhibiting MW suppression derive benefits from the MW integration term in decoding F1>F2. Together, these findings reveal how MW suppression improves and generalize the discriminability of neighboring stimulation frequencies.”

M3 - Results- For the analyses of Figure 5A-C, did this include all neurons, or only whisker selective neurons? If only whisker selective neurons, please mention this explicitly in the Results, and also include analyses from all neurons (as is likely to be experienced by a downstream reader).

In Figure 5, 6 and 7 (Except 5B), all neurons are included, including those with no or little whisker selectivity. We did so as we share the reviewer's view that a readout of the downstream reader is likely based on the overall activity in all neurons. Therefore, we intend to keep the figure unchanged.

Regarding other reviewers' comments, we included analyses restricted to subpopulations tuned to single whisker, MW suppression, choice or the interaction of choice and sensory coding (Fig. S7 and Fig. S11). The sensory readout was reliable across behaviors in all these subpopulations and representations of choice versus stimuli remained orthogonal (unless the readout is "rotated" or selective to specific combinations of neurons). Please see our response to R4M4 for these detailed analysis and corresponding changes in the text.

M4 - Results- The authors make strong conclusions about the columnar nature of layer 2/3 which is not supported by their results. Figures 2D and 2E illustrate a graded transition of selectivity, which peaks at |0.5| instead of 1. I believe their data is more in line with their own data presented in Figure 4B and the findings of Clancy et al 2015, arguing for mixed selectivity in layer 2/3. Note, this doesn't in any way nullify the elegant findings in Figure 3H-I, showing topographical organization of the sensory evidence that is used for decision-making. If fact, it is in further support of what I see as the central premise of the study, that representation alone does not prove function.

As suggested by the reviewer, we reformulated parts of the results text as follow P5 L27:

"IN population selectivity indexes were lower than the EN population, but ranged on a similar scale ($p < 0.001$, $|\mu SI_{IC}| \sim 0.33$, $|\mu SI_{EC}| \sim 0.40$ Mann-Whitney U test, Fig. 2e). **Whisker selectivity in both ENs and INs was intermixed spatially in a salt-and-pepper organization with a graded transition between the two columns (Fig. 2e).** ~~INs selectivity was also somatotopically distributed (Fig. 2e).~~ These results highlight the functional selectivity of inhibitory and excitatory neurons in two neighboring microcircuits **with heterogeneous selectivity at the single neuron level**. The spatial transition of functional selectivity illustrates ~~the underlying~~-somatotopic organization in cortical columns."

And in the discussion P19 L4:

"As described in previous work, we found that evoked activity in L2/3 **displays a somatotopic organization with salt-and-pepper whisker tuning of single neurons,** ~~is somatotopically organized (Woolsey and Van der Loos 1970; Clancy et al. 2015),~~ is sparsely distributed in the population (O'Connor et al. 2010; Mayrhofer et al. 2015; Barz et al. 2021) and adapting rapidly in most neurons (Musall et al. 2014; Gerdjikov et al. 2018). "

More importantly, we investigated the somatotomy of choice-related activity and found a weak, reversed somatotopic distribution (Supplementary Figure 11 A-B). We incorporated this aspect to support our interpretation of the optogenetic experiment results in the discussion. The weak/reversed somatotomy of choice signals suggests that our optogenetic manipulation selectively influences activity on the sensory axis but not on the choice axis. Hence, this would support the idea that the animal predominantly utilizes the sensory axis for its trial-by-trial perceptual readout.

We reframed the discussion section “Single column activation drives perception of the sensory feature it encodes” to incorporate this result and interpretation P20 L6:

“Thirdly, our experiment partially addresses the identity of signals used for the perceptual readout of S1 activity. A recent study (Buetfering et al. 2022) reported that selectively manipulating choice, but not sensory neurons, influences behavior during the session, raising the possibility that the animal relies on choice neurons only for its decision-making. In our task, choice-related activity exhibits weak and reversed somatotopy compared to sensory-related activity (Supplementary Figure 11). Hence, the optogenetic manipulation is aligned with the sensory representations but not the choice representations, which should result in a selective shift in the sensory representation only. Consequently, our results argue against the readout of choice neurons only and suggest that the trial-by-trial readout predominantly relies on the sensory encoding dimension.”

M5 - Results- Apparently contradictory findings of Figure 2D vs 3G – Cross-whisker suppression (2D) would predict dis-inhibition from the optogenetic suppression. However, optogenetic suppression is causing reduced activity of the adjacent whisker response (3G). Please elaborate in the Results and/or Discussion (possible sources could be direct optogenetic suppression or suppression of lateral excitation).

It seems that the results referred to here are those showing cross whisker suppression (2G) and suppression through optogenetic silencing of the adjacent barrel (3D).

In physiological conditions (i.e. without optogenetic), stimulation of a single whisker activates its principal column and much more moderately the adjacent column (2G left). Only when both whiskers are stimulated simultaneously, do we observe cross-whisker suppression (Fig 2G right and Fig. 4). However, in our recordings of the optogenetic effect on the two barrels, we only used conditions where a single whisker is deflected (Fig 3 A-D) and observe suppression of the adjacent whisker response. As suggested by the reviewer, we interpret these results as direct optogenetic suppression, possibly due to (1) recruitment of adjacent column's INs (ChR2 in INs' cross columnar dendrites) and (2) cross columnar inhibition of ENs by the columnar INs. We provide a description of these “off-target” effects in Fig S4.

In a hypothetical experiment where the response of the two columns is monitored during multi-whisker stimuli and optogenetic silencing, we agree with the reviewer's prediction of dis-inhibition, although it might “compete” with the off-target effect. This experiment would be interesting to investigate the cortical origin and circuits of cross-whisker suppression. However, we believe this is beyond the scope of the current study.

To communicate our interpretation better, we modified the text P7 L35:

“... optogenetic stimulation decreased sensory response to the preferred whisker in both target and neighboring columns, possibly due to direct inhibition or network dependent effects as described in Figure S4.”

and the discussion P20 L15:

“From a technical perspective, some limitations of our approaches could be overcome in the future. For instance, we observed inhibition outside of the stimulated area (Figure S4), a challenge that could be mitigated by the use of soma-targeted opsins. ~~Activation of inhibitory neurons induces a spread of inhibition larger than the stimulated area (Figure S4).~~”

M6 - Results- Figure 3I – Please include post-hoc pair-wise comparisons of each population. Are ‘block 1’ and ‘block 2’ both significantly different from ‘control’?

The blocking effects are in the expected directions. However, when using the Dunn-Sidak post-hoc comparison test, we find that blocking W2 versus control is significant but blocking W1 versus control is not (N = 8 animals):

blockW1 - Control	p = 0.94
blockW1 - blockW2	p = 0.004
Control - blockW2	p = 0.018

In our opinion, this lack of significance is due to the low sample size and the systematic bias towards ipsilateral response during optogenetic inhibition that we reported in the absence of whisker stimulation (2.2 to 5.9 % change in response side, which would accentuate effect of W1 blocking and reduce effect of W2 blocking). Given the low number of animals, we tested a different statistical approach to exert stronger statistical power.

We used a generalized binomial linear mixed model (GLME). It is an extension of the mixed effect model (LME) to fit binomial variables (left/right choices here). The model takes 15412 trials as n-value and predicts response side from stimulation frequencies and from the different blocking conditions as fixed effect (W1, W2 or both). In this analysis, we include the animal identity as a random effect to control for dependence between trials collected in the same animal.

Using the GLME, we find that W1 blocking is significantly promoting choice 2 (B = 0.10; 95% CI [0.056; 0.153]; p=0.0003) as compared to baseline while W2 blocking is significantly promoting choice 1 (B = -0.081 ; 95% CI [-0.150 to -0.013], p = 0.041) as compared to control/baseline.

Overall the behavioral effect is more subtle than we anticipated. We believe that this is due to the “off-target” inhibition in the adjacent column and/or at further distance. A simple strategy in the future would simply be to train the animal to discriminate between non-adjacent whiskers.

We included the results of this test in the text accordingly P8 L1:

“The single column inhibition **shifted significantly** ~~induced a bidirectional behavioral bias in the~~ psychometric curve depending on the silenced barrel (psychometric bias, p =0.002, Friedman test, n= 8 mice; Fig. 3i). **The trial by trial analysis shows a significant bidirectional behavioral effect with a bias of W1 blocking versus control (p < 0.001) and W2 blocking versus control (p = 0.041), tested using a Generalized Linear Mixed Effects (GLME) model on n = 15412 trials, (supplementary table 2).**”

Note that we modified most of our statistical approaches to take into account animal identity in all analysis; and provided detailed statistics in the new supplementary table 2.

M7 - Results- Figure 7F – Interpreting the Beta(F1xF2) data (green). My interpretation is that the neurons with negative B weights (below zero on x-axis) are the population of neurons showing multi-whisker suppression. The finding that these neurons increase firing with engagement (above zero on y-axis) would be consistent with the importance of these neurons for sensory discrimination (Figure 4E, middle).

We agree with the reviewer's interpretation. Besides, we showed that these neurons are often the most active and discriminate the stimulus better than non-suppressed neurons (please see our response to previous comments R1M2).

We modified the figure to make the axis label more readable: multi-whisker suppression is the best predictor (among B1 B2 and B1x2) for engagement related modulation, and the 10% neurons with strongest MW suppression tend to increase their activity level by ~30% in the engaged condition. We modified the text and figure P17 L3:

Weights for single whisker frequencies (β_{F1} or β_{F2}) are positively correlated to engagement related gain, **whereas the while weights for supralinear whisker interaction (β_{F1xF2}) of the two whiskers are negatively correlated with engagement modulation. Using a GLME model, we found that among the three sensory weights, β_{F1xF2} best captures engagement-related modulation related to sensory selectivity (Supplementary Table 2). ...**

More analyses were focused on these neurons in our reply to reviewers' comments R2M5 and R4M4, but we did not find prediction of choice in these neurons specifically. Nevertheless, the discussion section "**Sensory information and representation across states**" was modified to emphasize the importance of these neurons for sensory discrimination and possibly for behavioral readout.

M8 - Discussion- Some of the pupil findings seem odd to me. Pupil dilation with disengagement (Figure 7A) contrasts with decades of literature regarding the relationship between arousal and task engagement. Please discuss your findings in context of the existing literature.

In the current study when mice are engaged in the performance of a 2AFC task we find the pupils are smaller during the engaged state vs. states of task disengagement (Fig. 7b). While multiple previous studies reported increased pupil size under states of arousal and task performance (Vinck et al., 2015; McGinley et al., 2015; Lee & Margolis, 2016) these studies employed go noGo tasks and not 2AFC as in the current study. Our previous study using a 2AFC task also reported larger pupil size during periods of disengagement but with stimulus-evoked pupillary responses smaller vs the engaged state (Ganea et al., 2020). The discrepancy between these reports might arise due to different cognitive requirements during task performance. It has been shown that the activity emanating from different neuromodulatory loci correlates with pupillary dilations (Reimer et al., 2016), hence the summed effect of these systems might lead to different observed pupillary phenotypes depending on task requirements in terms of cognitive load or attention.

We updated the discussion as follows P21 L29:

"In our task, the engaged state is associated with a pupil contraction during pre-stimulus period, **in line with recent work using a whisker mediated 2AFC (Ganea et al. 2020), although at odds with earlier findings (Vinck et al. 2015; McGinley et al. 2015).** We also observed, ~~and~~ a suppression of slow oscillatory activity (Jacobs et al. 2020)."

M9 - Discussion- "This is a first indication that behavioral variability does not stem from alterations of sensory encoding but could originate rather from state related fluctuations in downstream processes" – seems like a strong conclusion, given ~70 years of recording sensory responses from neocortex under anesthesia. Perhaps reword?

We acknowledge the reviewer's concern that our statement may be misleading. Following the revision, we have extensively revised this paragraph in the discussion, and this

statement has been removed. Please refer to the updated discussion section for the revised content. **“Sensory information and representation across states”** P21 L28

M10 - Discussion- Choice probability in S1. I believe the Sachidhanandam et al. 2013 study is not interpreted correctly here. My read is that they attribute choice processing to the late activity. Also, other studies have compared CP from multiple areas in whisker-based tasks. For example, Zareian et al., 2021 demonstrated CP in motor cortex emerging before CP in S1, which may account for late CP signals in S1. The Zareian study should be added to the Discussion.

We agree with the reviewer that the Sachidhanandam et al. 2013 study found choice related signals mostly in the late component of S1 activity (50ms>X>150ms). However, in their optogenetic manipulation experiment, they disrupted either the early sensory activity or the late component of S1 activity and found that disruption of the early sensory activity had a larger impact on behavioral detection than disrupting the late component. We were referring to this result on causality of early sensory activity. We have partly reformulated the relevant section of the discussion. We thank the reviewer for the very relevant reference Zareian et al., 2021. It has been added to the bibliography and integrated to the discussion as follows P21 L14:

“Previous studies have indicated that subjects use the earliest temporal component of sensory activity to form their choice (Nienborg and Cumming 2009; Sachidhanandam et al. 2013), while the manipulation of later components correlating with decision signals have ~~little less~~ **little less** causal influence **onto them (Sachidhanandam et al. 2013)** and originate most likely from top-down modulation (Zareian et al. 2021; Bondy et al. 2018; Zhao et al. 2020). **Our results are in agreement with a different origin of sensory versus choice signals and further show that these later decision signals are unrelated to sensory encoding at the population level.** ~~Our results further show that these later decision signals are mostly unrelated to sensory encoding at the population level.”~~

“Zareian, B., Zhang, Z., & Zaghera, E. (2021). Cortical localization of the sensory-motor transformation in a whisker detection task in mice. *Eneuro*, 8(1).”

Minor Comments

m1- Pg 7, line 17 – Not clear what type of optogenetic stimulation is being implemented here. Please clarify which neurons are being stimulated and for what purpose.

P7 L21:

“As the two sensory alternatives have distinct spatial representations, we evaluated the possibility of manipulating them selectively. Optogenetic wild-field stimulation was combined with two-photon functional imaging (Fig. 3a; see methods) (Prsa et al. 2017). **Channelrhodopsin was conditionally expressed in GABAergic neurons (VGAT cre-line), whose optogenetic stimulation reduced the activity in the local network “**

m2- 1E legend refers to ‘circles’ which are not evident in the figure.

These circles were displayed in a previous version of the figure. This statement has been removed.

“E, Psychometric function drawn from conditions within the rectangle in D. (i.e. with a constant summed frequency $F1 + F2 = 90$ Hz). ~~circles represent average performance and thin traces are logistic fits for individual animals (see methods). ...~~”

m3- P15, line 31- I believe should refer to 7E.

“We next represented neuronal pooled activity in the engagement-related dimension as a function of the neural activity in the task-relevant sensory dimension (Fig. 7d~~e~~).”

m4- Figure 7E “dimenstion”

It has been corrected in the figure x-axis label

m5- P18, line 3, odd use of “Besides”

P19 L18 :

~~“Besides w~~We observed multi-whisker integration, found to be mostly suppressive “

m6- P20, line 30 “tThese”

The sentence was removed in the updated discussion

Reviewer #2 (Remarks to the Author):

The paper is valuable both in presenting 1) a dataset of calcium imaging population recordings from somatosensory cortex in a difficult, elegant, interesting and well-conceived task and 2) in aiming to investigate behavioral correlates in information encoded in neural activity rather than in activity itself.

While the dataset appears great and the questions of great interest, it seems to me that the paper still falls short of really making a breakthrough in understanding how sensory information is encoded and used for computations in populations of neurons.

M1 - The main conclusions from the abstract that perception depends on the formation of early sensory representations and that stimulus and choice signals (as well as engagement) are partly decoupled does not seem neither strongly supported by the data nor particularly unexpected or novel. In what follows, I give my constructive suggestions on how the analyses and conclusions could be improved and how further analyses of the dataset could relatively easily lead to more interesting advances. I hope that the authors find some of these suggestions to be useful.

We thank the reviewer for its insightful comments. We hope that new analyses and interpretation have significantly improved the quality of our study.

M2 - The whole column optogenetics suggests that the somatosensory area is involved in perception of sensory stimuli, but it is not entirely surprising (because this area has been long implicated in perception) and also does not bear evidence for the sensory populations code proposed and investigated here to be relevant for perception (it will perturb all neurons not only those in the imaged layers, etc). In my view, this seems more of interest as a

compelling and finely resolved optogenetic screening to prove that the investigated recording locations are involved for the task.

Our optogenetic manipulation experiments served purposes. First, previous studies demonstrated maintained tactile-driven behavior despite the total ablation of S1 (Hong et al. 2018), creating controversy regarding the role of S1 in driving tactile-driven behavioral responses. As emphasized by the reviewer, our results provide compelling evidence that S1 is involved in the task.

Secondly, recent attempts to manipulate perception by targeting small neuronal populations (i.e., holographic experiments) failed to show selective and compelling perceptual effects on a trial-by-trial basis, or used neuronal stimulation with possible confounds. Our results set a finer scale of inhibition for driving specific behavior, linking the focal activation of roughly one cortical column to the perception of the feature it encodes. To us, this result holds a particular significance given the columnar organization in many areas of the cerebral cortex.

Third, and perhaps most novel, is the interpretation of our optogenetic experiments in regards to topographic organization of choice versus sensory signals. In a new analysis (Supplementary figure 11), we show that choice-related activity has weak and reversed somatotopy compared to sensory signals. It implies that our optogenetic manipulation affected activity selectively on the sensory axis but not on the choice axis. Therefore, this is an important indication that the animal may preferentially use the sensory axis for its trial-by-trial perceptual readout.

However, we acknowledge that our experiments do not conclusively prove the importance of firing rate coding for perception nor whether multi-whisker suppression influences perception and nor whether the optogenetic effect arises from L2/3 neurons. We aim to address some of these questions in future studies.

To better convey the importance of these results, we largely revised the discussion section on the optogenetic manipulation P19 L28: **“Single column activation drives perception of the sensory feature it encodes**

The results of selective optogenetic manipulation experiments have several implications. Firstly, earlier studies demonstrated the persistence of tactile-driven behavior despite the total ablation of S1 (Hong et al., 2018), sparking controversy over the role of S1 in driving learned, tactile-driven behavioral responses. Besides, most previous studies involving S1 used inactivation to show transient impairments in behavioral performance. However, these impairments could stem from various factors such as creating distracting perceptions for the animal or inducing long-distance perturbations of neuronal homeostasis (off-target effects) (Otchy et al., 2015; Li et al., 2019). By employing somatotopically targeted manipulation to induce a bi-directional shift in behavior, our results provide compelling evidence supporting the involvement of S1 in the task.

Secondly, a contemporary debate revolves around the size of the neuronal population ensemble necessary to elicit learned perceptual-driven behaviors. Recent manipulations, involving the activation of excitatory neurons or stimulus-selective sub-ensembles, have been shown to trigger learned behaviors (Salzman et al., 1990; Huber et al., 2008; Sachidhanandam et al., 2013; Musall et al., 2014; Ceballo et al., 2019; Marshel et al., 2019). However, these manipulations might rely on perceptions from alternative pathways not utilized by the animal under normal sensory circumstances. Our results provide direct evidence that neural activity in a single column drives the behavioral report of the local point or feature it encodes. The columnar scale is particularly

significant, as functional columns are considered the fundamental building blocks of the mammalian brain, prevalent across nearly all brain areas (Mountcastle, 1997). While our study specifically addresses early sensory representation in the somatosensory system, its implications may extend to other perceptual representations in areas displaying cortical maps.

Thirdly, our experiment partially addresses the identity of signals used for the perceptual readout of S1 activity. A recent study (Buetfering et al. 2022) reported that selectively manipulating choice, but not sensory neurons, influences behavior during the session, raising the possibility that the animal relies on choice neurons only for its decision-making. In our task, choice-related activity exhibits weak and reversed somatotopy compared to sensory-related activity (Supplementary Figure 11). Hence, the optogenetic manipulation is aligned with the sensory representations but not the choice representations, which should result in a selective shift in the sensory representation only. Consequently, our results argue against the readout of choice neurons only and suggest that the trial-by-trial readout predominantly relies on the sensory encoding dimension.

From a technical perspective, some limitations of our approaches could be overcome in the future. For instance, we observed inhibition outside of the stimulated area (Figure S4), a challenge that could be mitigated by the use of soma-targeted opsins. In addition, optogenetic stimulation used here is likely to inhibit activity in deeper layers, which could be addressed by conditional genetic expression of the opsin in single layers. To address directly the identity of neurons used for readout however, it would be required to use holographic activation of single target neurons, in column size, functionally defined ensembles or using manipulation of different functional codes (e.g. temporal codes) (Zuo et al. 2015; Musall et al. 2014; O'Connor et al. 2013) which may provide further insights into perception-related circuits and dynamics (Packer et al. 2015; Adesnik and Abdeladim 2021). “

M3 Also, the strength of the claims of almost orthogonality between stimulus and choice signals is difficult to evaluate. The double regression with stimulus and choice will first pick up the main selectivity of the neuron (stimulus or choice) and then the residual selectivity which is not explained by the other. It is thus unclear whether selectivities are really similar (e.g. whether neurons that prefer stimuli associated with lick left also prefer the stimulus associated with lick left).

Double regression is a problem in determining selectivity. To our understanding, this issue arises in case of collinearity between regression variables (Dormann et al 2013). In our analyses, we use matched trial sampling that avoids any collinearity between choice and stimulus (an equal number of left/right choices are included for each pair of whisker stimulation; randomly sampled). Hence, in our analysis of choice related activity (ROC analysis, and regression including explicitly choice), the measured Pearson correlation between the two regression variables yields $r = 0.00$. We hope that this approach alleviates concerns about potential biases in the analysis. We modified the text to highlight our approach better P14 L20:

“We next questioned the representation of stimulus and choice in the entire population of layer 2/3 neurons. Choices 1 and 2 were included explicitly as regression variables to the multivariate linear model of single cell activity. We next questioned to which extent neuronal populations predict the animal's choice. To do so, left and right choices were included directly as regressors to our multivariate linear model of single cell activity (Fig. 6c; see methods). The model was fit with an equal number of right/left choices for each stimulation condition, which eliminates collinearity and enables independent calculation of neuronal selectivity for (1) choice and (2) whisker frequency.

Only trials with motor response at the end of the stimulus were included. The distribution of selective regressors model weights for choice selectivity ($\beta_{C1} - \beta_{C2}$) versus selective weights for whisker frequency selectivity ($\beta_{F1} - \beta_{F2}$) were uncorrelated ($r = -0.03$, $p > 0.1$, LME model test) display only a weak correlation (Pearson's $r = -0.06$).

In addition, it is important to note that, at the level of individual neurons where the regression analysis was conducted, sometimes overlapping coding between choice and sensory weights can be observed. We initially overlooked this aspect, as the two types of coding appears to be uncorrelated at the whole population level. This is addressed in Fig. S9. We updated the text to acknowledge the co-existence of sensory/choice coding at the single neuron level, while sensory/choice coding remains uncorrelated and orthogonal at the population level. P13, L27:

“We selected a matched number of left and right response trials, with small frequency differences ($\Delta F \leq 30$ Hz), to measure target side discriminability (i.e. $F1 > F2$ versus $F1 < F2$; Fig. 6a) and choice response side discriminability (i.e. choice 1 vs choice 2 GP; Fig. 6b). Restricting analysis to sensory ambiguous trials ($\Delta F \leq 30$ Hz) is expected to reveal the highest impact of sensory ‘noise’ on behavioral choice (Britten et al. 1996). Across these trials, 19.9 % of neurons have a different level of activity depending on the choice side; 22.7 % of neurons depending on the target side and 4.7 % have significant change in activity depending on both the choice and target sides (AUROC different from 0.5; comparison versus bootstrapped distribution with a 95% confidence interval, Fig. S9c). Importantly, we found a significant topographic distribution of AUROC choice (Fig. S9a-b; $p < 0.001$, LME model, $n = 3118$ neurons) that was weak and reverted compared to AUROC target.”

However, in regard to the modeling of sensory responses; sensory weights of individual frequencies and their interactions, it is essential to acknowledge that collinearity poses a challenge in estimating the actual weights. The enhancement of the overall regression fit with the inclusion of interaction terms (Figure 4a and Figure S5), along with supplementary analyses conducted in response to additional comments from the reviewer (R2M4, Figure S6, and S7), reassures us that multi-whisker interactions are indeed significant in single neurons and population activity.

Dormann, C. F., Elith, J., Bacher, S., Buchmann, C., Carl, G., Carré, G., ... & Lautenbach, S. (2013). Collinearity: a review of methods to deal with it and a simulation study evaluating their performance. *Ecography*, 36(1), 27-46.

M4 - However, the manuscript presents a nice discovery in terms of sensory coding. It shows the presence of an interaction term between the stimulations of the two whiskers (Eq 1), and the authors argue its important for sensory coding. (This has conceptual similarities with the recent study of Kira. Harvey Nature Communication 2023, that showed population codes with the presence of interaction terms between two types of sensory cues in 2AFC tasks) The authors use this result to argue that it is better to decode stimuli using the difference of activity, which does not seem convincing to me. I think that, instead, it would be really interesting of the authors could show that the presence of the interaction term is key to increase the information encoded in single trials by the population (for example, computing decoding accuracy from real data using decoding models that do or do not have the interaction term).

In the context of our task, multi-whisker (MW) suppression confers two advantages:

(1) It facilitates the generalization of decoding for (F1>F2) across a broader range of stimulation frequency pairs. This effect is illustrated in Figure 4 at the single-neuron level, and we now provide additional details on how decoding performance across the stimulus space depends on both selectivity and MW suppression at the single neuron level (new figure: Fig.S6A; see below).

(2) It allows to integrate the two simultaneous stimuli F1 and F2 in the firing rate of single neurons, so that MW suppressed neurons code for (F1-F2) better than for their preferred single frequency/whisker. We added a description of how decoding performance increases with MW suppression at the single neuron level, independent of selectivity in a new figure: (Fig. S6B; see below).

As suggested by the reviewer, we compared the decoding performance at the population level in two models: one with interaction term and one without (F1x F2 regressor variable were shuffled to remove the term from our decoding model). At the whole population level we saw no change in decoding accuracy across trials. Thus, we compared the models for different subpopulations displaying either MW enhancement, no MW interaction, or MW suppression. We found that the interaction term improved decoding performance in the population of MW suppressed neurons but not the other two. We describe these results in a new figure (Fig. S7), along with the modifications of the results section:

Supplementary figure 6: Multi-whisker (MW) suppression improves decoding performance at the single neuron level,

A, Discrimination of F1>F2 over the stimulus space depends on selectivity and MW suppression across neurons. Discrimination is measured as the area under the curve (AUC) of F1>F2 over the entire stimulus space, as in Fig 4E. The pixel colors represent the average AUC value for neurons in the bin. The neuronal population from all animals is split into bins defined in two dimensions on their MW suppression (BF1xF2; y-axis) and their absolute selectivity index (SI; x-axis). Note the increase of AUC along both x and y axis, suggesting the MW suppression improves discriminability independent of its relationship with whisker selectivity.

B, Decoding performance depends on selectivity and MW suppression. Decoding performance is quantified as the fraction of correctly decoded trials. Decoding was performed with a GLM using logit as the link function and assuming binomial distribution of F1>F2. Binning performed as in A. Please note the increase of AUC along both the x and y axis.

Supplementary figure 7: Impact of multi-whisker (MW) interaction term on (sub)population decoding performance.

A, schematic procedure for decoding with or without interaction. The difference is that trial to trial F1xF2 is shuffled in the model without interaction.

Each neuron activity is fitted trial per trial as described in Fig. 4. The activity of multiple neurons is combined into Rw1, Rw2 and Rw1xw2, (respectively latent representations of F1, F2 and F1xF2). Decoding of F1>F2 is then performed using a GLM with logistic regression using three distinct combinations of latent variables.

B, Comparison of decoding with or without interactions in different subpopulations of neurons (rows) with decoding of different combinations of latent variables (columns). Subpopulations contain 10 % of neurons with most MW enhancement (top), 10 % null MW interaction (middle) or 10 % most MW suppression. The interaction term significantly improves decoding performance for neurons with MW suppression. P-value is computed with Linear mixed effect model across FOV using mice identity as grouping variable.

Modifications of the results section P10 L17:

“Multi-whisker suppression improves and generalize discriminability of neighboring frequency comparison across the stimulus space”

“At the level of the whole population, multi-whisker (MW) integration drives only a subtle sub-linear integration (Fig. S6c-d), likely due to heterogeneity of MW integration across the population. Overall, this regression analysis confirms that single neuron’s response increases with stimulation frequency, and highlights multi-whisker integration as a contributing factor shaping the activity of L2/3 neurons and shows that multi-whisker suppression is an important factor of L2/3 neuronal activity.

Does multi-whisker suppression MW integration help solving the comparison task? To evaluate how this computation affects activity level in the stimulus space, three example neurons were compared: One with multi-whisker MW suppression, one without multi-whisker MW integration and one with multi-whisker supralinear MW responses (neurons 1, 2 and 3 respectively in Fig. 4c-e). The neuron 3 with multiple-whisker enhancement-supralinear MW responses shows no little-difference in activity level on either side of the boundary task (quantified by area under the curve measurement; AUC = 0.65). Therefore, its activity level is not little-informative in regard to the task, and may rather represent a source of noise in our paradigm. On the contrary, multi-whisker suppression-By contrast, MW suppression in neuron 2 leads the gradient of activity in the stimulus space to become perpendicular to the task boundary. Activity level reflects scales better to the difference in stimulation frequency F1-F2 rather than to F1 the stimulation frequency of their the neuron’s preferred whisker (Fig. 4e). Therefore, the output of this neuron encodes the stimuli in a relevant way for the generalization of frequency discrimination, as compared to the neuron without suppression. This comparison reveals how multi-whisker suppression increases the discriminability of neighboring stimulation frequencies.” As a result, this neuron’s output encodes MW stimuli in a manner that is relevant to the generalization of F1-F2 discrimination across the stimulus space (AUC = 0.97; Fig 4e). Across the population, we find that both the AUC across the stimulus space and direct decoding performance increase with the degree of MW suppression (Fig. S6a-b). Besides, the comparison between two models, one with a MW interaction term and one without, indicates that populations of neurons exhibiting MW suppression derive benefits from the MW integration term in decoding F1>F2. Together, these findings reveal how MW suppression improves and generalize the discriminability of neighboring stimulation frequencies.”

Of note, concerning the improvement of decoding using the difference of activity:

Our experiment initially aimed to independently track the representation of each whisker as the behavioral task necessitates a comparison between these two sensory alternatives. We demonstrated that this comparison task can be achieved by the readout of one of the two alternatives. Subtracting the two competing signals improved decoding of F1>F2. But readout is also improved through multi-whisker integration. We emphasized this effect in the discussion.

M5 - Also, it would be interesting to study if this term has an impact on accuracy of behavior. The authors report that their sensory information population code along the “difference between pools” has single trial information that is not diminished in incorrect trials with respect to error trials. The authors argue that errors are not due to failures in sensory coding. One alternative possibility is that the sensory code based on differences of activity (neglecting the non-linear interaction term) is not the sensory code used by the brain, but that the interaction between the two stimuli is instead the one encoding the key information. It would be interesting to test this hypothesis computing whether the population code including the interaction term (or alternatively, neurons that have primarily an interaction encoding) has information that is critically different between correct and error trials. In Kira et

al Nature Communications 2023, the authors did not find information difference between correct and error trials in population codes ignoring interactions between sensory cues, but they instead found that information in correct trials was diminished specifically in the interaction between the cues. Performing a similar analysis on these data would be of interest, and it would provide novel insights in the role of mixed selectivity for sensory coding.

We would like to highlight that we do see a link between behavior and MW integration term in Fig. 7h-h. Specifically, we observe an increase in activity in multi-whisker (MW) suppressed neurons, but not in MW enhanced neurons, with engagement. This effect enhances representations of the signals relevant to the task (Fig7 f-h). Following this comment and others, we emphasized these findings better in the discussion, using the elegant findings from the (Kira et al. 2023) study P22 L1:

“However, we discovered an intriguing and subtle pattern of engagement-related modulation: neurons suppressed by multi-whisker interaction exhibit an increase in stimulus responsiveness with engagement, while neurons enhanced by multi-whisker interaction do not (Fig. 7f). Consequently, the representation of single versus multi-whisker inputs is heightened in the engaged state, potentially favoring a more detailed spatial map of sensory inputs. This gain was independent of the stimulus condition, thus not enhancing discriminability of the stimulus (i.e., akin to an additive gain). However, a downstream reader could potentially benefit from this selective up-modulation. It has been proposed that attention can selectively increase the representation of stimulus features relevant to the task (Maunsell and Treue, 2006). Interestingly, a recent study showed that decoding of neurons integrating the two compared visual stimuli correlates with choice (Kira et al 2023). While we did not find direct evidence in our data that multi-whisker suppressed neurons are preferentially used for read out, the increased responsiveness of these neurons in the engaged state argue for their importance for downstream processing.”

Reference added to the bibliography:

Kira, S., Safaai, H., Morcos, A. S., Panzeri, S., & Harvey, C. D. (2023). A distributed and efficient population code of mixed selectivity neurons for flexible navigation decisions. *Nature Communications*, 14(1), 2121.

In addition, we fully agree with the reviewer's intuition and given their importance for sensory discrimination, we studied further the interaction term in relation to the behavioral choice.

To do so, we tested choice signals in subpopulations of neurons with different MW integration properties. We decoded the target ($F1 > F2$) in the whole population and in subpopulations of either MW suppressed neurons or MW enhanced neurons. As done in our previous response, the interaction term was either included or excluded (e.g. decoding using $Rw1$ w/wo $Rw1 \times w2$) and we compared the results between matched correct and error trials. We found that the interaction term was indeed beneficial to decoding but saw no significant difference between Correct and Error trials (see revision Figure 1 below). This result does hold for subpopulation analysis. A small negative correlation was seen with MW enhanced neurons ($r = -0.10$; promote errors) but only in some animals.

Across all of our analysis (displayed or not), we found that sensory parameters do not predict the choice preference significantly. This result seems at odd with recent work in mice

but is in line with previous studies in the non-human primate notably.

Revision Figure 1: Comparison of sensory information in error versus correct trials with and without multi-whisker interaction.

From left to right: 3 populations were tested. Left: Whole population, all neurons included. Middle: only the 20% neurons with lowest $\beta f1 \times f2$ are included ($<<0$). Right: only 20% neurons with highest $\beta f1 \times f2$ are included ($>>0$)

Top : Choice versus sensory selectivity in single neurons, r is the pearson correlation between the two, tested with a linear mixed effect model in 11 FOV with animal as grouping variable (7 mice), all $p > 0.1$.

Middle: Fraction $F1 > F2$ decoded correctly without introducing the interaction term in the decoder i.e. decoding of (Rw1), (Rw2) and (Rw1-Rw2) latent representation alone.

Bottom: Fraction $F1 > F2$ decoded correctly with the interaction term in the decoder. Rw1xw2 were integrated optimally to Rw1 and Rw2 to maximize the fit to (F1-F2). We note an improvement in decoding performance when transitioning from the model without interaction to the model with interaction. This improvement seems to be maximal in MW suppressed neurons but also present in MW enhanced population.

m1 Minor. Page 11, line 24..”is best for solving the task”. The use of “best” suggests that the author demonstrate that the difference in activity is optimal, whereas the author seem only to compare it to two alternatives. I suggest either proving optimality or write “better”.

We fully agree and updated the text as suggested P12 L21 :

“is **better** best for solving the task “

Reviewer #3 (Remarks to the Author):

Studies have shown that neurons in the barrel cortex respond to sensory stimuli as well as the animal's choice in behavioral tasks. Yet, it has remained unclear if these aspects are encoded in the same population of cells. Gardères et al. clearly show that different neuronal populations code for choice and input differentiation in S1 in this elegant study. .

Towards this aim, Gardères et al. developed a novel 2-AFC task in which mice were trained to compare stimulation frequencies applied to two adjacent vibrissae and reported by licking right or left based on the frequency difference of the two stimuli. Using calcium imaging of individual neurons, optogenetic silencing, and advanced analysis using linear regression models, they found that neurons encode the properties of sensory input, animal choice, and engagement in the task. These varied substantially but are orthogonal to each other. That led them to conclude that perceptual variability does not originate from sensory encoding neurons but from state or choice fluctuations in downstream areas. The experimental design is exceptional, and the methods and analysis are superb.

Accordingly, the delayed 2-AFC task, which they designed, effectively separates the animal's actions from the stimulation time. This is crucial for testing the hypothesis and is more reliable than the go/no-go paradigm. The analysis supporting the hypothesis is presented in figures 6-7. Both figures are based on a regression model that effectively supports their conclusions.

M1 - However, the manuscript contains numerous grammatical/spelling errors and formatting issues that detract from its overall quality and readability. These problems should be addressed to improve the text's clarity and coherence. In addition, while I strongly support this paper for publication, it was difficult to understand both the experimental details and the analysis in particular. The explanations for the methods used to analyze the data depicted in Figures 4 to 7 are limited, making it difficult to comprehend the information presented in each panel without making assumptions. The methods section is unfortunately scarce in explaining the analysis and did not help here. The authors should add more explanations both when describing the results and also in the methods section. I suggest the authors test their changes by sending the paper to colleagues and students and see if the figures and text are clear.

We thank the reviewer for highlighting the lack of clarity in the text. We made corrections following suggestions in the reviewer's comments. Besides, we added a section to the methods relating to figure 4 - 7: "Decoding and AUROC analysis" and improved the method section "Linear regressions and sensory evidence / choice side prediction" (Please find these additions and modifications below). Finally, we have introduced panel C in Figure S5 as a graphical summary of the procedure employed in population analysis in Figures 5 to 7:

C

Supplementary Figure 5c

C, Graphical summary of population analysis used in Fig 4-7.

In the initial step, linear regression is independently applied to each neuron (R_n representing the activity of neuron n) to ascertain the sensory features it represents. This regression process identifies the weights $\beta_{F1,n}$, $\beta_{F2,n}$, $\beta_{F1xF2,n}$, which each of the three sensory features (F1, F2 and F1xF2) drive the activity of neuron n . $\beta_{0,n}$ represents the baseline activity in the model, and ϵ represents the non-fitted residuals.

In the subsequent step, the activities of multiple neurons are pooled using the same sensory weights that drive them. These weighted pools embody the latent representations of stimulus features encoded by the populations of neurons: $Rw1$ represents the population response to the stimulation frequency of whisker 1 (F1), $Rw2$ corresponds to the population response to the stimulation frequency of whisker 2 (F2), and $Rw1xw2$ carries the population response to the cross-whisker interaction.

These latent representations are then leveraged for various analyses, including decoding, exemplified here as neurometric analysis, or for the trial-by-trial representational geometry of sensory versus choice signals, as depicted in Figures 6 and 7. Additional regression variables, namely choices (C1 and C2) or engagement (Eng), are incorporated explicitly in the models presented in Figure 6 and 7 respectively, to generate additional latent representations of choices and engagement (not represented here, described in the methods).

Modification applied to the methods applicable to Fig 5-7, P28 L42:

“” ~~Linear regressions and sensory evidence / choice side prediction reconstruction~~

Linear regression on single neuron firing rate

We used a linear model to regress the recorded spiking activity of single neurons against the series of different stimulation applied to the two whiskers. Models were fitted separately for each neuron. Spiking activity was averaged in all trials over a 1-second epochs during both intertrial intervals (1 sec before stimulus onset) and during whisker stimulation (0 to 1 sec after stimulus onset). All epochs of activity were concatenated into a single vector Y that was regressed against the square root of the two whisker frequencies (F_1 and F_2) and the product of these two regressors (F_{1x2}).

We used a linear model to regress the recorded spiking activity of single neurons against the stimulation frequency parameters applied to the two whiskers (F1, F2, F1x2). The same design model was fitted separately for each neuron. The fitted data points represent spiking activity averaged over 1-second intervals during baseline epochs (1 second before the stimulus onset) and whisker stimulation epochs (from 0 to 1 second after the stimulus onset; the whole stimulus duration) in individual trials. These averaged activities were then concatenated into a single vector, denoted as 'R_n' for the neuron n. R_n was then regressed against the square root of the two whisker frequencies (F1 and F2) and the product of these two regressors (F1x2) for each trial. F1, F2 and F1x2 were set to zeros for baseline epochs:

$$Y = \beta_0 + \beta_1 F_1 + \beta_2 F_2 + \beta_{F1x2} F_1 x_2 + \epsilon$$

$$R_n = \beta_0 + \beta_{F1} \sqrt{F1} + \beta_{F2} \sqrt{F2} + \beta_{F1x2} \sqrt{F1x2} + \epsilon$$

β_0 represents baseline activity (the intercept), other β s represent weights for the regression variables and ϵ represents residuals. In the versions of the model that explicitly included choice (Figure 6) or engagement (Figure 7), we used only delayed response or no response trials, so that activity related to the animal's face movement did not interfere with the analysis. **These models for engagement and choice were fitted with a 10-fold cross validation. For the choice model, we used two binary regressor separately for left and right choice (respectively C₁ and C₂, set to NaN during baseline epochs) set to 0 or 1 depending on the choice side and included disengaged trials in the ongoing trials:**

$$R_n = \beta_0 + \beta_{F1} \sqrt{F1} + \beta_{F2} \sqrt{F2} + \beta_{F1x2} \sqrt{F1x2} + \beta_{c1} C_1 + \beta_{c2} C_2 + \epsilon$$

To model engagement, we included a single regressor (**Engaged**) that was set to 0 in non-engaged trials and 1 in engaged trials, during both baseline and stimulation. **In this model**, non-engaged trials were defined as the animal not licking any spout and occurring in the last third of the daily sessions. Engaged trials were defined as trials with a response from the animal and occurring in the first half of the session:

$$R_n = \beta_0 + \beta_{F1} \sqrt{F1} + \beta_{F2} \sqrt{F2} + \beta_{F1x2} \sqrt{F1x2} + \beta_{eng} Engaged + \epsilon$$

~~Before regression was actually applied, we normalized the activity of single neurons by dividing it by its average value.~~ **Before regression was fitted, we first normalized the activity of single neurons within each session and Z-scored the concatenated vector of activity.** This normalization enables the comparison of weights across cells as they would be proportional to the variance explained by the regressor. With the aim of comparing the weights between regressors, we also z-scored the regressors values, establishing thus a direct proportionality between regressors weights and activity level. The model fitting was performed ~~using matlab fitlm function that uses a~~ **with a** QR decomposition algorithm. Later, reconstruction of sensory evidence is computed separately for each regressor as the sum of weighted neuronal activity divided by the **absolute** sum of weights. Pooled activity summaries (Rw1, Rw2, ...) are computed from independent pools of neurons, by choosing only cells tuned to the summarized signals more than to the other alternative (e.g., ~~pooling only neurons preferring w1 to compute Rw1, only neurons with B1 > B2 and B1 > B1x2 were pooled~~). The result of this computation is an estimate of F1's neuronal representation strength. In figure 6 and 7, **representations of choice included all neurons, as it yielded significantly better choice prediction performance.** Another decoding strategy would be to reconstruct the frequency via maximum likelihood estimation using Bayesian inference (Runyan et al. 2017). However, we chose linear pooling for its simplicity as it provides an intuitive way of summarizing activity over pools of neurons and does not rely on any assumption of independence between neuron's activity.

Decoding and AUROC analysis

To construct the neurometric function, we decoded the stimulus category (i.e., $F1 > F2$ versus $F1 < F2$) by applying thresholding to weighted population averages (either $Rw1$, $Rw2$, or $Rw1 - Rw2$; Fig. 5B-C). For example, if $(Rw1 - Rw2) > \text{threshold}$, the decoded response is $F1 > F2$, and vice versa. An optimal threshold was determined through AUROC analysis using the MATLAB `perfcurve` function. In decoding of the stimulus category across different behaviors, as shown in Fig. 5D-E, we utilized the same threshold across behaviors. This threshold was computed using sets of trials equalized for stimulus conditions (e.g., an equal number of left and right choices in the same stimulus conditions). For the decoding of choice and target side in Figures 6 and 7, the fitting of neuronal weights, average pooling, computation of the threshold, and decoding were all performed within a 10-fold cross-validation framework.

AUROC analysis, commonly employed in binary classification problems, serves to estimate classification accuracy. While it is conventionally set to yield values greater than 0.5, in Fig. 6A-B, we opted to use values < 0.5 when the firing rate is higher for $F1 > F2$ trials or Choice 1 trials, and values > 0.5 when the firing rate/activity is higher for $F1 < F2$ trials or Choice 2 trials. This method allows us to keep track of the preferred whisker (Fig. 6A) and choice (Fig. 6B). “

General remarks

m1 - Fig. 2F - Add color so it will be possible to distinguish between the lines.

Fig. 2G - Present the complementary stimulation frequencies. In other words, if data exists, present also 0/10; 0/45; 0/90.

In Fig. 2F, we have added color to the lines to enhance their distinguishability. In Fig. 2G, the information about complementary stimulation frequencies is in the form of dashed lines, labeled as 'Adjacent population'. We have incorporated additional information into Fig. 2G label axis to improve the visibility of this information.

m2 - Fig. 5C - Why is $Rw2$ missing in the panel? Please explain if there is a reason for that. Is it similar to $Rw1$? If not, explain.

We aimed to avoid overcrowding the figure, considering the similarity between $Rw2$ and $Rw1$. Figure 5C has been revised to incorporate both graphics and statistical comparisons for $Rw1$ and $Rw2$. The neurometric function parameters for the two are not significantly different (slope and lapse rate, $p > 0.1$). We have updated the text to enhance clarity on this matter. P12 L17:

“When including all sampled neurons, neurometric categorization of $Rw1$ and $Rw2$ shows more reliable discrimination than the behavior across trials (lower “lapse rate” for $Rw1$ and $Rw2$ neurometric fits; $p < 0.05$, Wilcoxon signed rank test) but have similar sensitivity (slopes compared to the psychometric function; $p > 0.1$). The integrated reading of these two signals by subtraction $Rw1 - Rw2$ is ~~best~~ **better** solving the task. ($p < 0.001$ slope and lapse rate compared to behavioral performance; $p < 0.001$ and $p = 0.005$ respectively slope and lapse rate compared to $Rw1$; LME post hoc test, Fig. 5c).”

m3 - P2 L36 - “two adjacent whiskers on the same row of the whisker pad (e.g., B1/C1)”. - B1 and C1 are in different rows. Should it be: two adjacent whiskers on the same column of the whisker pad?

Absolutely, we modified the text as suggested P2 L39.

“two adjacent whiskers on the same ~~row~~ column of the whisker pad (e.g. B1/C1). “

m4 - Add to methods how intrinsic imaging was performed, which system, and how the barrel borders were delineated.

Add a scale bar size to Fig. 2A - intrinsic imaging

We have added the missing scale bar legend in Fig. 2A and included a new paragraph in the methods section to explain how Intrinsic Optical Imaging was performed P23 L42:

“ **Intrinsic optical imaging.**

Prior to two-photon calcium imaging, intrinsic optical imaging (IOI) was performed to identify the single whisker representation in neocortex using a 50 mm tandem lens system. Imaging was conducted at 30 Hz with a 12-bit CCD camera such that the field of view (FOV) spanned the entire cortical region covered by the window. The imaged region was continuously illuminated with a red-light emitting diode (630 nm wavelength). For stimulation, a single whisker was inserted into a glass capillary mounted on a piezoelectric element powered by a piezo-controller (MTD693B, Thorlabs, USA). The whisker was moved rostro-caudally for 6 s with a 10 Hz square wave pulse (amplitude: ~ 0.5 mm). This stimulation protocol was repeated at least 10 times with inter-trial intervals (ITI) of 20 s. The change in absorbed light was computed as:

$$\Delta r = r_{\text{stim}} - r_0,$$

with r_0 being the average image in the 5 s window prior to stimulation and r_{stim} the average image in the 5 s window prior to the end of stimulation. Δr revealed discrete areas of functional activity corresponding to the single whisker representation (see example in Fig. 2A, areas are delineated as a change in absorbed light > 0.15%, following an isotropic gaussian smoothing with $\sigma = 38\mu\text{m}$). A green-light emitting diode (530 nm) was used to visualize the vasculature pattern in order to match IOI images to subsequent two-photon imaging and optogenetic experiments. Cortical columns centers, were defined manually from two photon neuropil response when available (in Fig. 2 and Fig. 3), and from IOI otherwise”

m5 - P11 L29-31 - (Fig. 5D) - “If the behavioral read out depends directly on the relative sub-populations firing rate, we should observe a left/right shift in the neurometric functions drawn from trials with left/right choice”

A clarification is needed on how the shift would look like for the two curves. Could such a shift be simulated and shown as an additional panel/line?

We have included the suggested panel. Below is the revised text with an enhanced explanation of the concepts, P12 L27:

“Sensory encoding was then compared across different behavioral outcomes. If the behavioral read-out depends ~~directly on the relative sub-populations firing rate, we should observe a left/right shift in the neurometric functions drawn from trials with left/right choice~~ on the difference of activity between Rw_1 and Rw_2 relative sub-populations firing rate, we expect that animal respond left as the sensory evidence $Rw_1 - Rw_2 > 0$ and right as the sensory evidence $Rw_1 - Rw_2 < 0$. This bias is

expected to be maximal in trials with high sensory uncertainty (i.e. with $|F1-F2|$ low) leading to a left/right shift in the neurometric functions drawn from trials with left/right choice (Fig. 5d, expected).“

Regarding the neurometric analysis in engaged/disengaged trials, we have also made adjustments to Fig. 5e to illustrate the potential expected outcomes, and we have modified the corresponding text accordingly, P12 L38:

“The same analysis was repeated to compare trials with or without behavioral responses (Fig. 5e). One might expect a decline in sensory discrimination when the animal is not engaged in the task, visible as a decrease in the slope of the neurometric (Fig. 5e, expected)”

m6 - P11 L29-37 - A more in-depth explanation and clarification of the graphs are needed to understand the concepts better.

Please see our response to the comment above.

m7 - P12 L22 - More information on the AUROC is needed in the Methods. Not all readers are familiar with this method. In particular, it is not well explained in the context of the plots in panels 6A and B.

We added information on AUROC in the method. We now refer to the “AUROC and decoding” method section P13 L23:

“We then sought to describe whether and how choice is represented in wS1. Area under the receiver operating curve (AUROC, see methods) is a standard metric to test whether a neuron’s activity is correlated to the choice of the subject (termed choice probability or CP (Britten et al. 1993; Crapse and Basso 2015). AUROC measures how well the firing rate of a neuron discriminates between two conditions.”

The corresponding paragraph in the methods P30 L6:

“AUROC analysis, commonly employed in binary classification problems, serves to estimate classification accuracy. While it is conventionally set to yield values greater than 0.5, in Fig. 6A-B, we opted to use values <0.5 when the firing rate is higher for $F1 > F2$ trials or Choice 1 trials, and values >0.5 when the firing rate/activity is higher for $F1 < F2$ trials or Choice 2 trials. This method allows us to keep track of the preferred whisker (Fig. 6A) and choice (Fig. 6B).“

m8 - P 13 - second paragraph - Please explain why one choice factor (Rc1) predicts the choice much better than the other choice factor. Is it experimental bias?

This is an intriguing observation! We observe that Rc1 decoding exhibits its peak information during the response epoch, coinciding with the period when the animal is licking (decoding of

choice peaks around ~1.2 seconds post-stimulus onset, corresponding to maximum licking occurrences). During this epoch, we also observe a complete overlap in information between Rc1 and Rc1-Rc2 (Fig 6E). We posit that this may be a characteristic of motor reafference predominantly lateralized to contralateral licking.

In contrast, during the whisker stimulation epoch, both choices are represented in complementarity (as quantified by significantly superior decoding of Rc1-Rc2 compared to Rc1 only or Rc2 only; Fig S10; please also see our response to the reviewer question below **R3m12**). In our view, the significant complementarity of choice information in Rc1 and Rc2 within the stimulus presentation epoch is indicative of choice-related activity while information about contralateral choice only (as observed during licking responses) might represent more motor-related (or motor preparation related) activity. Therefore, we interpret the dominance of C1 coding during the stimulus epochs as possible contamination by contralateral, preparatory motor activity.

m9 - Fig. 6F - Write $f1 > f2$ and $f2 < f1$ on top of each panel to speed up understanding of the graphs.

Figure was modified as suggested.

m10 - Fig. 6F. Can the two point sets be plotted in different distinguishable colors?

We replaced points by right and left pointing arrows, in different colors.

m11 - Fig. 6 - Explain why $\Delta F > 30$ Hz was selected.

We calculated choice information in trials with 'ambiguous/difficult' sensory evidence, driven by two considerations (Britten et al., 1996). First, these trials were deemed to exhibit the highest choice-related activity. The rationale behind this is that, when sensory evidence is limited, trial-to-trial noise fluctuations exert a more pronounced impact on perception, influencing the subject's choice. Secondly, these 'ambiguous' trials present a more even distribution of choices on either side. The chosen criterion value (30Hz) was a consensus value for our dataset, ensuring a sufficient number of trials for each field of view and each animal (at least 25 trials per choice side). It is worth noting that we conducted a similar analysis with different criteria (both higher and lower) and obtained comparable results (as detailed in response to the question R3m13 below). We have included a clarification in the main text to enhance understanding, P13 L27:

"We selected a matched number of left and right response trials, with small frequency differences ($\Delta F \leq 30$ Hz), to measure target side discriminability (Fig. 6a) and response side discriminability (CP; Fig. 6b). Restricting analysis to sensory ambiguous trials $\Delta F \leq 30$ Hz trials is expected to reveal the highest impact of sensory 'noise' on behavioral choice (Britten et al., 1996). Across these trials, ... "

m12 - Fig. 6E bottom panel - why do the curves for Rc1 and Rc1-RC2 overlap? Any meaning for this or is it expected?

We acknowledge that in Fig. 6E, decoding choices at a fine temporal resolution (frame by frame) shows roughly an overlap between Rc1 and Rc1-Rc2. The overlap in Fig 6E is a consequence of noisy decoding of Rc1-Rc2 on a frame by frame basis and dominant coding of choice 1 (more neurons prefer choice 1, as shown in the AUROC analysis and discussed above in response to R3m8).

However Rc1-Rc2 is slightly but consistently above Rc1 decoding performance (we modified transparency to illustrate that difference better). This might be due to preferred representation of preparatory motor activity to the contralateral side.

Importantly, the comparison throughout the entire epochs (average over 30 frames) reveals complementary choice information. Combined, Rc1-Rc2 choice decoding accuracy reach 0.69 correct (± 0.018 ; mean \pm s.e.m.) which is significantly superior to decoding choice accuracy from isolated Rc1 and Rc2 (CPs of 0.63 ± 0.011 and 0.59 ± 0.018 , respectively; $p < 0.001$ for either compared to Rc1-Rc2; LME posthoc test, data shown in Fig. S7). The difference we observed is the result we expect, indicating that either choice information is encoded separately.

m13 - What are the results for trials with a big frequency differences ($\Delta F > 30$ Hz)?

We computed AUROC for sensory and choice discrimination in trials with $\Delta F > 30$ Hz, and found the choice information to be very similar as when trials selected are $\Delta F < 30$ Hz (see table below). Proportion of significant AUROC for sensory discrimination increases substantially. This is largely expected because trials with $\Delta F > 30$ Hz are more easily discriminated than trials with $\Delta F < 30$ Hz (as seen, for instance, in neurometric functions). In either case, we observe uncorrelated coding of C1 or C2 in W1 and W2 tuned populations:

	trials with $\Delta F \leq 30$ Hz	trials with $\Delta F > 30$ Hz
% C1 selective neurons	14.9 %	14.0 %
% C2 selective neurons	5.0 %	5.0 %
% W1 selective neurons	9.0 %	19.4 %
% W2 selective neurons	13.6 %	24.4 %
average choice CP (W1 pop.)	0.481	0.487
average choice CP (W2 pop.)	0.486	0.481

For $\Delta F > 30$ Hz; $n = 3119$ neurons (some FOV don't have 25 left and 25 right choice trials when equalizing stimulation conditions).

*Selective neuron means it has a CP significantly different from a distribution with shuffled choice/ stimuli. neurons are selective and sorted to their preferred conditions (stimuli /choice)

m14 - Fig. 7 G-H - How come that similar labels have a three orders of magnitude difference? Perhaps it is obvious to the authors, but this discrepancy is an example of the limited explanations in the methods and results.

We were showing: the difference of activity in F (single neurons) and G (population), and ratio of evoked activity (normalized to baseline) in H. We agree that it is confusing and homogenization of the axis is needed. We modified the analysis to have the same y-axis metric and label in Fig 7 F-G; in each of these, the Y-axis now measures the ratio of activity in Engaged/ Disengaged trials. Statistical values don't change significance except Rw1 compared to Rw1xw2 for which $p = 0.067$ before and $p = 0.004$ now. However, please note that the gain amplitude in the figure 7H is now lower, because the effect of engagement previously described arises from both a decrease of activity in baseline and an increase in stimulus response. We changed the text accordingly.

Updated panels:

Figure legend is modified accordingly:

“F, Engagement related modulation as a function of neuronal sensory weights. Computed as the ratio of activity in engaged/disengaged trials with matching stimulus conditions. For each plotted line, the entire neuronal population is split into 10 equal bins of beta weights (β_{F1} , β_{F2} or β_{F1xF2}). Engagement related modulation as a function of sensory weights. Computed as the difference in activity in engaged minus disengaged trials (matching stimulus conditions). The neuronal population is split into 10 deciles of beta weights (either β_{F1} , β_{F2} or β_{F1xF2}).”

G, Engagement related modulation of pooled response R_{W1} , R_{W2} and R_{W1xW2} over time. Computed as the difference in activity in engaged minus disengaged trials (matching stimulus conditions). Each population is an independent pool of different neurons selective for $W1$, $W2$ or combination of the two. $n = 865, 916, 1925$ neurons respectively. Selectivity is defined as having the highest Beta weights among β_{F1} , β_{F2} or β_{F1xF2} . Shaded error bars represent s.e.m. across $n = 11$ FOVs.

H, Quantification of the engagement related modulation of activity during the stimulus period in (G). Statistical comparison across $n = 11$ FOVs. LME post hoc test. Quantification of the engagement related modulation as the ratio $R(\text{engaged})/R(\text{disengaged})$. statistical comparison across $n = 11$ FOVs. Wilcoxon rank signed test.”

As well as the corresponding result section P17 L7:

“Accordingly, The two pooled average R_{W1} and R_{W2} show increased response amplitude in the engaged state, starting immediately at stimulus onset, while the third pooled average of multi-whisker tuned neurons R_{W1xW2} shows the opposite effect (Fig. 7g-h). Engagement modulation of single whiskers pools R_{W1} and R_{W2} is significantly different from R_{W1xW2} during the stimulus period. When considering stimulus evoked activity (i.e., normalized by baseline activity), engagement modulation amount of +22% and +16 % for R_{W1} and R_{W2} and -7 % for R_{W1xW2} . a

limited decreased response amplitude (fraction of FR engaged/disengaged corresponding to 22% and 14% increase in activity for Rw1 and Rw2 versus a 6% decrease for Rw1xw2; median across n = 11 FOVs. Fig. 7g-h). Engagement thus promotes selectively **representations of activity related to single whiskers versus representation of the multi-whisker combination.** ~~supralinear activity.~~"

m15 - Fig. S1B - Show the real data (along with the fits).

We modified the graph to display data points along with the fit in S1B and corrected the caption. We reworded the caption:

Legend has been updated:

"B, Psychometric fit of the proportion correct choice $P(\text{correct})$ for different categories of stimuli. Vertical bar represents sensitivity threshold **at half of the fitted "lapse rate" performance** $P(\text{correct}) = 0.65$. **Data point size represents the relative amount of trials (normalized within each 3 conditions).** Discrimination ~~is-seems~~ easiest when a single stimulus is presented (**black**). When two stimuli are presented on W1 and W2 simultaneously, discrimination ~~is-seems~~ harder when the sum of the two frequency is high ($F1 + F2 > 90\text{Hz}$, **magenta versus blue**)."

Reviewer #4 (Remarks to the Author):

We would like to thank the reviewer for thoroughly reading and commenting on our article; We could address the majority of the comments, and hope that this significantly improved the quality of the article.

Gardères et al. develop a novel 2AFC whisker discrimination task and use it to study choice predictive activity in mouse whisker S1. Specifically, they stimulate two whiskers, with one whisker receiving a higher frequency and the other a lower frequency. The animal must indicate the whisker that was given the higher frequency by licking one of two lickports. First, they show how different populations respond to each whisker, reflecting the somatotopy of the area. Next, they perform an elegant barrel-specific silencing experiment that demonstrates the whisker-specific contribution of the stimulus they provide, imaging with 2P simultaneously and thereby demonstrating how the neural perturbation relates to the activity changes they induce. They then show that cross-whisker suppression is a key contributor to neural activity. Next, they show that it is fairly easy to decode the frequency difference from the sensory stimulus, with only 6 neurons needed to reach animal performance; larger pools of neurons exceeded animal performance. Finally, they show that sensory activity does not predict choice, and that engagement does not explain whisker-evoked activity differences.

Overall, the data are of high quality, the analyses are well done, and I found the paper fairly clearly written. However, there are issues - both major and minor - that need to be addressed before publication.

Major:

M1. The task design was probably my favorite thing in this paper. This is a great behavior and will be useful for many questions. I really commend the authors on this - training mice for 3+ months is no small thing, and developing a true 2AFC task in mouse vS1 is really impressive.

We thank the reviewer for the encouraging comment. We are also very excited about the possibilities offered by this 2AFC task, and we are currently continuing to expand on it.

M2. It is unclear to me what trials are used in the decoding analyses. In Fig. S2D, the authors very commendably segregate trials based on whether orofacial (nose) movements could be discerned. It sounds, however, as though perhaps most analyses (Figs. 4 onward) that purport to look at choice look at both trials with and without early movements; only trials with pre-sensory movements are excluded, from what S2D indicates. The major analyses in the paper that are looking at choice should only use trials without early movement, as any movement will contaminate the result with refference. I *think* this is being correctly done based on Fig 6.E, but it was very difficult to find a clear and concise statement to this effect; please make it very very clear that you ONLY use the trials with no early movement. If this is not the case, analyses should be restricted to these trials only.

Trials with early movement were included in Fig 1-3 and 5 to gain statistical power in analysis of sensory coding. We did exclude any early movement when analyzing choices related activity in Figures 6 and 7, as indicated in Fig. 6e. We thank the reviewer for pointing out the lack of clarity on this important matter. To address this problem, we systematically documented selection of trials used for each statistical analysis in the statistical summary **table 2** (Please see our response to the question below **R4M3**).

The only analysis of choice using with early reaction time are Fig. 5 D-E. The reason is that neurometric analysis requires a large amount of trials. When selecting late trials only, the goodness of neurometric fits is degraded for some FOVs with less trials. Besides, we wanted a maximal statistical power to confirm the absence of significant changes between behavioral outcomes. It seems reasonable to us to include these trials in the plots as long as we describe it clearly and provide additional statistical tests with conservative trial selection (added in the text and in Supplementary table 2).

As it is one of the main results of our study, we did the comparison of neurometric sensitivity and bias across behavior in many different conditions not shown here: with early trials or only late trials, with/without selecting FOV for which at least 200 trials are available (to avoid bad fits), or considering only early activity before any movement of the animal (i.e., first 150 ms of sensory evoked activity), etc. We always consistently find that sensory activity is reliable and no significant difference in sensitivity or bias of the neurometric fit across behavior.

Please see below the result of analysis with/without early response trials. Note the widening of confidence intervals for the slope (but not change in the estimate of effect size) when excluding early trials.

Data	N points used for the test	Grouping variable	Statistical test	Mean/Effect size [CI 95%]	P Value	Note on trials inclusion
Fig. 5D Neurometric. bias Left vs Right choice	11 FOV	7 mice	LME bias ~ item +(1 animal)	(A) 3.1 [-2.6; 8.8] (B) -1.7 [-12.8; 9.4]	(A) 0.26 (B) 0.75	Animal engaged; F1+F2 =90; sessions with >65% correct (A) Trials with RT > 0 sec (B) Trials with RT > 1 sec
Fig. 5D Neurom. slope Left vs Right choice	11 FOV	7 mice	LME slope~ item +(1 animal)	(A) -0.11 [-0.49; 0.27] (B) -1.7 [-4.9; 1.5]	(A) 0.55 (B) 0.28	same as above; (A) Trials with RT > 0 sec (B) Trials with RT > 1 sec
Fig 5E Neurom. bias Engaged vs Disengaged	11 FOV	7 mice	LME bias ~ item +(1 animal)	(A) 4.0 [-2.8; 10.8] (B) 2.9 [-6.0; 11.8]	(A) 0.24 (B) 0.51	same as above; (A) Trials with RT > 0 sec (B) Trials with RT > 1 sec; At least 50 trials per FOV.
Fig 5E Neurom. slope Engaged vs Disengaged	11 FOV	7 mice	LME slope~ item +(1 animal)	-0.43 [-0.94; 0.07] -1.5 [-3.9; 0.9]	(A) 0.088 (B) 0.200	same as above; (A) Trials with RT > 0 sec (B) Trials with RT > 1 sec

Therefore, we would like to keep the same analysis, although making clearer to the readers which trials were used. We modified the text as follow P12 L31:

“We observed no difference in the bias of the neurometric functions of Rw_1 , Rw_2 or $Rw_1 - Rw_2$, whether the animal responded left or right ($p > 0.05$, $n = 11$ FOVs from 7 animals; Wilcoxon sign-rank test; Fig. 5d, Fig. S6). **Excluding all trials with impulsive movements, (i.e., reaction time <1s), to avoid measuring activity related to licks or uninstructed facial movements, we obtained the same results (all $p > 0.05$, $n = 11$ FOVs from 7 animals, see LME results in table 2).**”

And for neurometric analysis of the engaged/disengaged behavior P12 L40:

“Again, the behavioral outcome has no impact on the sensory encoding for frequency categorization (lapse rate, slope, and bias, $p > 0.05$, $n = 11$; decoding in engaged versus disengaged trials; Fig. 5e, Fig. S6).

Excluding all trials with impulsive movements confirms the absence of significant change in bias or sensitivity ($p > 0.05$, see LME results in table 2). “

M3. There are two major statistical issues I see with in this paper:

First, in several analyses, the authors aggregate across all neurons in the dataset (e.g., Fig. 2E, 4B, many Fig. 6 and 7 panels). This is highly problematic. First, there is clearly a large variability in neuron count and this skews the results toward specific animals. Second, within-animal one can reasonably expect things to co-vary, and independence assumptions are thus very much violated relative cross-animal. It would be ideal to adopt per-animal summary statistics and then do any tests on those statistics. For instance, in 6A, show an example animal's choice propability histogram and then do statistics across the medians from each animal.

Second, the authors segregate animals into 11 FOVs and use these as the level of analysis. Sometimes this may be appropriate, but in many cases, within animal behavior and neural activity will vary less than across animals, and using FOVs as the independent observation stops being appropriate. At minimum, this needs to be discussed, but I would recommend against this in most cases used here; use the animal as the level of analysis.

We agree with the reviewer's comment of statistical inadequateness for (1) aggregated neuronal population analysis, and for (2) statistical analysis at the FOV level. We modified all these statistical testing as follows to include individual mouse information:

- 1) In all population analysis, we replaced standard Pearson's correlations testing with a linear mixed effect model that treats mice as a random effect and the variable of interest as a fixed effect. It means that the data (e.g., in Fig 2E, neurons' selectivity index) fit was allowed to vary within each animal for an independent intercept; and we report the test for the fixed effect (e.g., in Fig 2E does selectivity depends on the neuronal type IN or EN?). In summary, this method takes into account dependance of data collected in the same animals while maintaining high statistical power (Yu et al, 2022)

We prefer this approach over the testing of each animal independently as our statistical power might be too low given the number of mice. We observed a single statistical reversal: in Fig 6C the weak correlation between sensory and choice discrimination weights was significant ($r = 0.06$, $p < 0.001$); it is now non-significant using the LME model ($E = -0.003$ (Zscore), $p = 0.88$)

- 2) In all statistical tests using FOV as the n value, we replaced analysis with linear mixed effect models analysis, including animal as random effect, such that non-independence of samples collected for the same animals are taken into account.

In summary, we believe that the new statistical approach is more robust thanks to reviewers' comments. The novel analysis allows us to maintain all our initial conclusions. In accordance with nature communication guidelines, and following the reviewer comments we added a paragraph detailing the statistical approach undertaken, and provided a full statistical table

detailing: n values, p-values, grouping variables (N mice), LME used, effect size (Z-scored) and their confidence interval. We hope these updates alleviate concerns about statistical weakness across the study.

Table 2 now provided in the supplementary information:

Data	N points used for the test	Grouping variable	Statistical test	Mean/Effect size [CI 95%]	P Value	Note on trials inclusion
Fig. 2 Selectivity index in IN versus EN	267 INs vs 974 ENs	3 mice	LME 'SI~ Ntype +(1 animal)'	-0.053 [-0.11; 0.00]	0.051	Trial with single stim. discrimination (i.e., F1/F2 = 90/0 or 0/90)
Fig. 2G 'Princip. pop.' Testing 3 levels of PW stim. frequencies	18 Columns	5 mice	LME 'Resp.~ PW+(1 animal)'	0.66 [0.46; 0.87]	<0.001	Trials with RT > 0 sec; with the detailed PW/AW pairs (available in 9 FOV)
Fig. 2G 'Princip. pop.' Testing 3 levels of AW stim. frequencies	18 Columns	5 mice	LME	-0.18 [-0.44; 0.09]	0.21	as above
Fig. 2G 'Adj. pop.' Testing 3 levels of AW stim. frequencies	18 Columns	5 mice	LME	0.82 [0.67; 0.98]	<0.001	as above
Fig. 2G 'Adj. pop.' Testing 3 levels of PW stim. frequencies	18 Columns	5 mice	LME	0.33 [0.07; 0.59]	0.009	as above
Fig. 3I (A) W1 blocking effect on Right choice (B) W2 blocking effect on Right choice	15412 trials	8 mice	GLME 'Right choice~ F1 +F2+W1 block +W2 Block+(1 animal)'	(A) 0.09 [0.06; 0.13] (B) -0.06 [-0.09; -0.03]	<0.001 <0.001	Opto/sham paired trials with W1 or W2 blocking, all F1/F2 condition; sessions with >65% correct (without opto)
Fig. 3G W1&W2 blocking effect on Correct response	9298 trials	10 mice	GLME 'Correct choice~ F1-F2 + F1+F2 + Block+(1 animal)'	-0.35 [-0.44; -0.27]	<0.001	Opto/sham paired trials with W1&W2 blocking; all F1/F2 condition; sessions with >65% correct (without opto)
Fig. 4B β_{F1} vs β_{F2}	3706 neurons	7 mice	LME ' β_{F1} ~ β_{F2} +(1 animal)'	0.38 [0.31; 0.45]	<0.001	Trials with RT > 0 sec;
Fig. 4B β_{F21} vs β_{F1xF2}	3706 neurons	7 mice	LME ' β_{F1} ~ β_{F1xF2} +(1 animal)'	-0.61 [-0.69; -0.53]	<0.001	as above
(not shown) β_{F2} vs β_{F1xF2}	3706 neurons	7 mice	LME ' β_{F2} ~ β_{F1xF2} +(1 animal)'	-0.60 [-0.66; -0.53]	<0.001	as above

Fig. 5C neuron. slope (A) Rw1-Rw2 vs Rw1 (B) Rw1-Rw2 vs bhv (C) Rw1 vs Rw2 (D) Rw1 vs bhv	11 FOV	7 mice	LME with post hoc comparison $\text{slope} \sim \text{item}^3 + (1 \text{animal})$	(A) 0.71 ¹ (B) 0.75 (C) 0.08 (D) 0.04	<0.001 <0.001 0.69 0.85	Trials with RT > 0 sec; animal engaged; F1+F2 =90; sessions with >65% correct
Fig. 5C neuron. lapse (A) Rw1-Rw2 vs Rw1 (B) Rw1-Rw2 vs bhv (C) Rw1 vs Rw2 (D) Rw1 vs bhv	11 FOV	7 mice	LME with post hoc comparison $\text{lapse} \sim \text{item}^3 + (1 \text{animal})$	(A) -0.05 ¹ (B) -0.20 (C) 0.004 (D) -0.15	0.005 <0.001 0.81 <0.001	Trials with RT > 0 sec; animal engaged; F1+F2 =90; sessions with >65% correct
Fig. 5D Neuron. bias Left vs Right choice	11 FOV	7 mice	LME $\text{bias} \sim \text{item}^3 + (1 \text{animal})$	(A) 3.1 [-2.6; 8.8] (B) -1.7 [-12.8; 9.4]	(A) 0.26 (B) 0.75	Animal engaged; F1+F2 =90; sessions with >65% correct (A) Trials with RT > 0 sec (B) Trials with RT > 1 sec
Fig. 5D Neuron. slope Left vs Right choice	11 FOV	7 mice	LME $\text{slope} \sim \text{item}^3 + (1 \text{animal})$	(A) -0.11 [-0.49; 0.27] (B) -1.7 [-4.9; 1.5]	(A) 0.55 (B) 0.28	as above (A) Trials with RT > 0 sec (B) Trials with RT > 1 sec
Fig 5E Neuron. bias Engaged vs Disengaged	11 FOV	7 mice	LME $\text{bias} \sim \text{item}^3 + (1 \text{animal})$	(A) 4.0 [-2.8; 10.8] (B) 2.9 [-6.0; 11.8]	(A) 0.24 (B) 0.51	Animal engaged; F1+F2 =90; sessions with >65% correct; (A) Trials with RT > 0 sec (B) Trials with RT > 1 sec; At least 50 trials per FOV.
Fig 5E Neuron. slope Engaged vs Disengaged	11 FOV	7 mice	LME $\text{slope} \sim \text{item}^3 + (1 \text{animal})$	-0.43 [-0.94; 0.07] -1.5 [-3.9; 0.9]	(A) 0.088 (B) 0.200	same as above (A) Trials with RT > 0 sec (B) Trials with RT > 1 sec
Fig. 6A & B change in AUROC W1 vs W2 pop. (A) AUROC target side (B) AUROC choice side	(A) 3118 neurons	5 mice (9 FOV)	LME $\text{AUROC} \sim \text{item}^3 + (1 \text{animal})$	(A) 0.067 [0.047; 0.087] (B) 0.004 [-0.014; 0.023]	<0.001 0.64	Trials with RT > 1sec; stim. condition paired for left and right choice; F1-F2 <30Hz; sessions with >65% correct. At least 50 trials per FOV.
Fig. 6C correlation b/w choice and sensory discrimination (A) all neurons (B) 20% "best sensory" neurons (C) 20% "best choice" neurons (D) 20% most sensory- choice interaction neurons 2 (E) 20% most multi-whisker suppression neurons	(A) 3118 neurons (B) 311 neurons (C) 311 neurons (D) 311 neurons (E) 311 neurons	5 mice (9 FOV)	LME $(\beta_{C1} - \beta_{C2}) \sim (\beta_{F1} - \beta_{F2}) + (1 \text{animal})$	(A) -0.003 [-0.04; 0.03] (B) 0.05 [-0.06; 0.16] (C) 0.025 [-0.07; 0.12] (D) 0.70 [0.63; 0.78] (E) -0.09 [-0.21; 0.02] 0.62531 0.78368	0.88 0.37 0.62 <0.001 0.09	Trials with RT > 1sec; stim. condition paired for left and right choice; F1-F2 <30Hz; sessions with >65% correct; At least 50 trials per FOV.
Fig. S7 pop CPs (choice information) (A) Rc1 (B) Rc2 (C) Rc1-Rc2	11 FOV	7 mice	LME $\text{Cp} \sim (\text{Rc}) + (1 \text{animal})$	(A) 0.63 [0.60; 0.65] (B) 0.59 [0.56; 0.62] (C) 0.69 [0.66; 0.71]	<0.001 <0.001 <0.001	Trials with RT > 1sec; stim. condition paired for left and right choice; sessions with >65% correct; At least 50 trials per FOV.

Fig. S7 pop CPs multiple comparison (choice information) (A) Rc1 vs Rc2 (B) Rc1 vs (Rc1-RC) (C) Rc2 vs (Rc1-RC)	11 FOV	7 mice	LME with post hoc comparison $C_p \sim (Rc) + (1 animal)$	(A) 0.037 ¹ (B) -0.061 (C) -0.098	<0.008 (overlapping CI) <0.001 <0.001	same as above
Fig. 7C correlation b/w engagement modulation and sensory discrimination all neurons	3706 neurons	7 mice (11 FOV)	LME $Beng \sim (BF1-BF2) + (1 animal)$	-0.03 [-0.06; 0.00]	0.08	Trials with RT > 1sec; stim. condition paired for engaged and disengaged trials
Fig. 7G correlation b/w engagement weights and sensory weights (A) β_{F1} (B) β_{F2} (C) $\beta_{F1 \times F2}$	3706 neurons	7 mice (11 FOV)	LME $Beng \sim \beta_{F1} + \beta_{F2} + \beta_{F1 \times F2} + (1 animal)$	(A) -0.01 [-0.12; 0.10] (B) -0.06 [-0.16; 0.04] (C) -0.60 [-0.77; -0.42]	0.85 0.24 < 0.001	Trials with RT > 1sec; stim. condition paired for engaged and disengaged trials NOTE: $\beta_{F1 \times F2}$ alone capture the dependence
Fig. 7H engagement related change in evoked firing rate (A) Rw1 (B) Rw2 (C) Rw1xw2	11 FOV	7 mice	LME $Beng \sim item + (1 animal)$	(A) 0.21 [0.11; 0.30] (B) 0.24 [0.14; 0.33] (C) 0.03 [-0.07; 0.12]	<0.001 <0.001 0.55	Trials with RT > 1sec; stim. condition paired for engaged and disengaged trials
Fig. 7H Multiple comparison (A) Rw1 vs Rw2 (B) Rw1 vs Rw1xw2 (C) Rw2 vs Rw1xw2	11 FOV	7 mice	$Engmod \sim 1 + Item + (1 animal)$	(A) -0.03 ¹ (B) 0.18 (C) 0.21	0.61 0.004 (overlapping CI) <0.001	Trials with RT > 1sec; stim. condition paired for engaged and disengaged trials

Table 2; summary statistics: Statistical tests are grouped figure wise and/or topically, when the test was repeated with variations, variations are indicated by letters (A), (B), ... These use the same model but e.g. a different set of trials or a different population of neurons. ¹There is no confidence interval for post-hoc comparison using LME, but we noted when confidence intervals of effect estimate were overlapping. ²“sensory-choice interaction” is computed as the absolute value $|(\beta_{F1} - \beta_{F2}) * (\beta_{C1} - \beta_{C2})|$, i.e. the product of correct sensory and choice discrimination weights. ³“item” relates to a Boolean variable use to set the comparison in the first column: e.g. Rw1 vs Rw2 is coded as 0 and 1s.

Paragraph added to the methods P30 L12:

“Statistical methods

For statistical comparisons between animals, standard non-parametric methods were used: Mann-Whitney U test for independent groups, Wilcoxon signed-rank test for paired samples, and Tuckey’s post-hoc tests for Friedman’s test when multiple comparisons of paired samples were needed. When dealing with multiple observations per mouse (comparing FOV or individual neurons), which could introduce dependencies among samples, we employed linear mixed-effects models (LME). LME mitigates these potential confounding effects, treating individual mice as random effects (Yu et al. 2022). We have included a comprehensive table detailing the sample sizes (n), p-values, the number of mice (N), the statistical model employed, grouping variable, the magnitude of the effect sizes, and their corresponding confidence intervals (see Table 2 in the supplementary information). All statistical tests were two-sided, and an alpha value of 0.05 was used to determine significance.”

Reference added to the manuscript:

Yu, Z., Guindani, M., Grieco, S. F., Chen, L., Holmes, T. C., & Xu, X. (2022). Beyond t test and ANOVA: applications of mixed-effects models for more rigorous statistical analysis in neuroscience research. *Neuron*, 110(1), 21-35.

M4. I found many of the population analyses problematic. Fig. 4A histogram shows that, as with many vS1 studies, a small minority of cells is actually doing the interesting work. If you look at all the cells in all your analyses (Figs. 4B, 6, 7), you will generally be recording noise and masking the few neurons that are doing interesting things. This likely accounts, for example, for the "orthogonality" results in Fig. 6F and Fig 7E. This analysis would be far more compelling if it were restricted to the most discriminative/sensory responsive neurons. Do these have a choice signal? Do they show modulation by engagement? As it stands, I don't find F6 and 7 particularly compelling. An easy fix would be to shift these analyses towards single neurons, and focus on the most interesting ones.

I would recommend 1) Fig. 6: subselect the 5-10% of neurons that have the highest/lowest CP ; repeat the analyses with only these cells, and you may see something more interesting; 2) Fig. 7: subselect neurons with highest sensory selectivity > how much are these neurons impacted by engagement?

1) Choice analysis (Fig 6): Several analyses in our draft point to the absence of relation between sensory and choice coding: (1) the absence of differences between neurometric curves in correct and error behavior (Fig. 5D). (2) The similarity of CPs in population coding for W1 and population coding for W2 (across degrees of single neuron selectivity for one or the other whisker) (Fig. 6A). (3) The absence of correlation in the population coding for choice versus sensory preference (Fig. 6C). (4) The orthogonality of trial to trial fluctuations between populations coding for choice and "sensory noise" (Fig. 6E). However, the reviewer raises the important concern that using the entire population may obscure the selectivity of the most selective neurons.

Therefore, we investigated whether the robustness of our primary findings was influenced by the selectivity or activity levels of neurons, by subsampling neuronal population based on their tuning to whisker frequencies or choice.

Comparative analyses were conducted across the following groups: (A) the entire neuronal population; (B) the top 20% of neurons with the highest selectivity for individual whisker movements; (C) the top 20% of neurons with the highest selectivity for either choice; (D) the top 20% of neurons exhibiting the greatest suppression of multi-whisker input; (E) the top 20% of neurons displaying the highest selectivity at the intersection of correct choice and sensory input (i.e. most selective to c1 and f1 or c2 and f2).

For groups (A) through (D), we observed no correlation between the neurons' selectivity for choice and stimulus ($r \sim 0$; row 1). Decoding performance for Rw1, Rw2 or Rw1-Rw2, remained consistent across correct and incorrect trials (except a small change for Rw1 in (B), row 2). This resulted in comparable neurometric functions for trials with left versus right responses (row 3). The orthogonality of sensory and choice representations persisted across these populations (row 4). These results indicate that sensory discrimination was reliable, irrespective of the choice, provided an optimal sensory decoder was used. Besides, the encoding of sensory versus choice representations remained orthogonal within the population, regardless of the neurons' activity level or sensory selectivity.

In contrast, for the neuronal population (E), a substantial correlation between choice and stimulus selectivity was detected ($r = 0.64$), which aligns with the selection criteria for these neurons. Significant disparities were found between correct and incorrect trials when analyzing Rw1, Rw2, and the difference Rw1-Rw2, with a clear left/right bias emerging

based on the animal's response direction. In this scenario, the representations of choice and stimulus became highly correlated, rendering the sensory and choice representations indistinguishable. Thus, selecting neurons with mixed choice/sensory selectivity could account for the behavioral choice of the animal. The temporal dynamics of sensory and choice encoding within this group paralleled those observed in the overall population (not shown, similar to Fig. 6e). This subset of neurons exemplifies how collinear coding can influence the representation of sensory information.

While the intersection coding of sensory and choice variables exists at the single-neuron level, the representation at the whole population level is nonspecific, with choice being unrelated to sensory coding. Our results do not preclude that the animal could have a selective readout of the population as in e.g. E. However the assumption of a selective readout is not supported by our data, given the equilibrated and orthogonal representation of choices and sensory alternatives. Instead, orthogonality suggests different origins of choice versus sensory signals and argues against a feedforward readout of choice neurons. In an additional analysis of the spatial distribution of choice coding, we found it to be weak and reversed compared to sensory coding. Thus, the results of our optogenetic manipulation strengthen the view that the animal uses preferentially the sensory cues in S1 rather than choice signals.

We added the subpopulation analysis in Supplementary Figure 11, illustrated the sensory choice intersection at the single neuron level in Supplementary Figure 10c-d and the somatotopy of choice in Supplementary figure 10a-b intersection. We introduced these additional results in the text and developed the interpretation of the optogenetic experiments in the discussion.

Figure S11: Analysis of sensory choice coding in sub-populations of neurons with strong selectivity.

We compared sensory/choice coding of the entire population (A) to 4 subpopulations that code best for sensory selectivity (B), choice selectivity (C), multi-whisker suppression (D), or intersection of sensory features and choice in correct trials (E). For the (E) population only, sensory and choice become non-orthogonal. Selection criteria are indicated on top of the column. 20% with highest criterion were included in analysis (B) to (E).

From top row to bottom: (1) Single neurons' selectivity for choice as a function of selectivity for whisker, quantified as $\beta_{C1}-\beta_{C2}$ and $\beta_{F1}-\beta_{F2}$ respectively. (2) Sensory decoding performance ($F1 > F2$) from Rw1, Rw2 and Rw1-Rw2 compared between correct versus error trials in matching stimulus condition (with $|\Delta F| < 30$). LME model analysis * indicates $p < 0.05$; ** indicates $p < 0.01$ and *** indicates $p < 0.001$. (3) neurometric performance in left and right choice trials. (4) trial to trial representation of different trial categories (color coded) spanning the four possible combinations of target and choice side.

2) Engagement modulation (Fig 7):

In Fig. 7 F, we show that engagement leads to increased evoked response for neurons having strong response to single whiskers (large β_{F1} and/or β_{F2} weights) or being suppressed by multi-whisker interaction (negative $\beta_{F1 \times F2}$). It corresponds to a change in firing rate of $\sim 20\%$ on average for the MW suppressed neurons. In the first version of the manuscript, labels were misleading as we were displaying different metrics for the same effect in Fig 7F-H. As suggested by reviewer comment **R3m14**, we homogenized the labels.

The updated panels:

Following other comments, we emphasized this result in the discussion and in the corresponding text P17 L3:

“Weights for single whisker frequencies (β_{F1} or β_{F2}) are positively correlated to engagement related gain, whereas the while-weights for supra-linear whisker interaction (β_{F1xF2}) of the two whiskers are negatively correlated with engagement modulation. Using a GLME model, we found that among the three sensory weights, β_{F1xF2} alone captures engagement-related modulation related to sensory selectivity (Supplementary Table 2). Accordingly, The two pooled average R_{W1} and R_{W2} show increased response amplitude in the engaged state, starting immediately at stimulus onset, while the third pooled average of multi-whisker tuned neurons R_{W1xW2} shows the opposite effect (Fig. 7g-h). Engagement modulation of single whiskers pools R_{W1} and R_{W2} is significantly different from R_{W1xW2} during the stimulus period (Fig. 7g-h). When considering stimulus evoked activity (i.e. normalized by baseline activity), engagement modulation amount of +22% and +16 % for R_{W1} and R_{W2} and -7 % for R_{W1xW2} . Engagement thus promotes selectively representations of single whiskers versus representation of the multi-whisker combination. shows a limited decreased response amplitude (fraction of FR engaged/disengaged corresponding to 22% and 14% increase in activity for R_{W1} and R_{W2} versus a 6% decrease for R_{W1xW2} ; median across $n = 11$ FOVs. Fig. 7g-h). Engagement thus promote selectively activity related to single whisker versus multi-whisker supralinear activity“

M5. Along this line, while you do cite Buetfering 2022, I think this is really undercited as it is incredibly relevant to the present work and seems to show the opposite -- that a few neurons do have choice modulation and that manipulating them perturbs behavior. At minimum this should be more explicitly discussed.

It is indeed an extremely relevant study. First, they also describe non overlapping population coding for the sensory stimulus and for upcoming decisions, largely in line with our results. In addition, Buetfering et al, showed a shift in Error/Correct behavior when manipulating neuronal population coding for choice but not sensory information. However, if we are not mistaken, they show a change in the discrimination performance from session to session and not from trial to trial (Fig. 7 of their study). This could have several interpretations, including that of reinforcement learning within the session. Also the intriguing finding that manipulation of choice but not sensory evidence biases behavior is at odds with the notion that mice use available sensory evidence in S1.

Our manipulation induces shifts in choice on trial by trial level and is more aligned with classical view, of sensory evidence being passed to higher order decision areas. However we could not test specifically manipulation of choice signal, only lever the somatotopy to manipulate sensory evidence. In our opinion, choice signals might be used as reinforcement

to help selection of the appropriate sensory evidence from trial to trial rather than guiding behavior within the trial. Further experiments will be needed to find the specific neurons that govern decisional readout on a single trial basis.

Overall, we largely updated our discussion and included the Buetfering study at two levels: First in the discussion section about our optogenetic results P20 L6:

“Thirdly, our experiment partially addresses the identity of signals used for the perceptual readout of S1 activity. A recent study (Buetfering et al. 2022) discovered that selectively manipulating choice, but not sensory neurons, influences behavior during the session, raising the possibility that the animal relies on choice neurons only for its decision-making. In our task, choice-related activity exhibits weak and reversed somatotopy compared to sensory-related activity (Supplementary Figure 11). Hence, the optogenetic manipulation is aligned with the sensory representations but not the choice representations, which should result in a selective shift in the sensory representation only. Consequently, our results argue against the readout of choice neurons only and suggest that the trial-by-trial readout predominantly relies on the sensory encoding axis. “

And second in the section concerning sensory/choice representation P21 L21:

“Choice-related activity does not appear to causally drive behavior, but may fulfill a different function. In a recent experiment, the manipulation of choice neurons’ ensembles in S1 biased bidirectionally performance throughout the session but not between stimulated versus non-stimulated trials (Buetfering et al. 2022). This finding is compatible with a hypothetical function of choice neurons in reinforcing sensory-motor association and guiding subsequent decisions. In this regard, orthogonal coding provides the advantage of not corrupting sensory information, which could allow for flexible use of sensory information, should the task or behavioral requirement change.”

P21 L6:

“Instead, our results add to the growing evidence that choice coding lies in dimensions orthogonal to that of sensory representation, enabling multiplexing of information in different neuronal subspaces (Zhao et al. 2020; Buetfering et al. 2022). “

M6. 5. Behavioral example images are missing and need to be added. Especially with regards to the analysis in Figure S2, which to me is a linchpin of this study since early movement will contaminate the choice signal and put the conclusions of the paper into question.

We now show a series of example trials representative of the dataset supplementary video 1. In our view, exclusion of trial was conservative, for instance small startles at the onset of the stimulus were counted as movement and excluded from choice analysis.

Supplementary video 1; Nose, whisker and tongue tracking for 6 representative trials
Video added

Minor:

m1. In general there are several things that are unclear / difficult to discern: how many neurons and, if applicable, how many from what category ; depth at which you imaged; how many days post injection ; what whiskers were used ; how many neurons per FOV and how many FOVs per animal - I would recommend a table to make this clear, as there are so many different things being done.

As suggested, we added a table to detail identity of the animals used and acquisition of all two photon datasets (**Table 1.** Summary of all two-photon imaging field of views):

Field of view/ Dataset	Animal/ stimulated whiskers	depth of recording / days of imaging	virus/ indicator	numbers of neurons (type)	FOV size / pixels / imaging rate
sp1 /2pbehavior	RRED1 / C1-D1	~150µm / 56-75 (1 day /2)	jRGECO1a	342 (syn ¹)	ML ² : 738; AP: 605 / 30Hz
sp2 /2pbehavior	RRED1 / C1-D1	~180µm / 59-76 (1 day /2)	jRGECO1a	299 (syn)	ML: 590; AP: 484/ 30Hz
sp3 /2pbehavior	RRED2 / C1-D1	~125µm / 58-76 (1 day /2)	jRGECO1a	470 (syn)	ML: 738; AP: 605 / 30Hz
sp4 /2pbehavior	RRED2 / C1-D1	~160µm / 60-77 (1 day /2)	jRGECO1a	282 (syn)	ML: 590; AP: 484/ 30Hz
sp5 /2pbehavior	RRED5 / C1-D1	~120µm / 52-72 (1 day /2)	jRGECO1a	390 (syn)	ML: 738; AP: 605 / 30Hz
sp6 /2pbehavior	RRED5 / C1-D1	~150µm / 55-72 (1 day /2)	jRGECO1a	289 (syn)	ML: 590; AP: 484/ 30Hz
sp7 /2pbehavior	RRED5 / C1-D1	~180µm / 113- 130 (1 day /2)	jRGECO1a	291 (syn)	ML: 738; AP: 605 / 30Hz
sp8 /2pbehavior	RRED6 / B1-C1	~140µm / 58-61	jRGECO1a	328 (syn)	ML: 738; AP: 605 / 30Hz
sp9 /2pbehavior	RRED7 / B1-C1	~130µm / 58-89 (1 day /2)	jRGECO1a	371 (syn)	ML: 738; AP: 605 / 30Hz
sp10 /2pbehavior	RRED8 / B1-C1	~130µm / 57-88 (1 day /2)	jRGECO1a	385 (syn)	ML: 738; AP: 605 / 30Hz
sp11 /2pbehavior	GADRED1 / C1- D1	~130µm / 74-88 (4 days total)	jRGECO1a; EGFP	259 (syn)	ML: 738; AP: 605 / 30Hz
sp1 /2p Inhib	GADRED1 / C1- D1	~100-140µm -180 / 20 -25	jRGECO1a; EGFP	300 (syn) of which 49 (GAD+) 183 (GAD-)	ML: 590; AP: 484/ 10Hz (3 planes)
sp2 /2p Inhib	GADRED1 / C1- D1	~120-160µm -200 / 21 -26	jRGECO1a; EGFP	359 (syn) of which 58 (GAD+) 235 (GAD-)	ML: 590; AP: 484/ 10Hz (3 planes)
sp3 /2p Inhib	GADRED1 / C1- D1	~130µm / 74-88 (4 days total)	jRGECO1a; EGFP	258 (syn) of which 57(GAD+) 138(GAD-)	ML: 738; AP: 605 / 30Hz
sp4 /2p Inhib	GADRED2 / C1- D1	~100-140µm -180 / 23 -27	jRGECO1a; EGFP	211(syn) of which 14(GAD+) 258(GAD-)	ML: 590; AP: 484/ 10Hz (3 planes)
sp5 /2p Inhib	GADRED2 / C1- D1	~120-160µm -200 / 23 -27	jRGECO1a; EGFP	194 (syn) of which 16 (GAD+) 139 (GAD-)	ML: 590; AP: 484/ 10Hz (3 planes)
sp6 /2p Inhib	VGATEGFP13 / B1-C1	~120µm / 21	jRGECO1a; EGFP	248 (syn) of which 73(VGAT+) 121(VGAT-)	ML: 590; AP: 484/ 10Hz (3 planes)

sp1/Opto2p(dual)	VGATCHRCAMP 1 / C1-D1	~110-130-150- 170-190µm / 121- 123	GCaMP6s hChr2-EYFP	2448 (syn)	ML: 948; AP: 784/ 4.6 Hz (5 planes)
sp2/Opto2p(selective)	VGATCHRCAMP 1/C1-D1	~110-130-150- 170-190µm / 124- 132	GCaMP6s hChr2-EYFP	1592 (syn)	ML: 948; AP: 784/ 4.6 Hz (5 planes)
sp3/Opto2p(dual)	VGATCHRCAMP 2/B1-C1	~110-130-150- 170-190µm/ 20- 24	GCaMP6s hChr2-EYFP	1562 (syn)	ML: 948; AP: 784/ 4.6 Hz (5 planes)
sp4/Opto2p(selective)	VGATCHRCAMP 2/B1-C1	~110-130-150- 170-190µm / 26- 30	GCaMP6s hChr2-EYFP	1595 (syn)	ML: 948; AP: 784/ 4.6 Hz (5 planes)
sp5/Opto2p(dual & selective)	VGATCHRCAMP 4 / C1-D1	~110-130-150- 170-190µm / 20- 24	GCaMP6s hChr2-tdtomato	2459 (syn)	ML: 1185; AP: 980/ 4.6 Hz (5 planes)

Table 1. Summary of all 2p imaging FOVs.

¹ syn stands for human synapsin

²ML stands for medio-lateral axis, AP for antero-posterior

m2. Please refrain from calling yes/no tasks, like the typical MT dot kinetogram, 2AFC. I am not sure where this error originated, but it is clearly pervasive in the literature, with luminaries like Britten and Salzman committing it. 2AFC has a clear relationship to yes/no tasks mathematically in terms of psychometric performance because 2AFC tasks have two stimuli whereas yes/no tasks have one. Both have two response contingencies. More information in the most basic case means $\sqrt{2}$ improvement in certain performance metrics for a 2AFC task vs. equivalent yes/no task. Romo's task is indeed 2AFC (2IFC, to be specific, but the math is the same), as is the present task. If you have two dot kinetograms and you ask the monkey to indicate the one with, e.g., more motion then you have yourself a 2AFC task. One dot field = yes/no, which is what Salzman and the rest of that literature use.

We corrected the text as follow:

P2 L3 :

"Pioneering studies addressed that issue using ~~two alternative forced choice~~ **two choice discrimination** tasks (~~2-AFC~~), where primates had to judge the motion direction of a cloud of points (Britten et al. 1992; Salzman et al. 1990; Britten et al. 1996)."

and defined later 2AFC: P2L20 :

" It has thus been proposed that more complex designs, such as **two alternative forced choice tasks (2AFC) featuring** ~~with a~~ a delayed response are needed to disentangle choice from these other sources of modulation"

m3. I disliked the characterization of go/nogo as problematic due to the early onset of movement. That is not a fundamental problem of go/nogo designs; it is a problem of designs without a delay period and exclusion of trials where such movement occurs. Please adjust the relevant paragraph.

We rephrased as follow to insist on the need for a delay P2 L15:

“First, many studies in the rodent model used a go/nogo paradigm and studied the detection of a stimulus close to the detection threshold. The widespread cortical activity related to onset of facial movement (Musall et al. 2019; Stringer et al. 2019) – including in sensory areas– renders ambiguous the signals associated with **choice if sensory integration and motor preparation are temporally overlapping, go trials**. These approaches are also relatively vulnerable to animal biases and changes in motivation. It has thus been proposed that more complex designs, such as **two alternative forced choice tasks (2AFC) featuring with a delayed response** are needed to disentangle choice from these other sources of modulation (Zagha et al. 2022).”

m4. Behavior: this task is a potentially important contribution to the field; please quantify other basic psychometric parameters (lapse rate, bias). Perhaps a vertical dotted line at $F1-F2=0$ and %F1 higher=50 would be of use in 1E.

All psychometric parameters were added to supplementary figure 1 and legend updated:

D

“ D, Parameters of the psychometric fit for 18 animals; number of trials per animal : average 6371; ranging from 1958 to 13260.”

m5. Fig 1C: can you use color to make it possible to see individual animals? as it is there are too many to discriminate the individual animal lines; propagate this to E

We find it hard to read colors of individual animals; we would prefer to keep individual lines grey. Here is a colored version for 18 mice:

m6. Fig 2C: I can understand why you opted to have time be vertical, but that is, at least for me, rather odd. Please see if you can't rotate it so time is horizontal as it is almost everywhere else.

Following your suggestion, Figure 2C has been rotated, and the rest of the figure reorganized.

m7. Depth of recordings is never reported ; please report it

We recorded in depth ranging from 110 um to 200 um below pia, this information was added in the methods section "Two-photon calcium imaging data acquisition" P2 L26:

"...in order to cover two barrels of whiskers within the same FOV. We recorded in depth ranging from 110 to 200 μm (see Supplementary table 1 for detailed parameters of recording in each FOV). ..."

m8. Choice probability should be computed s.t. the more common occurrence is values > 0.5. Reported values of 0.486 are equivalent to .514, it is an arbitrary choice how one computes it (also, please don't do the pooling neuron aggregation -- that is bad for reasons outlined above).

In Fig 6A, we provide a distribution of individual neurons CPs and the dependence of CPs on sensory selectivity. We also added a graphical display of individual neurons AUROC significance in the new figure S9. We added a method section on AUROC following reviewers 3 request. In this section, we explain why we would like to keep CPs of opposite signs for discriminability of the target side (Fig 6A) and choice side (Fig. 6B).

P30 L6:

" Decoding and AUROC analysis

...

AUROC analysis, commonly employed in binary classification problems, serves to estimate classification accuracy. While it is conventionally set to yield values greater than 0.5, in Fig. 6A-B, we opted to use values <0.5 when the firing rate is higher for F1>F2 trials or choice 1 trials, and values >0.5 when the firing rate/activity is higher for F1<F2 trials or choice 2 trials. This method allows us to keep track of the preferred whisker (Fig. 6A) and choice (Fig. 6B)."

m9. You use chronic AAV-based expression of jRGECO and GCaMP but training takes nearly 3 months. This makes me worry about expression-based cytopathology. Please provide some data showing that your cells are not filled or some other control demonstrating that cells are still healthy that long after infection.

The jRGECO construct used in this study contains a nuclear export sequence that prevents nuclear filling. In practice, we observed no nuclear fillings, but some signs of fluorophore small (subcellular size fluorescent aggregates) aggregation, slowly appearing over a long time-span (generally > 15 weeks) while stimulus responsiveness remains consistent even after long-term expression of 1 year in some cases. Please find below some example FOVs between 5 and 48 weeks of time. Our imaging was carried out before 14 weeks (with the exception of 1 FOV). The good preservation of tissues along with the high sensitivity for whisker stimulation are the reasons why we have chosen jRGECO for this study.

Concerning Gcamp6s we do indeed have more concerns related to toxicity and nuclear filling. We have experienced visible nuclear filling and reduced number of stimulus responsive neurons after 3 to 4 weeks of expression. Most of our imaging was below 4

weeks with this indicator (for the optogenetic experiments), although not all. For this experiment, we only included comparisons for neurons that are stimulus responsive, and the conditions are paired (same neurons, same recording sessions). We therefore believe that the conclusions of our analyses remain valid.

m10. Z-scoring of dFF traces does not make sense. Proper dFF calculation will appropriately normalize so that 0 is 'correct' (equivalent to mean subtraction in proper z-score). As you did quite nicely with the videography analysis, you should instead compute a noise statistic (e.g., variance of the dFF signal during periods of inactivity) and divide dFF by that to get an 'noise-adjusted' dFF. That is all that needs to be done to a properly computed dFF trace.

Z-scoring in the conventional sense will be especially bad as distributions of dFF in active neurons are long-tailed, and the mean is actually not the correct additive normalizer - F0 is, and presumably this is already incorporated in the dFF measurement.

We absolutely agree with the reviewer's comment. We are not z-scoring the dFF but computing separately the dFF and Z-score from raw fluorescence. We used the Z-score to identify periods of putative inactivity, which are used to compute the baseline fluorescence

F0 in the formula $dFF = F/F_0$. In fact, the terminology ‘noise adjusted dFF’ seems appropriate for what we are actually doing. Our description of the pre-processing method was crucially lacking details and remained inaccurate, so we largely updated it as indicated below P27 L44:

“In 11 FOV from 7 animals with jRGECO1a, we carried detailed analysis over complete population statistics. Raw images from calcium imaging were first motion corrected in X and Y dimensions using the Suite2p motion correction module. Movies were motion corrected and ROIs were then delineated using the routine from Suite2p and (Pachitariu et al. 2016) ROIs were manually curated and every visible cell with an event rate > 0.005 Hz was included. The mean ROI fluorescence and neuropil fluorescence was corrected by local neuropil, and then by subtracting a 30 s rolling 10th percentile (to exclude slow drifts). Neuropil alpha subtraction factor was computed independently for each ROI as the slope of a linear regression of neuronal fluorescence against neuropil fluorescence (Runyan et al. 2017). The regression was carried out only outside of activity epochs as defined by the fluorescence being below the 16th percentile of the entire time series (putative period of non-activity). A local neuropil fluorescence was defined as an annulus surrounding the ROI, with an external diameter of 3 times the ROI diameter, and an internal diameter of 1.5 times the ROI diameter(padding). A “noise adjusted Df/F_0 ” was then computed from the mean ROI fluorescence and neuropil fluorescence using a custom algorithm. In brief, this algorithm runs 4 processing stages iteratively until finding a stable estimate of the neuropil correction factor α : (1) correction of neuropil and slow trends,(2) percentile-based Z-scoring, (3) estimation of putative periods of non-activity (E_{na}) from the Z-score, (4) linear regression to estimate the neuropil correction factor α . α is initialized to 0.5 (other values return the same results, but converge more slowly).

(1) Neuropil is subtracted the with factor α : $F_1(t) = F(t) - \alpha * F_p(t)$. We then subtract a 10th percentile in a running window of 30 s. $F_2(t) = F_1(t) - F_{10th}(t)$.

(2) A percentile-based Z-scoring is performed that scales to the noise level. Based on the assumption that activity-related calcium transients are upward, we predict that lower percentiles of the fluorescence distribution are less “affected” by activity, and closer to the underlying noise statistics. Following an assumed gaussian distribution of the residual noise, we define the variance $\sigma = pr(16) - pr(2.3)$, and the average baseline fluorescence $\mu_f = pr(2.3) + 2 \times \sigma$; with $pr(2.3)$ and $pr(16)$ being the 2.3rd and 16th percentile of the fluorescence distribution.

$$Z = (F_2(t) - \mu_f) / \sigma$$

(3) Z is smoothed with a boxcar filter of width 3 and an index of auto-correlation of order 1 is calculated (AR1). Increase in AR1 track with high sensitivity consistent increases of fluorescence (i.e. increase in fluorescence consistent in more than 2-3 frames).

$AR1(t) = Z(t) * Z(t+1)$. Values of AR1 above an arbitrary threshold of 0.5 are considered periods of possible activity and epochs of at least 5 frames with AR1 below 0.5 are considered possible epochs of non-activity (E_{NA}).

(4) factor α was calculated on by least squares regression of $F(t_{ENA}) = \alpha * F_p(t_{ENA})$, multiple time in 30s long time windows (to avoid regressing slow drifts) and only including timepoints in E_{NA} . Updated α is finally computed as the average across all 30 sec windows with at least 5 data points.

steps (1) to (4) were repeated until, α varies by less than 0.025, or iterated at least 10 steps.

More than 95% neuropil correction factors computed this way were between 0.5 and 0.85, values below or above this range were respectively set to these boundaries. We then applied steps (1) and (2) to compute the final percentile-based Z-score, and a noise adjusted Df/F_0 . Df/F_0 was computed as $F_2(t)/F_0$ with F_0 being the highest value between (a) the median of F_{10th} or (b) the median of

neuropil fluorescence $F_p(t)$ was taken. Finally ~~single ROI fluorescence were Z-scored and~~ a constrained deconvolution was applied (Pachitariu et al. 2016) to return a continuous spiking estimate. This preprocessing strategy and criteria were optimized from the freely available jRGECO1a dataset from the CRCNS website (Dana et al. 2016, Mohar et al. 2016) to match the state of the art algorithmic performance in event detection (Berens et al. 2018).”

REVIEWERS' COMMENTS

Reviewer #1 (Remarks to the Author):

I appreciate the thoughtful and complete responses to my concerns (and the concerns of the other reviewers). I believe this is an important study that deserves publication, with one minor additional suggestion.

In Figure S7B, I encourage the authors to show the results from the full population. The response to reviewers mentions that "we found that the interaction term did not change decoding of the entire population". I believe that this is important to show. It doesn't minimize the importance of this study, but rather suggests a more nuanced interpretation of the conditions in which MW suppression may impact downstream decision making.

Reviewer #2 (Remarks to the Author):

The authors have improved the manuscript very substantially taking advantage of the suggestions and views of all reviewers. They should be commended for their major effort. The interpretation of the results is much clearer and the results are much more solid.

Reviewer #3 (Remarks to the Author):

The authors sufficiently addressed our concerns. However, I believe that the additional comments below will further improve the clarity and presentation of the paper. I will strongly support the acceptance of the MS if the comments below will be addressed.

Moderate

1. In 5c, I understand you picked the threshold on the R_s that maximizes the AUC (according to methods). However, I still think you should mention this in text, along with the mean and s.e.m. of the threshold for each case. There is also something I am still not clear about – R_{w2} would presumably be negatively correlated with $F1-F2$, so did you threshold trials by having a lower R_{w2} value than a certain threshold? If so, state this in the methods and the text.

2. It is not clear to me how what you are doing in 6F is linear discriminant analysis (which involves solving for the eigenvectors of the between-class scatter matrix). Seems rather to be a straightforward application of plotting the differences in Rs (sums of weights from the linear model) of stimulus and choice against each other. If it is LDA, please elaborate more on this in the methods section.

3. It seems the y-labels in Fig. 7G and H are incorrect. I believe they should be “ratio engaged/disengaged”, since it’s not the firing rates you are using but rather you compare the R values themselves in the engaged and disengaged states.

4. It is unclear why in the discussion you hypothesize “that presence of choice signals in primary sensory areas depends mostly with the sensory stimulation parameters, with near detection threshold stimuli leading to highest choice predicting activity”, since earlier in the introduction you correctly noted that the report itself is associated with widespread cortical activation. It thus seems more reasonable to conclude that, at least in mice, the fact that some studies have found very sparse/non-existent choice coding in the primary sensory cortex is a result of the report modality (go/no-go vs. lick-right/lick-left) rather than the sensory task (detection vs. discrimination). In fact, Steinmetz et al. 2019 directly shows this – every region recorded encoded whether the mouse made a report or not, while the left/right encoding was very sparse indeed.

* That said, if you do want to stick with this interpretation because it’s more consistent with the NHP studies, I would at least raise both interpretations and state that it is still unclear which one is correct. It is not difficult to imagine how one could test this – make mice perform a detection task with a lick-right/lick-left report.

Minor

1. “go/nogo -> go/no-go”

2. In line 51, add “but see Buetfering et al. 2022”.

3. In line 55 you mix report-related activity during the go/no-go paradigm with activity related to facial movements. They are of course related, but not the same – the report (licking) involves some facial movement, but facial movement can occur without the report. So Stringer et al. isn’t really appropriate to cite. I would instead suggest you talk specifically about activity related to the report, and cite Musall (which finds both instructed and uninstructed movement correlates) and Salkoff et al. 2020 – this paper most directly shows widespread report-related activity during the go/no-go task.

4. The population representations of sensory vs. choice coding have been compared by Raposo et al. 2014 in the PPC, where they found similar results (i.e. orthogonality and co-existence of two types of coding in the same population). This is an important result to cite in the discussion, and perhaps also in line 500.

5. “wild-field’ -> “widefield”

6. “r2” -> “r²”

7. Question mark characters in several places, e.g. line 477

8. y-label in Fig. 7F should be “FR ratio” not “fraction FR”.

9. Line 760 – “whisker’s” -> “whiskers”

10. Line 706 – “mostly with” -> “mostly on”

11. P5L184 “Finally, adding increasing adjacent whisker’s stimulation over principal whisker’s stimulation held constant, seemed to lower the response of the principal whisker’s population (Fig. 2g), suggesting a suppressive effect of multi-whisker stimulation.” – Is the effect of the adjacent whisker significant? From the figure it seems not to be the case, so perhaps consider removing this statement. Otherwise, show more support to this claim (maybe other frequencies 45/0, 45/45, 45/90) or dismiss it.

12. (previous remark) “Present the complementary stimulation frequencies” – It seems that the request wasn’t clear enough. Add to the figure the complementary data, where the AW stimulation is increasing, and the PW stimulation is zero. (ilan please see if I understand correctly that they didn’t reply to that)

Reviewer #4 (Remarks to the Author):

In this revision, all my major concerns have been addressed, including use of appropriate statistical tests and clarifying several things that were not (e.g., depth of imaging). The paper as it stands is a substantial contribution to the field and I recommend publication.

Color and formatting code:

Original comments from the reviewer*

Response from the authors

“Original manuscript text”

Addition to the original manuscript

Suppression of text

Upon proofreading and re-analysis following reviewer’s and editor’s request, we identified several minor errors in our manuscript. None of these modifications impact our text nor the conclusions (beyond the modifications proposed here). We have implemented these changes to ensure the accuracy of our work.

(1) We identified an error in the reported p-value for Figure 3I ($p = 0.041$; GLME model p value for W2 blocking versus Sham). The correct p-value is already documented in Supplementary Table 2 ($p < 0.001$). A correction has been made in the text ($p < 0.001$ ~~$p = 0.041$~~). The W2 blocking effect versus sham was described as significant so this change doesn’t impact the text or conclusion.

(2) We identified an error in the text and Supplementary Table 2: statistics for Fig 6B were erroneously attributed to $n = 3118$ neurons instead of $n = 9$ FOV (comparison of CPs in the population of W1 selective neurons versus the population of W2 selective neurons). In our statistical test, the population of W1 selective and W2 selective neurons were averaged for each FOV before running the LME model test. We re-ran the analysis using $n = 3118$ neurons which yield a comparable statistical outcome to $n = 9$ FOV (no significant differences in CPs, with respectively of $p = 0.06$ and $p = 0.64$; comparing the mean CPs for w1 and w2 selective neurons populations; CPs = 0.481 and 0.486 respectively). We have corrected the table and text to reflect the original test " $n = 9$ FOV. ~~$n = 3118$ neurons~~ "

(3) In the text for Fig 2 we made a correction, as the average of activated neurons (“48% +- 4.7”) was not averaged across $n = 11$ FOV but directly across the whole population of $n = 3706$ neurons. We modified this proportion to have it across $n = 11$ FOV as described initially “a larger fraction of the population had significant response compared to baseline: 32.3 ± 4.8 % of neurons were suppressed and ~~48.1 ± 4.7 %~~ 52.3 ± 4.7 % of neurons were activated, mean \pm s.e.m across $n = 11$ FOV, all trials included.” The fraction of inhibited neurons is correct.

(4) In figure 4B, we used Spearman correlation instead of the Pearson. The text was updated accordingly. Similarly in the text corresponding to figure 2 we report now “Spearman’s $r = 0.49$; $p < 0.001$ ” instead of “ ~~$r^2 = 0.49$~~ ”.

Reviewer #1 (Remarks to the Author):

I appreciate the thoughtful and complete responses to my concerns (and the concerns of the other reviewers). I believe this is an important study that deserves publication, with one minor additional suggestion.

In Figure S7B, I encourage the authors to show the results from the full population. The response to reviewers mentions that "we found that the interaction term did not change decoding of the entire population". I believe that this is important to show. It doesn't minimize the importance of this study, but rather suggests a more nuanced interpretation of the conditions in which MW suppression may impact downstream decision making.

We added entire population decoding with/without interaction term in Fig. S7B.

Reviewer #2 (Remarks to the Author):

The authors have improved the manuscript very substantially taking advantage of the suggestions and views of all reviewers. They should be commended for their major effort. The interpretation of the results is much clearer and the results are much more solid.\

Reviewer #3 (Remarks to the Author):

The authors sufficiently addressed our concerns. However, I believe that the additional comments below will further improve the clarity and presentation of the paper. I will strongly support the acceptance of the MS if the comments below will be addressed.

We thank the reviewer for the additional comments and questions that further improve the clarity of the manuscript.

Moderate

1. In 5c, I understand you picked the threshold on the Rs that maximizes the AUC (according to methods). However, I still think you should mention this in text, along with the mean and s.e.m. of the threshold for each case. There is also something I am still not clear about – Rw2 would presumably be negatively correlated with F1-F2, so did you threshold trials by having a lower Rw2 value than a certain threshold? If so, state this in the methods and the text.

As pointed out by the reviewer, we threshold – Rw2 instead of Rw2; We updated the figure label to “– Rw2” and the figure caption as :

“C, Neurometric and psychometric functions compared. Left: psychometric/neurometric curves. We decoded negative Rw2 so neurometric curves are in the same direction. Right: Comparison of the fitted slope and lapse rate of the three psychometric/neurometric functions. “

Concerning the threshold for decoding, the mean and s.e.m. of the threshold are very near 0 when decoding Rw1 – Rw2. Specifically $5.5 \cdot 10^{-4} + / - 2.2 \cdot 10^{-3}$ for engaged/ disengaged and $8.1 \cdot 10^{-4} + / - 2.7 \cdot 10^{-3}$. We included these thresholds in the figure 5DE caption.

2. It is not clear to me how what you are doing in 6F is linear discriminant analysis (which involves solving for the eigenvectors of the between-class scatter matrix). Seems rather to be a straightforward application of plotting the differences in Rs (sums of weights from the linear model) of stimulus and choice against each other. If it is LDA, please elaborate more on this in the methods section.

The LDA analysis was performed but not displayed in the figure, nor detailed in the text. It was an additional control that the axis that separates best the trial by trial representations of the two choices is orthogonal to the sensory discrimination axis. For simplicity, and because the conclusion from this LDA analysis (i.e. sensory/choice orthogonality) is redundant with the conclusion from the “representational angle” analysis (sensory/choice orthogonality), we removed the statement referring to the LDA analysis in the text:

“To visualize this relationship, we selected two dimensions of relevance that separate best the stimulus target side and response side respectively (Rw1 - Rw2 and Rc1 - Rc2, Fig. 6f). ~~Overall, we observe that the best separation of left and right choice trials is almost parallel to the sensory axis (Linear discriminant analysis, LDA).~~ A “representational angle” is calculated from the translation of left and right choices in the sensory and choice dimensions. In accordance with our previous observations, choice and sensory representations are encoded orthogonally, (90.2° and 90.0° on average for left and right target trials, n= 2300 and n= 2152 trials; Fig. 6f). “

3. It seems the y-labels in Fig. 7G and H are incorrect. I believe they should be “ratio engaged/disengaged”, since it’s not the firing rates you are using but rather you compare the R values themselves in the engaged and disengaged states.

As suggested by the reviewer, we updated the label to “FR ratio” rather than “fraction FR” in 7H. and “activity ratio” instead of “fraction FR” in 7GH (R values are weighted averages of neuronal firing rates) Legend of the figure was also updated.

4. It is unclear why in the discussion you hypothesize “that presence of choice signals in primary sensory areas depends mostly with the sensory stimulation parameters, with near detection threshold stimuli leading to highest choice predicting activity”, since earlier in the introduction you correctly noted that the report itself is associated with widespread cortical activation. It thus seems more reasonable to conclude that, at least in mice, the fact that some studies have found very sparse/non-existent choice coding in the primary sensory cortex is a result of the report modality (go/no-go vs. lick-right/lick-left) rather than the sensory task (detection vs. discrimination). In fact, Steinmetz et al. 2019 directly shows this – every region recorded encoded whether the mouse made a report or not, while the left/right encoding was very sparse indeed.

* That said, if you do want to stick with this interpretation because it’s more consistent with the NHP studies, I would at least raise both interpretations and state that it is still unclear which one is correct. It is not difficult to imagine how one could test this – make mice perform a detection task with a lick-right/lick-left report.

We updated the text as follow to argument our hypothesis better and include the alternative interpretation:

“The relatively subtle encoding of choice observed here in wS1 is congruent with previous studies of vibrotactile discrimination in non-human primates (Romo and de Lafuente 2013). However, it may seem contradictory with more recent results in the murine model, showing a large difference in evoked activity between hits and misses before the onset of licking in a go/no-go paradigm, and correlation between sensory and choice activity (Kwon et al. 2016; Yang et al. 2016). It is still unclear whether these discrepancies are due to the different animal models, the nature of the perceptual tasks or the different behavioral paradigms. During detection of near threshold stimuli, activity in the primary sensory cortices convey choice signals, in both primates (Palmer et al. 2007; van Vugt et al. 2018), and rodents (Yang et al. 2016). In a situation when animals are asked to perform discrimination of multiple visual stimuli, the choice might actually be poorly predicted from the primary visual cortex activity (Steinmetz et al. 2019). **In our paradigm, trials with response only slightly increase the total L2/3 activity, arguing against a large contribution of motivation and motor related activity on total activity change.** Therefore, we hypothesize that presence of **strong** choice signals in primary sensory areas depends mostly ~~with~~ **on** the sensory stimulation parameters, with near detection threshold stimuli leading to highest choice predicting activity. **A distinction between detection and discrimination has been made in non-human primate studies (Nienborg et al. 2012) that could help design studies in rodent research: To ideally disambiguate representation of choice from motor related activity, motivation of reward expectations, future studies could probe near threshold stimulus detection with a 2AFC and using task reversal. ~~It is important to distinguish between signals related to detection and those related to discrimination (Nienborg et al. 2012).~~”**

Minor

1. “go/nogo -> go/no-go”

Corrected L54

2. In line 51, add “but see Buetfering et al. 2022”.

This study is referenced in the discussion and its results discussed specifically L677 and L739. It seems to us that the study from Buetfering does not contradict our statement L54 that it is difficult to assess the causality of choice signals in S1 on actual choice.

3. In line 55 you mix report-related activity during the go/no-go paradigm with activity related to facial movements. They are of course related, but not the same – the report (licking) involves some facial movement, but facial movement can occur without the report. So Stringer et al. isn’t really appropriate to cite. I would instead suggest you talk specifically about activity related to the report, and cite Musall (which finds both instructed and uninstructed movement correlates) and Salkoff et al. 2020 – this paper most directly shows widespread report-related activity during the go/no-go task.

We thank the reviewer for the reference Salkoff et al 2020, and updated the text accordingly.

The widespread cortical activity related to onset of facial movement (Musall et al. 2019; ~~Stringer et al. 2019~~; Salkoff et al 2020)– including in sensory areas– renders ambiguous the signals associated with choice if sensory integration and motor response are temporally overlapping.

4. The population representations of sensory vs. choice coding have been compared by Raposo et al. 2014 in the PPC, where they found similar results (i.e. orthogonality and co-existence of two types of coding in the same population). This is an important result to cite in the discussion, and perhaps also in line 500.

We thank the reviewer for the important reference and added it to the discussion:

“Instead, our results add to the growing evidence that choice coding lies in dimensions orthogonal to that of sensory representation, enabling multiplexing of information in different neuronal subspaces (Raposo et al. 2014; Zhao et al. 2020; Buetfering et al. 2022). This pattern is allowed by the uncorrelated coding of sensory and choice variables at the single neuron level ,...”

5. “wild-field’ -> “widefield”

Corrected L228

6. “r2” -> “r²”

7. Question mark characters in several places, e.g. line 477

May it be a formatting problem of the “Beta” glyph? We used the glyphs β or β

8. y-label in Fig. 7F should be “FR ratio” not “fraction FR”.

Corrected (see our response to Minor 1)

9. Line 760 – “whisker’s” -> “whiskers”

Corrected L760

10. Line 706 – “mostly with” -> “mostly on”

Corrected L706

11. P5L184 “Finally, adding increasing adjacent whisker’s stimulation over principal whisker’s stimulation held constant, seemed to lower the response of the principal whisker’s population (Fig. 2g), suggesting a suppressive effect of multi-whisker stimulation.” – Is the effect of the adjacent whisker significant? From the figure it seems not to be the case, so perhaps consider removing this statement. Otherwise, show more support to this claim (maybe other frequencies 45/0, 45/45, 45/90) or dismiss it.

It shows only a non-significant trend for suppression ($p= 0.21$) that we found valuable to describe, since AW stimulation in the absence of PW lead to a significant opposite effect (increased PW. Pop. activity $p= 0.009$), suggesting non-linear interactions. Besides, this multi-whisker suppression trend is confirmed later at the single neuron level using statistical power of the linear model analysis in Fig4B. We considered Fig. 2B trend as a hint for the need of the interaction term later explored in Figure 4.

12. (previous remark) “Present the complementary stimulation frequencies” – It seems that the request wasn’t clear enough. Add to the figure the complementary data, where the AW stimulation is

increasing, and the PW stimulation is zero. (ilan please see if I understand correctly that they didn't reply to that)

It seems that our explanations lacked clarity. The complementary data is already displayed by the dashed line in figure 2g. However the labels are 'inverted' compared to the reviewer's request. i.e, in Fig. 2g, the bottom left dashed line is the adjacent population activity when $AW = 0$ and PW is increased, which is the same as the principal population activity when $PW = 0$ and AW is increased. Similarly, the bottom right dashed line is the Adjacent population activity when $PW = 90$ and AW is increased, which is the same as principal population when $AW = 90$ and PW is increased. These are numerically the exact same, because stimulus design is symmetric and we included all cortical column as both adjacent pop. and principal pop averages, (18 column from 9 FOV with these stimulation conditions). We added in the figure caption : "N = 18 columns from 9 FOV."

Reviewer #4 (Remarks to the Author):

In this revision, all my major concerns have been addressed, including use of appropriate statistical tests and clarifying several things that were not (e.g., depth of imaging). The paper as it stands is a substantial contribution to the field and I recommend publication.